# Predictive Performance is Often Insensitive to Feature Selection in High-Dimensional Biological Classification

## Abstract

Feature selection (FS) is commonly assumed to improve predictive performance and to identify meaningful features in high-dimensional data. We systematically evaluate this assumption across 30 classification benchmarks, primarily drawn from computational genomics, including microarray, bulk RNA-Seq, mass spectrometry, and imaging datasets. Across these datasets, we observe that small random subsets of features (0.02–1% of available features) frequently achieve predictive performance that is comparable to, and in some cases statistically indistinguishable from, models trained on full feature set. Notably, performance variability across random subsets of a given size is often low, suggesting substantial redundancy in the predictive signal. Together, these results suggest that, for many widely used high-dimensional biological benchmarks, predictive accuracy alone is not sufficient to justify claims about the importance of specific selected features. They also underscore the need for rigorous validation before interpreting selected features as biologically meaningful or actionable, particularly in computational genomics.

## 1. Introduction

Feature selection (FS) is considered a critical part of nearly all machine learning research on high-dimensional datasets. Beyond just computational efficiency, FS is often used to identify features deemed "important" to a given outcome, especially in fields such as computational genomics. Some highly cited, influential studies which do this include Golub et al. (1999); Guyon et al. (2002; 2004); Li et al. (2017); Lall & Bandyopadhyay (2019); Cilia et al. (2019); Chen & Dhahbi (2021); Zanella et al. (2022). As of January

[1]Anonymous Institution, Anonymous City, Anonymous Region, Anonymous Country. Correspondence to: Anonymous Author <anon.email@domain.com>.

Preliminary work. Under review by the International Conference on Machine Learning (ICML). Do not distribute.

2026, a Google Scholar search for *"feature selection on high dimensional datasets"* lists about 3, 010, 000 results - it is beyond doubt that large resources are being invested in this area.

The aim of such work is typically to identify a small subset of features which, if used to train a machine-learning model, results in high accuracy (sometimes surpassing that of the full feature set). These cleverly-selected features are also often deemed to be important to the underlying task. Strikingly, we found that a critical baseline is almost always missing: a simple null hypothesis, comparing the results against a random subset of features of the same size. Without this comparison, it is unclear whether sophisticated FS methods truly outperform chance.

In this work, we systematically evaluate the null hypothesis of random feature selection across datasets varying in sample size, feature dimensionality, number of classes, and data modality. Across many settings, models trained on all features achieve performance that is matched by small (as low as $0.02\% - 1.0\%$) random feature subsets. Consequently, attempts to reject the null hypothesis of random feature selection frequently fail. We find that in many high-dimensional biological datasets, predictive performance is largely insensitive to the specific features chosen, with low variance across random subsets. In practice, small random subsets often match, and in some cases outperform, all features or features selected by established methods.

Across 27 out of 30 diverse datasets that we tested (dataset inclusion criteria specified in Section 3.2), including microarray, RNA-Seq (bulk and single-cell), mass spectrometry, and imaging, we find that extremely small, random subsets of features ($0.02 - 1\%$ of all features) match or outperform the predictive performance of full feature sets. The only three cases where full feature set does better than chance are in journalistic text categorization, drug activity prediction, and synthetic data. Results for all 30 datasets are shown in Table 1.

We also find that, for a given dataset, performance (accuracy or AUC) obtained from randomly sampled feature subsets rapidly stabilizes as subset size increases, exhibiting low variance beyond a dataset-specific scale. Sampling with or

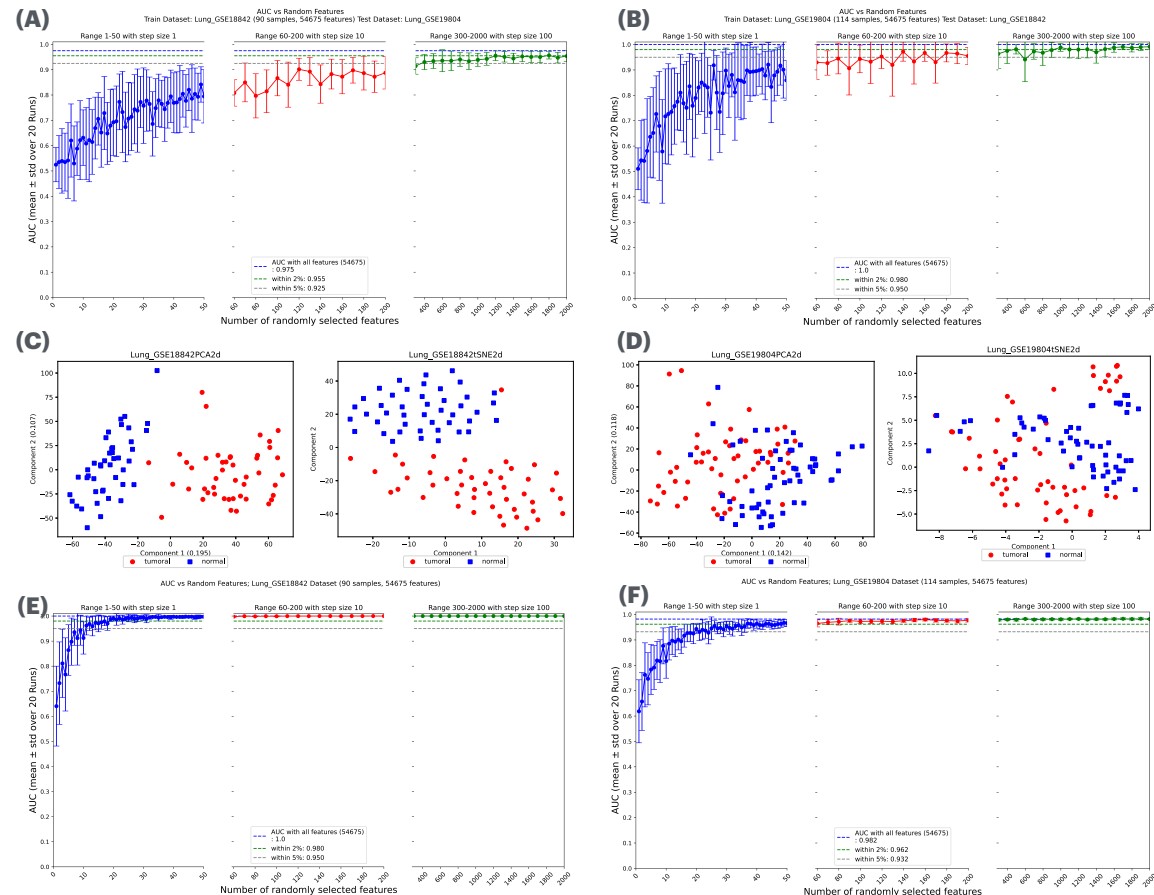

*Figure 1.* RF results on lung cancer **microarray** datasets (A) Training on GSE18842 and testing on GSE19804 shows that random subsets of only 400 features (∼0.8% of all features) achieve AUC comparable to using all features. (B) Training on GSE19804 and testing on GSE18842 shows similar performance with just 200 (∼0.4%) randomly selected features. (C–D) PCA and t-SNE visualizations illustrating class separation in GSE18842 and GSE19804, respectively. (E–F) With an 80:20 train-test split, subsets of only 20 features (∼0.04%) for GSE18842 and 100 features (∼0.2%) for GSE19804 match the performance of the full feature set.

without replacement seems to make little difference. To formalize this behavior, we introduce Minimum Sufficient Random Sample Size (MSRSS) (Section 3.4.2), defined as the minimum number of randomly sampled features such that increasing the subset size yields less than $\epsilon$ relative improvement in mean performance across random draws, where $\epsilon \in [0.02, 0.05]$, relative to the asymptotic performance.

In our analysis, for each dataset, we compare model performance when trained on:

1. all features.

2. randomly selected subsets of features

   - of the same size as in the published feature selection (FS) studies, for all datasets where FS results are available,

   - across various subset sizes ranging from 1 to 2000 features

3. features selected using standard feature selection methods

This emphasizes the importance of rigorous validation before interpreting selected features as biologically or functionally significant. The work carries broad implications across genomics, neuroscience, imaging, and other fields that rely on high-dimensional data for discovery and prediction.

We realize that this is a very strong claim to make [1] ; as such, all of our code and sample datasets are provided at the following anonymous GitHub link: https://anonymous.4open.science/r/

---

[1]We asked multiple other teams to independently rewrite all code and re-test everything, with similar results.

Feature_Selection_HD-D853/. All the datasets used in our experiments are publicly available and their original sources are cited in appendix A.

This work makes the following contributions:

1. We present a large-scale empirical study across 30 high-dimensional datasets to evaluate the effectiveness of randomly selected feature subsets for classification tasks, comparing full feature set results against random subsets.

2. We find that randomly selected subsets, sometimes comprising as little as $0.02\% - 1\%$ of features, can match or exceed the performance of models trained on all features.

3. We discuss the broader implications of these findings, including discussing possible metrics (MSRSS) for evaluating datasets and FS methods, understanding variance in model performance and interpreting computational results in genomics and other high-dimensional domains.

## 2. Related Work

Early work on feature selection in genomics laid the foundation for much of the field. Seminal studies showed how gene expression data could support molecular cancer classification (Golub et al., 1999), how SVM-based gene selection could be applied to high-dimensional biology (Guyon et al., 2002). Gene expression profiles were also applied to predict breast cancer outcomes (West et al., 2001), underscoring the clinical promise of FS. More recent methods include manifold-preserving FS for detecting rare cell types in single-cell data (Liang et al., 2021) and Wx, a neural network–based FS algorithm for transcriptomic datasets (Park et al., 2019). Applied studies have used overlapping FS to distinguish lung cancer subtypes (Chen & Dhahbi, 2021) and $\ell_1$-norm copula-based FS for microarray data (Lall & Bandyopadhyay, 2019), and pan-cancer classification of TCGA expression data (Li et al., 2017).

Additionally, comparative evaluations have sought to benchmark FS methods. For example, Guyon et al. (2004) reported the results of the NIPS 2003 FS Challenge, providing one of the earliest systematic benchmarks of FS methods across diverse datasets, Zanella et al. (2022) evaluated FS algorithms for cancer phenotype classification, and Cilia et al. (2019) provided a systematic evaluation of FS and classification methods on microarray datasets. Yet, performance has rarely been compared against a simple null baseline of randomly chosen feature subsets, leaving open a critical question of whether complex FS methods genuinely outperform chance.

Additionally, while signature multiplicity theory (Statnikov & Aliferis, 2010; Borboudakis & Tsamardinos, 2021; Statnikov et al., 2013) shows that several distinct but causally meaningful gene sets may achieve similar predictive performance, our results go even further: the accuracy of any random feature subset is comparable to selected features across many datasets. This indicates that predictive signal is highly redundant and distributed - to the point where feature selection may provide little practical advantage in high-dimensional biological classification tasks. Thus our results are stronger than presented in these papers. To the best of our knowledge, previously no one has ever shown a universal random subset is good enough - there is no cleverness involved (which is the whole point, making the FS results fail in terms of significance vs the null hypothesis.).

## 3. Method

### 3.1. Datasets

We conducted our experiments on a diverse set of 30 high-dimensional datasets, spanning multiple data modalities: 21 microarray gene expression, 4 RNA-Seq (bulk and single-cell), 1 mass spectrometry, 2 image, and 2 other data types( Table 1, sorted by sample size, with NIPS 2003 FS challenge datasets listed at the end (Guyon et al., 2004)). We intentionally chose this heterogeneous collection of datasets spanning multiple cancer types and molecular profiling platforms, varying widely in tissue origin, sample size, feature dimensionality, datatypes and class distribution. In addition, we include five benchmark datasets from the NIPS feature selection challenge, covering domains such as cancer prediction via mass-spectrometry data, handwritten digit recognition, text classification, and molecular activity prediction along with one synthetically generated dataset. Notably, all five of these challenge datasets (marked with † in Table 1) were constructed with random probe features intentionally added as distractors, making them especially relevant for evaluating the robustness of feature selection methods. Most of the microarray dataset are sourced from Feltes et al. (2019), which curated them specifically for ML model training. All the datasets are publicly available. None were imputed before download, and we did not apply an imputation. For all the datasets, the original sources are cited in Table 3 in the Appendix A.

### 3.2. Dataset Inclusion Criteria

We include datasets only if they satisfy the following:

1. at least 100 samples (three canonical FS benchmarks, ALL/AML, Colon and Madelon are included despite slightly fewer samples due to their widespread use),

2. at least 2,000 features,

*Table 1.* Summary of datasets and random subset size (percentage of full features) required to match full feature performance within 5%. For 27 out of 30 datasets, randomly selected feature subsets can match or even outperform models trained on the full feature set.

| S No | Name | Classes | Samples | Features | Random Subset Size (matching percentage) | Datatype | Domain |
|---|---|---|---|---|---|---|---|
| 1 | Colon (Alon)* | 2 | 62 | 2000 | 100 (5%) | microarray | colon cancer |
| 2 | ALL/AML* | 2 | 72 | 7129 | 200 (3%) | microarray | leukemia |
| 3 | GSE6008 | 4 | 98 | 22283 | 45 (0.2%) | microarray | ovarian cancer |
| 4 | GSE18842 | 2 | 90 | 54675 | 20 (0.04%) | microarray | lung cancer |
| 5 | GSE42743 | 2 | 103 | 54675 | 200 (0.4%) | microarray | oral cavity cancer |
| 6 | GSE19804 | 2 | 114 | 54675 | 100 (0.2%) | microarray | lung cancer |
| 7 | GSE6919_U95B | 2 | 124 | 12620 | 120 (1%) | microarray | prostate cancer |
| 8 | GSE3365 | 3 | 127 | 22814 | 300 (1.5%) | microarray | bowel disease |
| 9 | GSE50161 | 5 | 130 | 54575 | 60 (0.1%) | microarray | brain tumours |
| 10 | GSE22820 | 2 | 139 | 33579 | 50 (0.15%) | microarray | breast cancer |
| 11 | GSE53757 | 2 | 143 | 54675 | 60 (0.1%) | microarray | kidney cancer |
| 12 | GSE30219 | 2 | 146 | 54675 | 10 (0.02%) | microarray | lung cancer |
| 13 | GSE21510 | 3 | 147 | 54675 | 60 (0.1%) | microarray | colorectal cancer |
| 14 | GSE45827 | 6 | 151 | 54675 | 60 (0.1%) | microarray | breast cancer |
| 15 | GSE76427 | 2 | 165 | 47322 | 100 (0.2%) | microarray | liver cancer |
| 16 | GSE4115 | 2 | 187 | 22215 | 110 (0.5%) | microarray | lung epithelial |
| 17 | GSE44076 | 2 | 194 | 49386 | 50 (0.1%) | microarray | colorectal cancer |
| 18 | GSE11223 | 3 | 202 | 40991 | 200 (0.5%) | microarray | colon inflammation |
| 19 | GSE28497 | 7 | 281 | 22284 | 110 (0.5%) | microarray | pediatric leukemia |
| 20 | GSE70947 | 2 | 289 | 35981 | 110 (0.3%) | microarray | breast cancer |
| 21 | GSE14250 | 2 | 357 | 22277 | 50 (0.2%) | microarray | liver cancer |
| 22 | TCGA (LUAD/LUSC) | 2 | 1016 | 20253 | 50 (0.3%) | bulk RNASeq | lung cancer |
| 23 | TCGA pan-cancer | 33 | 10223 | 20253 | 50 (0.25%) | bulk RNASeq | pan-cancer |
| 24 | Lung (scRNA-Seq) | 9 | 20966 | 33514 | 1600 (5.0%) | scRNA-Seq | lung adenocarcinoma |
| 25 | Lung (scRNA-Seq) | 2 | 24421 | 33514 | 1200 (3.6%) | scRNA-Seq | lung adenocarcinoma |
| 26 | Arcene† | 2 | 200 | 10000 | 100 (1.0%) | mass-spectrometry | ovarian vs prostate cancer |
| 27 | Dexter† | 2 | 600 | 20000 | **20000 (100%)** | text | corporate acquisition |
| 28 | Dorothea† | 2 | 1150 | 100000 | **100000 (100%)** | fingerprint | drug activity prediction |
| 29 | Madelon†* | 2 | 2600 | 500 | **200 (40.0%)** | synthetic | cluster classification |
| 30 | Gisette†* | 2 | 7000 | 5000 | 90 (1.8%) | image | digit (4 vs 9) |

† NIPS 2003 FS Challenge datasets; all five datasets contain random probes, ranging from 30% (Arcene) ∼ 50% (Dexter, Gisette, Dorothea) to 96% (Madelon).
* indicates subsets sampled with replacement (datasets with < 20,000 features).

3. prior use in feature-selection studies. We have included add datasets from NIPS 2003 FS challenge (Guyon et al., 2004).

### 3.3. Models

We evaluate a representative set of models covering major learning paradigms: linear (Logistic Regression (LR), Ridge), margin-based (Support Vector Machine (SVM)), Decision Tree (DT), ensemble trees (Random Forest (RF), Gradient Boosting Classifier (GBM), HistGradient Boosting Classifier (HistGB), eXtreme Gradient Boosting (XGB)), and neural networks (Multilayer Perceptron (MLP)). This selection balances simple baselines with non-linear models, and overlaps with models used in prior work, enabling direct comparison. We exclude Naive Bayes, SGD and KNN, which perform poorly or scale unfavorably in high-dimensional settings, and omit additional boosting variants (LighGBM, CatBoost) as redundant given XGB, GBM, and HistGB. Our goal is representative, not exhaustive, coverage.

### 3.4. Experimental Setup

All datasets were split into train/test subsets using an 80/20 stratified split. For non-tree-based models, the features were standardized priot to feature selection and model training. For each dataset, we compared model performance using

1. all features

2. random feature subsets (baseline) as described in next section

3. embedded (lasso, elastic net) and ensemble-based (random forest importance) feature selection methods

4. reported results from prior published feature selection studies, where available

3.4.1. Random Feature Subsets

We sampled features subsets of various sizes $(1, 2, \ldots, 50; 60, 70, \ldots, 200; 300, 400, \ldots, 2000)$, resulting in a total of 83 distinct subset sizes. When the total number of features exceeded 20,000, subsets were drawn without replacement; otherwise, sampling was with replacement. For each subset size, we generated 20 independent subsets by random sampling and repeated the full training–evaluation procedure on each. We then trained and evaluated a classifier on every subset, reporting mean accuracy and AUC across the 20 runs. The plots in figures 1, 2, 3 display these results, with error bars indicating the standard deviation across runs.

3.4.2. Evaluation Metrics

We present our results with Area Under the ROC Curve (AUC-ROC) and Accuracy as the primary metrics. For random subsets, we report the mean performance over 20 runs. For each subset size, standard deviation is shown with an error bar. Each plot has three horizontal reference lines: for AUC (or Accuracy) with all features; for within-$2\%$; and within-$5\%$ AUC (or Accuracy).

One interesting metric to explore would be the minimum subset size of randomly-selected features which contain enough information to perform as well at the task as any larger subset - that is, the accuracy becomes asymptotic after that point, indicating a "saturation" of information provided by these random subsets. Interestingly, we notice that this is model-independent, remarkably low in performance variance (across different random selections), and almost always matching full-feature set performance (after all, this is the random subset size with maximum cardinality). We call the random subset feature size at which this happens the *Minimum Sufficient Random Subset Size (MSRSS)*; graphically, this captures the "elbow" at which the information provided by random subsets saturates, resulting in asymptotic, flat performance thereon. We believe that exploring this metric could provide interesting insights into various datasets, especially in computational genomics.

## 4. Results

We begin by showing Random Forest (RF) results on one representative dataset (or pair, if different datasets exist for train and test) from each type—microarray, bulk RNA-Seq, single-cell RNA-Seq, and imaging—chosen to illustrate breadth. Across 27 out of 30 datasets tested, we find that very small, randomly selected subsets of features $(0.02-5\%$ of all features) match or even outperform the predictive performance of full feature set. A full summary appears in table 1, with complete plots for all datasets using Random Forest in appendix B (and results using other models like Decision trees, Support Vector Machine, Logistic Regression etc. are in appendix C).

### 4.1. Random Forest results with cross-dataset evaluation

We start with two independent microarray datasets of lung disease, each measured on an identical set of features. To assess model performance under a strict train-test separation, one dataset was designated as the training set and the other as the external test set. This design avoids overfitting to dataset-specific noise and provides a rigorous test of generalization. Model performance was quantified primarily by the Area Under the ROC Curve (AUC); classification accuracy is reported in appendix B. To ensure robustness and rule out dataset-specific biases, we repeated the experiment

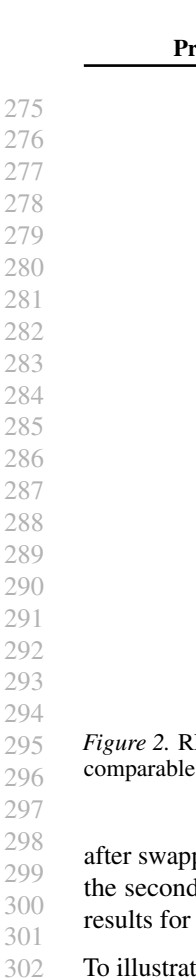

*Figure 2.* RF results on Gisette **image** dataset. (A) Randomly selected subsets of only 90 features ( 1.8% of all features) achieve AUC comparable to using the full feature set when training on the Gisette training data and evaluating on the validation set.

after swapping the roles of the two datasets (i.e., training on the second dataset and testing on the first), and we report results for both directions.

To illustrate, we first train a model using GSE18842 (Feltes et al., 2019) (90 samples, 54,765 features) and test it using GSE19804 (Feltes et al., 2019)(114 samples, 54,765 features). As shown in figure 1(A), randomly selected subsets of just 200 features ( 0.4% of all features), selected without replacement, achieve AUC comparable to using all features. figure 1(B) shows the reverse setting, where the model is trained on GSE19804 and tested on GSE18842. Both plots show similar results. The 2D projections of these two datasets using PCA and t-SNE show that GSE18842 exhibits better class separability than GSE19804. We hypothesize that such low-dimensional separability enables models trained on small random subsets of features to perform well. Accuracy plots for this dataset pair show similar trends in the appendix B: even very small random subsets can match the performance of models using all features.

We also show results for non-disease dataset, Gisette, with cross-dataset evaluation shown in Figure 2. The Gisette image dataset is from the NIPS 2003 FS challenge (Guyon et al., 2004). The task is to discriminate between two confusable handwritten digits: the four and the nine. The challenge organisers provided a separate training and validation set. We find that with 90 (1.8%) randomly selected features, the models could match the AUC with all features. As there are 30% spurious features in this dataset, the results are even more significant – effectively, any random subset of size 60

is enough to discriminate between the two classes in this dataset.

### 4.2. Random Forest results with intra-dataset splits

In addition to cross-dataset evaluation, we assessed model performance using standard intra-dataset train–test splits of 80:20. For each dataset, we randomly divided the samples into 80% for training and 20% for testing, while ensuring class balance was preserved. For example, figures 1(E) and 1(F) show that for GSE19804 and GSE18842, with a random subset we can match the performance of all features.

Figure 3 summarizes results for bulk and Single-cell RNA-Seq datasets. For the bulk RNA-Seq data, (Figure 3(A)), Random Forest models trained with just 50 randomly selected features ($\sim$0.22% of all features) achieves performance comparable to the full feature set. For the Single-cell RNA-Seq dataset(Figure 3(E)), subsets of about 1200 ($\sim$3.6% of all features) results in AUC within 5% of the full-feature set. We hypothesize that better low-dimensional separability for TCGA_LUAD_LUSC (bulk RNA-Seq dataset) enables random subsets to perform better. Even for a bulk RNA-Seq dataset with 33 classes (TCGA), random subsets can match the performance of all features as shown in figure 26 in appendix B. **Note:** We also report sparsity via three metrics: (i) global fraction of zeros, (ii) median zero fraction per feature, and (iii) median zero fraction per sample. The scRNA-Seq datasets are extremely sparse ($> 96\%$ global zeros), while bulk rna-seq shows near-zero sparsity. Crucially, random feature subsets still match full-model performance

*Table 2.* Comparison with published studies. Accuracy (%) with number of features in parentheses.

| Publication | Dataset (samples) | Published FS | Random FS baseline | All features | Standard FS baselines | | | Remarks |
| --- | --- | --- | --- | --- | --- | --- | --- | --- |
| | | | | | Lasso | Elastic Net | RF-Imp. | |
| Chen & Dhahbi (2021) | TCGA (LUAD/LUSC) (1016) | 94.2 (500) | 93.6 (500) | 94.8 (20253) | 95.48 | 95.28 | 95.18 | See Fig. 6 (Appendix B) |
| Lall & Bandyopadhyay (2020) | ALL/AML (72) | 86.0 (35) | 80.0 (35) | 94.3 (7129) | 95.81 | 94.38 | 98.57 | See Fig. 8 (Appendix B) |
| Golub (1999); Cilia (2019) | ALL/AML (72) | 98.6–99.4 (51) | 86.0 (51) | 94.3 (7129) | 95.71 | 93.05 | 97.14 | Despite newer FS methods, Golub (1999) performance remains competitive. |
| Zanella et al. (2022) | GSE4115 (187) | 67.9 (5) | 57.9 (5) | 68.9 (22215) | 62.11 | 66.27 | 67.31 | See Fig. 20 (Appendix B) |

**Legend.** Published FS: results reported in the original study using feature selection. Random FS: random subsets with matched feature counts. All features: random forest trained on the full feature set. Lasso, Elastic Net, RF-Imp.: standard embedded or importance-based feature selection baselines with matched feature counts.

even in scRNA-Seq with $> 96\%$ zeros, indicating the effect is NOT driven by trivially selecting nonzero columns. See Table **??**.

We observe similar patterns in three of the five benchmark datasets from the NIPS 2003 Feature Selection Challenge (Guyon et al., 2004). Consistent with the cross-dataset results, we found that models trained on extremely small subsets of randomly selected features often outperformed those trained on the full feature set. In several cases, using just 0.02% - 0.1% of the available features led to improved accuracy and AUC, with performance saturating well before reaching the full dimensionality. Table 1 summarises the results across all 30 datasets, showing that for 28 datasets this pattern holds consistently across diverse data types, including microarray, single-cell RNA-seq, bulk RNA-Seq and benchmarks datasets from NIPS FS challenge.

The only three cases where full feature set performs better than chance are in the categorization of the journalistic text, the prediction of drug activity, and synthetic data. The results for these datasets (Dexter, Dorothea, and Madelon) are shown in Figures 29, 30, and 28.

### 4.3. Comparison with published studies on FS and standard FS methods

Table 2 provides a comparison of RF results for selected datasets and related published studies. The first column lists the published study followed by the name of the dataset and sample count. The columns list accuracy with (A) selected features from the published study; (B) randomly selected features of the same size as the published study; (C) all features (D-F) features selected using standard FS methods. All results are obtained using random forests trained on the corresponding feature subset. We note that published FS

methods do not consistently outperform standard embedded FS baselines, and are often matched or exceeded by them.

## 5. Conclusion and Future Work

Our findings challenge the conventional, intuitive assumption that more features, and cleverly selected features lead to better classification performance in high-dimensional settings. Across 27 out of 30 diverse datasets, we observe that small, randomly selected feature subsets—sometimes comprising as little as 0.02% of all features—can match or even outperform models trained on the full feature set or on FS sets. This result holds across both cross-dataset and intra-dataset validation and spans multiple data types.

These results reinforce earlier insights on feature redundancy in high-dimensional data and resonate with prior work showing that Random Forests are robust to noise and overfitting due to their ensemble nature and internal feature sampling mechanisms (Breiman, 2001; Díaz-Uriarte & Alvarez de Andrés, 2006). Interestingly, we observe similar results with other non-tree, non-ensemble models. Our findings further demonstrate that even explicit external subsampling of features—performed entirely at random—can yield stable and often superior classification performance. This robustness is reinforced by our observation that the standard deviation in performance decreases consistently with increasing random subset size, suggesting the presence of many equally informative, often non-overlapping, feature combinations.

This phenomenon may have conceptual ties to random subspace methods (Ho, 1998) and the theory of random projections, particularly the Johnson–Lindenstrauss lemma (Johnson & Lindenstrauss, 1984), which shows that high-dimensional data can be embedded in lower dimensions

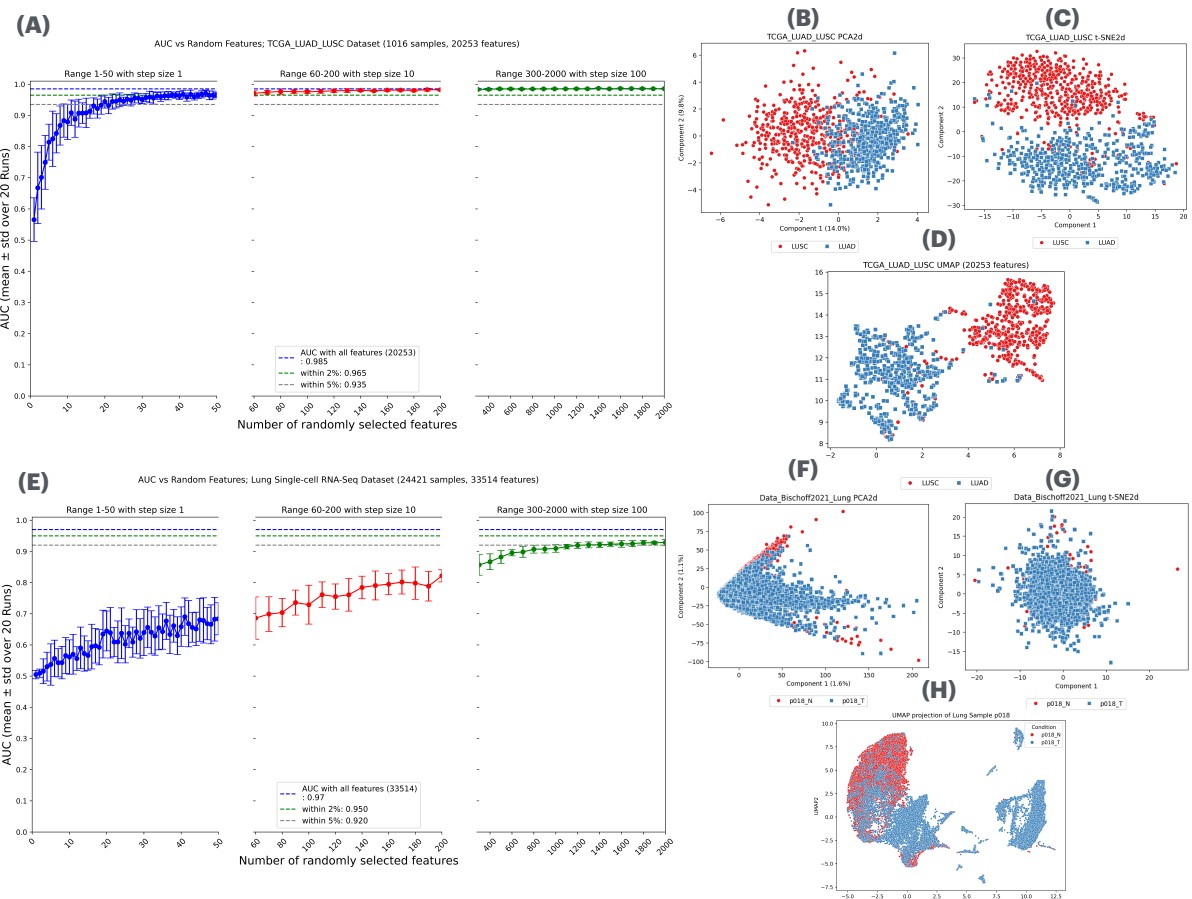

*Figure 3.* RF results on **bulk and single-cell RNA-Seq** datasets. (A) On the TCGA-LUAD-LUSC bulk RNA-Seq dataset (80:20 split), a random subset of only 50 features(∼0.3% of the total) achieves AUC comparable to the full feature set. (B–D) PCA, t-SNE and UMAP visualizations illustrating class separation in the bulk dataset. (E) On the lung cancer single-cell RNA-Seq dataset (80:20 split), random subsets of 1200 features (∼3.6%) achieve AUC within 5% of the full-feature set. (F–H) PCA, t-SNE, and UMAP visualizations showing class separation in the single-cell dataset.

while preserving pairwise distances. While our method does not perform explicit projections, the empirical success of randomly selected subspaces suggests that useful structure can be retained without sophisticated transformations. Unlike methods such as Principal Component Analysis (PCA), which apply global, often opaque transformations, random feature selection is both conceptually and computationally simple: the only surprising thing is that it works so well!

Our results also raise critical questions regarding the interpretation of feature importance in gene expression datasets – especially because such results sometimes guide downstream medical research. Our findings suggest that computationally derived feature importance may reflect statistical signals more than underlying biological causality. We do not argue against the search for biologically meaningful genes; on the contrary, we emphasize that identifying causal or mechanistically relevant genes requires biological validation. Features identified as important by computational

models should ideally be corroborated through independent experimental methods, such as perturbation assays or wet-lab validation.

For future work, investigating the diversity and overlap among high-performing random subsets may be interesting and reveal deeper insights into the intrinsic dimensionality, redundancy, and structure of high-dimensional biological data. Incorporating random subspace strategies into ensemble learning or active learning pipelines could enhance both performance and generalization, especially in domains like genomics where data is high-dimensional and sample sizes are often limited. Additionally, our MSRSS metric appears to have some interesting properties (e.g., it remains remarkably stable across totally different models - the reasons for this are unclear). It will be interesting to investigate this further and possibly quantify its effect on feature selection.

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

# A. Datasets source

*Table 3.* Summary of datasets used in this study. Citations correspond to the original dataset sources.

| No. | Name | (Samples, Features, Classes) | Datatype | Domain (Citation) |
|---|---|---|---|---|
| 1 | Colon (Alon) | (62, 2000, 2) | microarray | colon cancer (Alon et al., 1999) |
| 2 | ALL/AML | (72, 7129, 2) | microarray | leukemia (Golub et al., 1999) |
| 3 | GSE6008 | (98, 22283, 4) | microarray | ovarian cancer (Hendrix et al., 2006) |
| 4 | GSE18842 | (90, 54675, 2) | microarray | lung cancer (Sánchez-Palencia et al., 2011) |
| 5 | GSE42743 | (103, 54675, 2) | microarray | oral cavity cancer (Lohavanichbutr et al., 2013) |
| 6 | GSE19804 | (114, 54675, 2) | microarray | lung cancer (Lu et al., 2010) |
| 7 | GSE6919_U95B | (124, 12620, 2) | microarray | prostate cancer (Chandran et al., 2007) |
| 8 | GSE3365 | (127, 22814, 3) | microarray | bowel disease (Burczynski et al., 2006) |
| 9 | GSE50161 | (130, 54575, 5) | microarray | brain tumours (Griesinger et al., 2013) |
| 10 | GSE22820 | (139, 33579, 2) | microarray | breast cancer (Liu et al., 2011) |
| 11 | GSE53757 | (143, 54675, 2) | microarray | kidney cancer (Roemeling et al., 2014) |
| 12 | GSE30219 | (146, 54675, 2) | microarray | lung cancer (Rousseaux et al., 2013) |
| 13 | GSE21510 | (147, 54675, 3) | microarray | colorectal cancer (Tsukamoto et al., 2011) |
| 14 | GSE45827 | (151, 54675, 6) | microarray | breast cancer (Gruosso et al., 2016) |
| 15 | GSE76427 | (165, 47322, 2) | microarray | liver cancer (Grinchuk et al., 2018) |
| 16 | GSE4115 | (187, 22215, 2) | microarray | lung epithelial cancer (Spira et al., 2007) |
| 17 | GSE44076 | (194, 49386, 2) | microarray | colorectal cancer (Solé et al., 2014) |
| 18 | GSE11223 | (202, 40991, 3) | microarray | colon inflammation (Noble et al., 2008) |
| 19 | GSE28497 | (281, 22284, 7) | microarray | pediatric leukemia (Coustan-Smith et al., 2011) |
| 20 | GSE70947 | (289, 35981, 2) | microarray | breast cancer (Quigley & Kristensen, 2016) |
| 21 | GSE14250 | (357, 22277, 2) | microarray | hepatocellular carcinoma (Roessler et al., 2010) |
| 22 | TCGA (LUAD/LUSC) | (1016, 20253, 2) | bulk RNA-Seq | lung cancer (The Cancer Genome Atlas Research Network, 2014; 2012) |
| 23 | TCGA pan-cancer | (10223, 20253, 33) | bulk RNA-Seq | pan-cancer (The Cancer Genome Atlas Research Network, 2018) |
| 24 | Lung (scRNA-Seq) | (20966, 33514, 9) | scRNA-Seq | lung adenocarcinoma (Bischoff et al., 2021) |
| 25 | Lung (scRNA-Seq) | (24421, 33514, 2) | scRNA-Seq | lung adenocarcinoma (Bischoff et al., 2021) |
| 26 | Arcene† | (200, 10000, 2) | mass-spectrometry | ovarian vs prostate cancer (Guyon et al., 2004) |
| 27 | Dexter† | (600, 20000, 2) | text | corporate acquisition (Guyon et al., 2004) |
| 28 | Dorothea† | (1150, 100000, 2) | fingerprint | drug activity prediction (Guyon et al., 2004) |
| 29 | Madelon† | (2600, 500, 2) | synthetic | cluster classification (Guyon et al., 2004) |
| 30 | Gisette† | (7000, 5000, 2) | image | digit classification (4 vs 9) (Guyon et al., 2004) |

*Table 4.* Dataset statistics and sparsity metrics

| dataset | samples | features | global zero frac | median feat zero frac | median sample zero frac |
|---------|---------|----------|------------------|-----------------------|-------------------------|
| Bischoff 2021 (Lung cancer sc-RNA-Seq) | 21052 | 33514 | 96.4% | 99.9% | 96.5% |
| TCGA Pan-cancer (bulk RNA-Seq) | 10223 | 20253 | 0.0% | 0.0% | 0.0% |

# B. Results with Random Forest for all datasets

Additional figures supporting the main text are provided here (figs. 4–30). They include RF results across datasets.

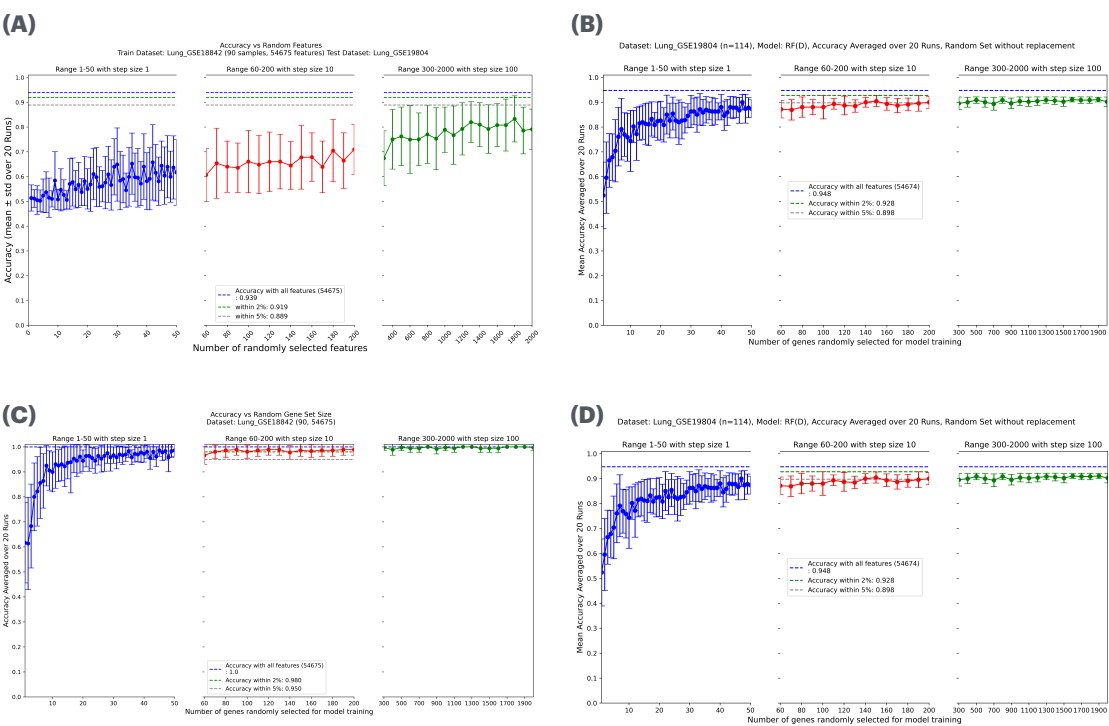

*Figure 4.* Random Forest performance with lung microarray dataset pairs (mean and standard deviation are reported over 20 runs) (A) RF models trained on GSE18842 and tested on GSE19804 show that randomly selected subsets never achieve accuracy comparable to using all features. (B) RF models trained on GSE19804 and tested on GSE18842 show that 200 randomly selected features ( 0.4% of all features) perform comparable to all features. (C) Model performance with an 80:20 train-test split using randomly selected feature subsets. For GSE18842, just 50 randomly selected features are sufficient to match the accuracy comparable to all features. (D) Similarly, for GSE19804, 200 features suffice to match accuracy with all features.

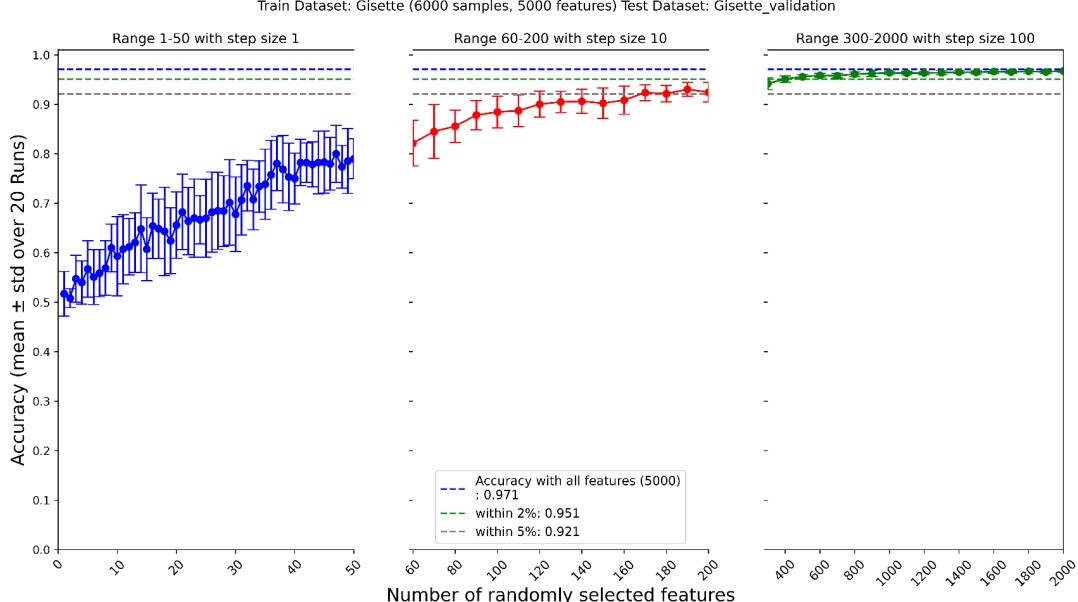

*Figure 5.* Random Forest performance with Gisette image dataset (mean and standard deviation are reported over 20 runs). The task of GISETTE is to discriminate between two confusable handwritten digits: the four and the nine. The model is trained on Gisette train dataset and tested on Gisette validation dataset shows that randomly selected subsets of just 200 achieve accuracy comparable to using all features.

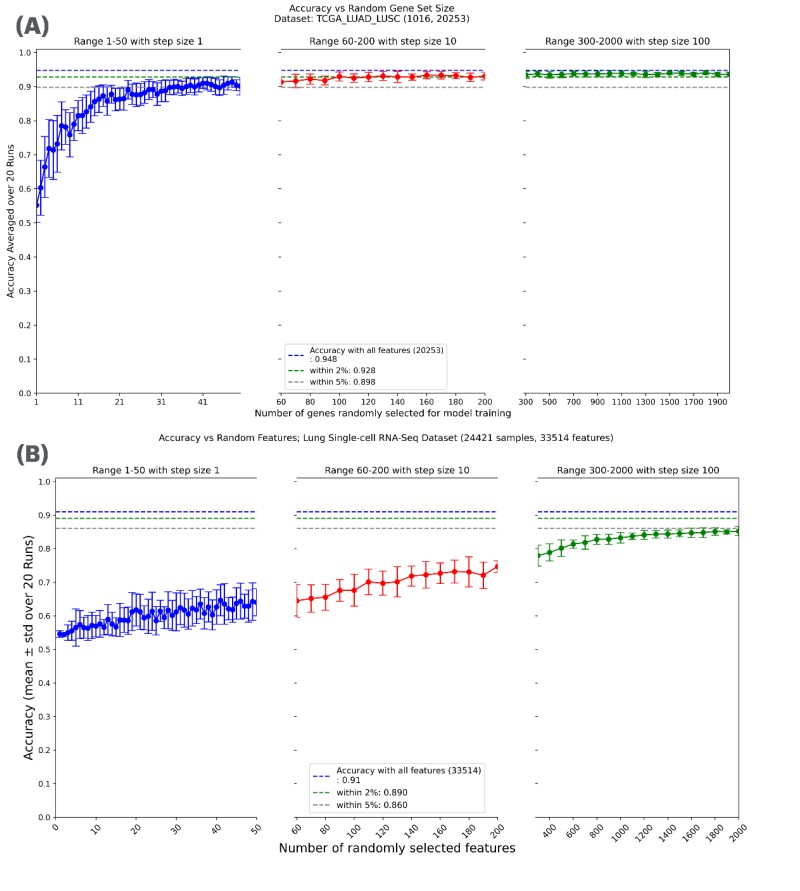

*Figure 6.* Random Forest performance with bulk RNA-Seq and Single-cell RNA-Seq datasets (mean and standard deviation are reported over 20 runs). (A) Models trained and tested on TCGA-LUAD-LUSC bulk RNA-Seq dataset (80:20 split) shows that a random subset of size 50 (<0.3%) is able to match within-5% accuracy of all features. (B) On the lung cancer single-cell RNA-Seq dataset (80:20 split), randomly selected subsets of size 2000 achieve accuracy within 5% of the full-feature model.

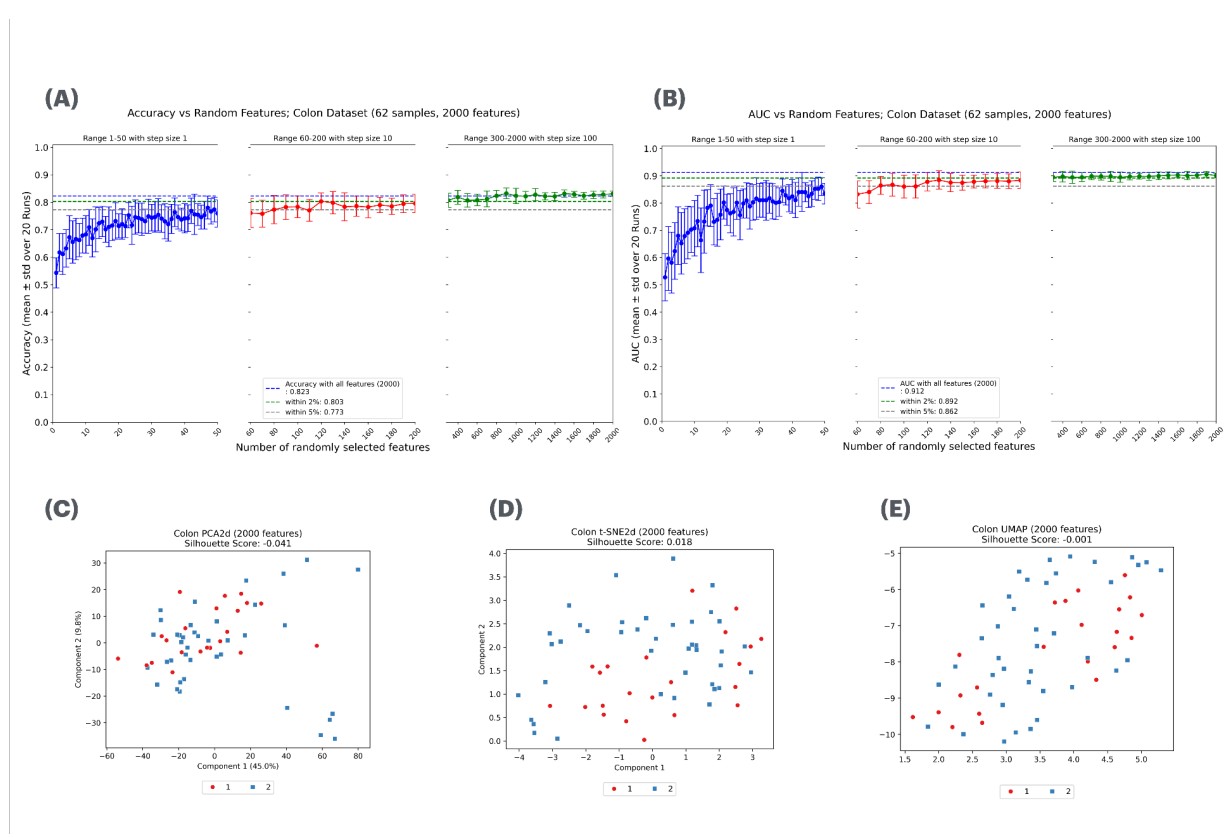

*Figure 7.* Random Forest performance with Colon microarray dataset (mean and standard deviation are reported over 20 runs). (A) & (B) models trained and tested on 80:20 split shows that a random subset of size ∼100 is able to match accuracy and AUC with all features, respectively. (C) & (D) & (E) PCA, t-SNE and UMAP plots showing class separation.

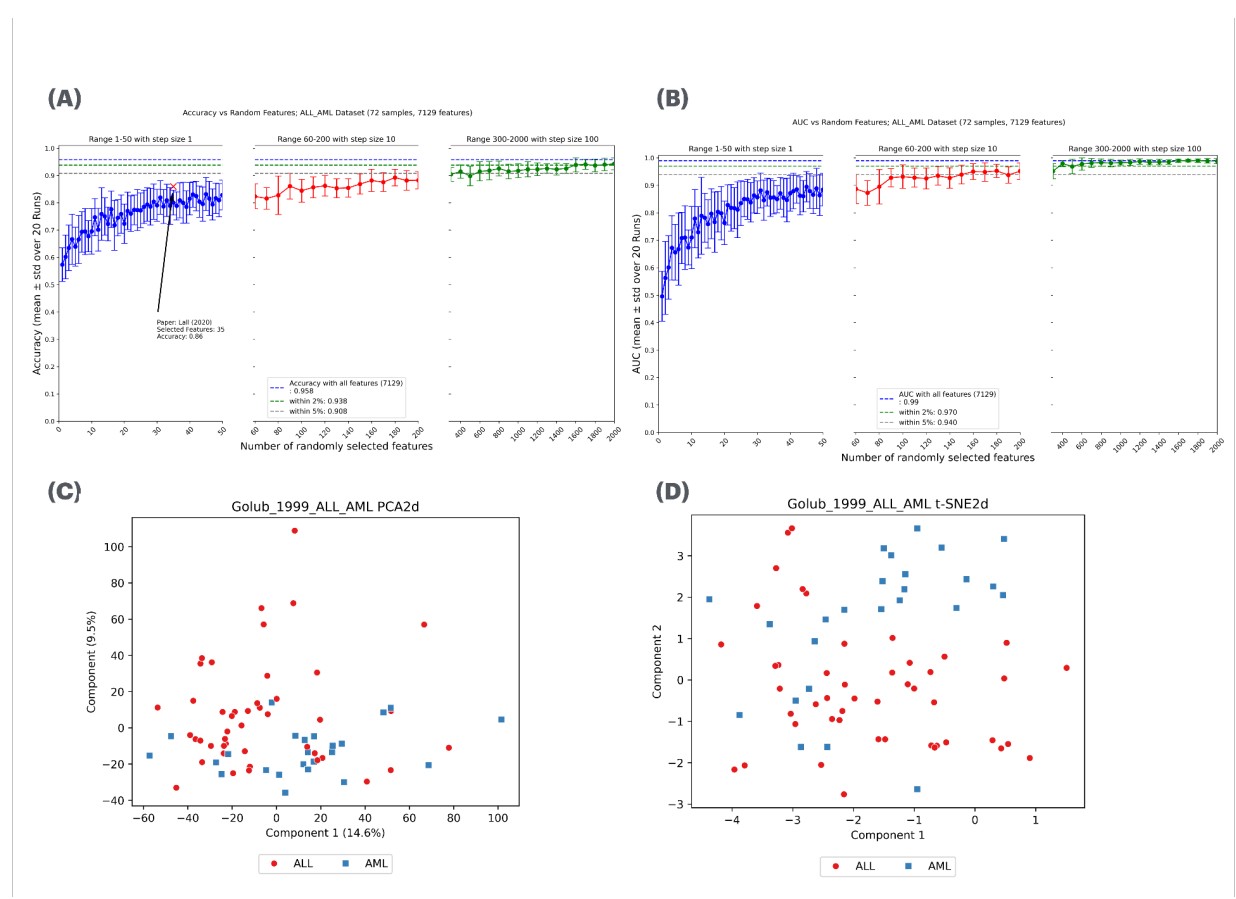

*Figure 8.* Random Forest performance with ALL/AML Leukemia microarray dataset (mean and standard deviation are reported over 20 runs). (A) & (B) models trained and tested on 80:20 split shows that a random subset of size ∼200 is able to match accuracy and AUC with all features, respectively. (C) & (D) PCA, t-SNE plots showing class separation.

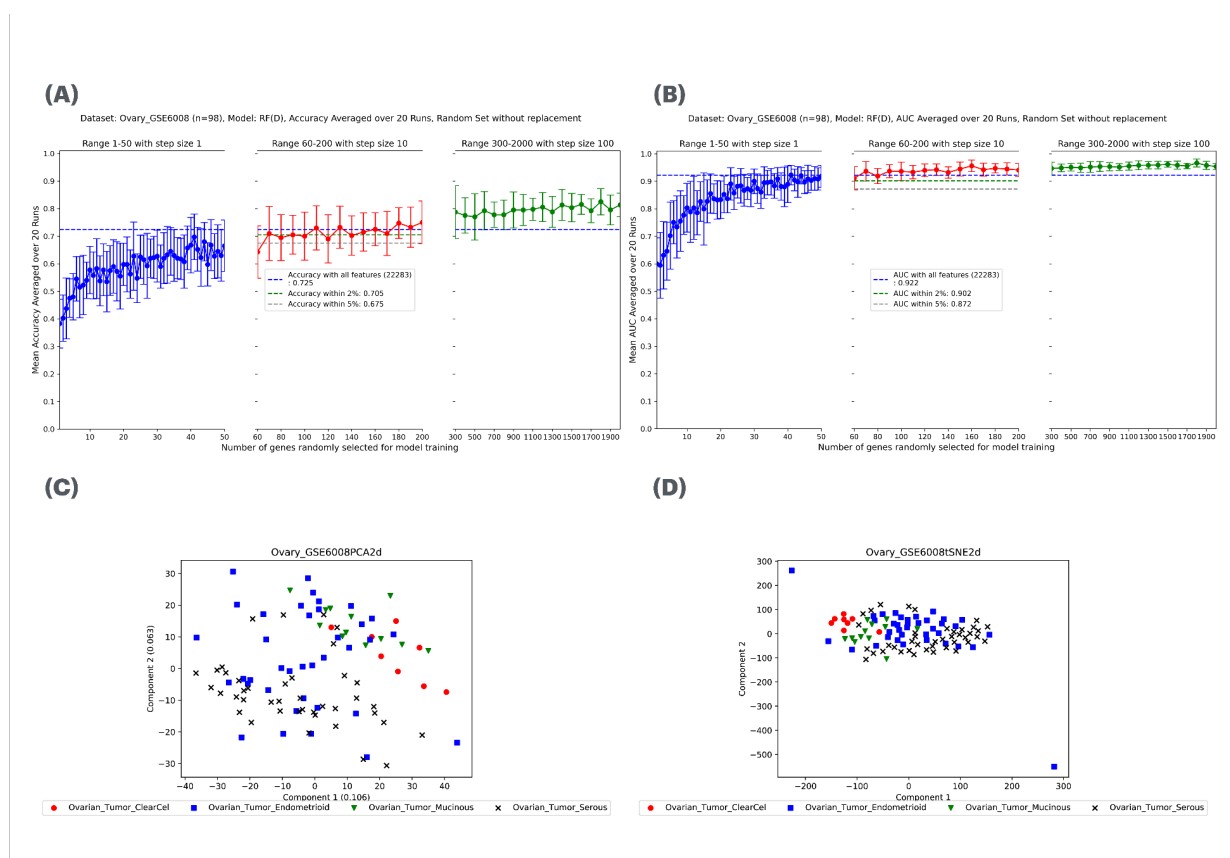

*Figure 9.* Random Forest performance with Ovary (GSE6008) microarray dataset (mean and standard deviation are reported over 20 runs). (A) & (B) models trained and tested on 80:20 split shows that a random subset is able to match accuracy and AUC with all features, respectively. (C) & (D) PCA, t-SNE plots showing class separation.

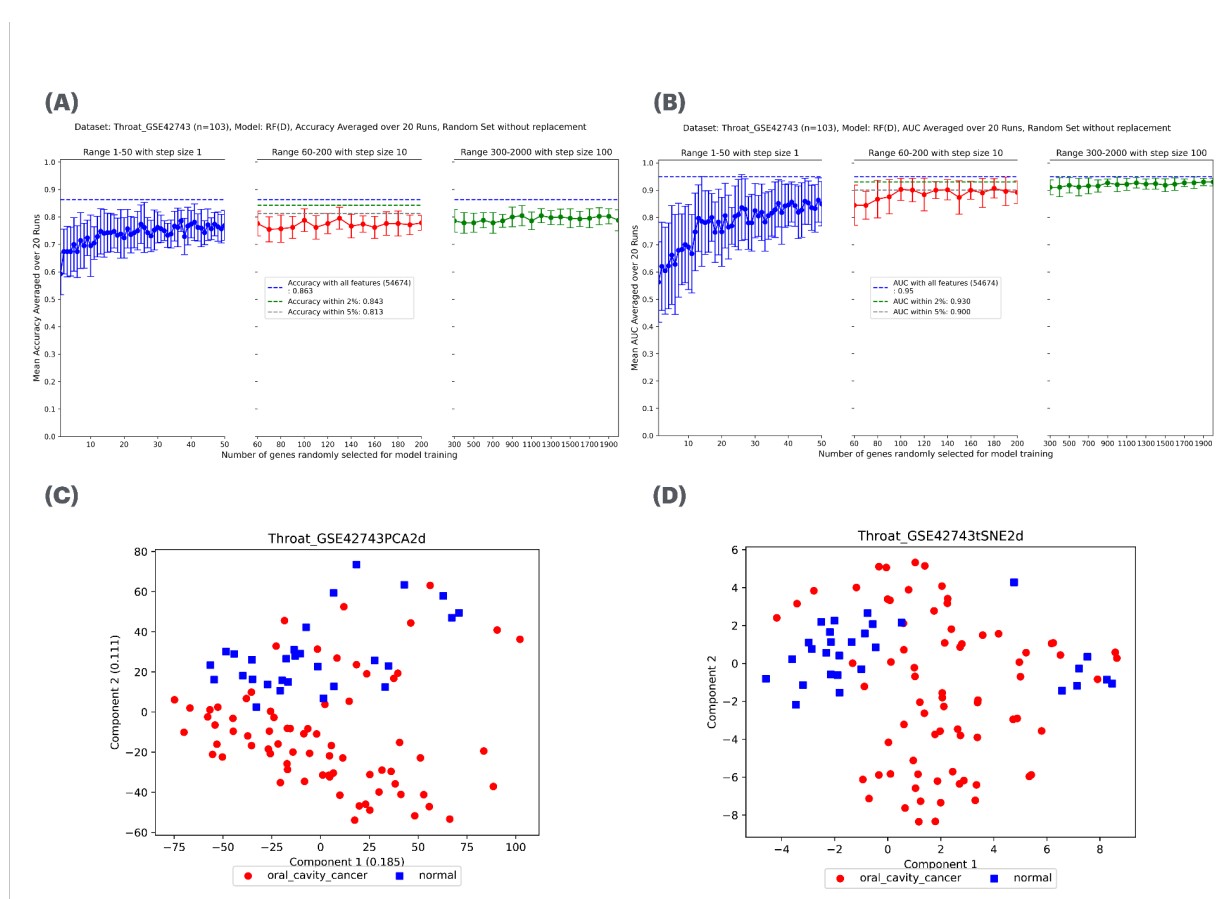

*Figure 10.* Random Forest performance with Oral/Throat (GSE42743) microarray dataset (mean and standard deviation are reported over 20 runs). (A) & (B) models trained and tested on 80:20 split shows that a random subset is able to match within-5% accuracy and AUC with all features, respectively. (C) & (D) PCA, t-SNE plots showing class separation.

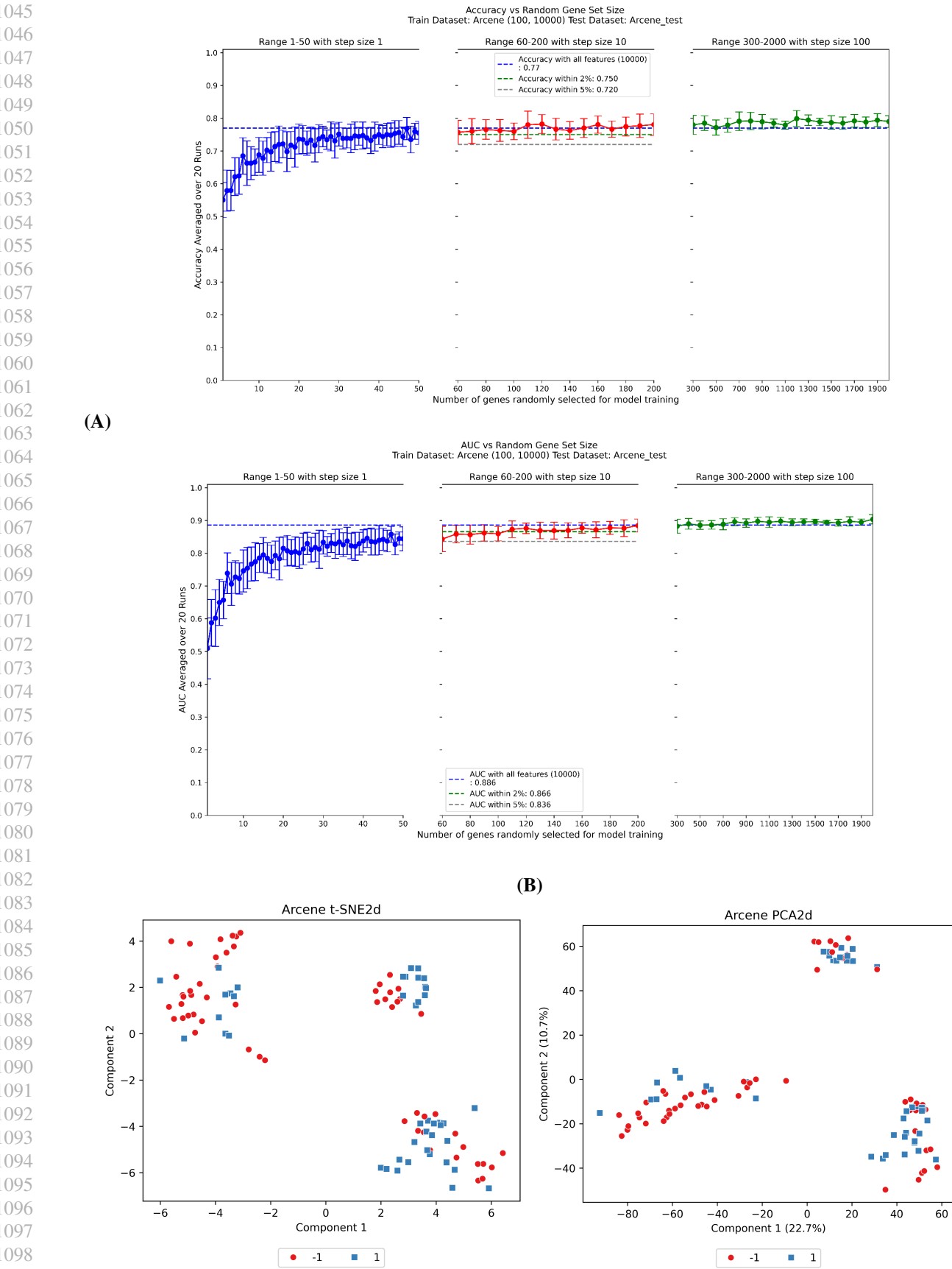

*Figure 11.* Random Forest performance with Arcene mass-spectrometry dataset (mean and standard deviation are reported over 20 runs). The task of ARCENE is to distinguish cancer versus normal patterns from mass-spectrometric data. (A) Models trained and tested on 80:20 split shows that a random subset of size ∼50 (0.5% of all features) is able to match within-5% Accuracy and AUC of all features. (B) PCA, t-SNE plots showing class separation.

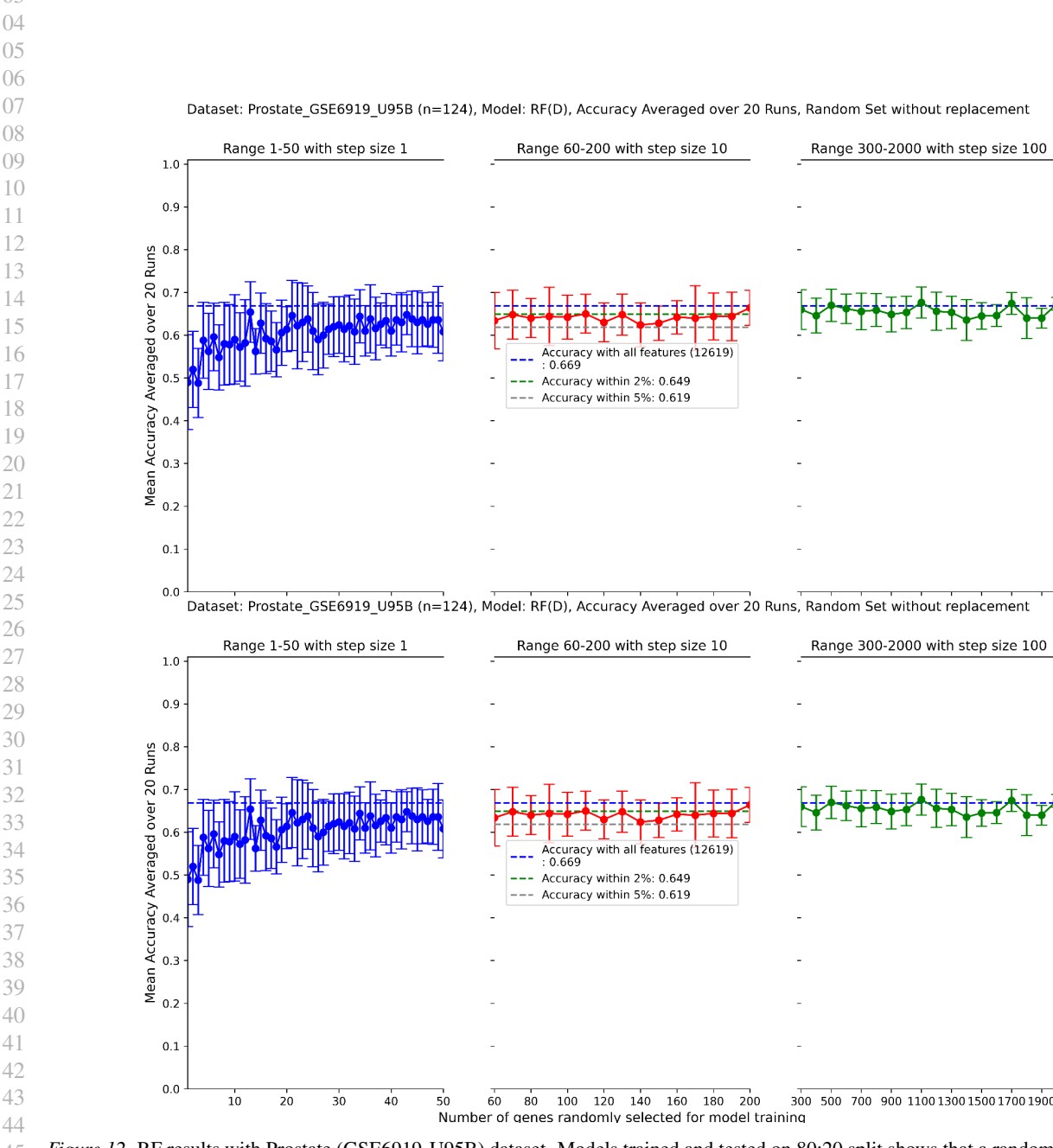

*Figure 12.* RF results with Prostate (GSE6919_U95B) dataset. Models trained and tested on 80:20 split shows that a random subset of size ∼50 (0.4% of all features) is able to match within-5% Accuracy and AUC of all features.

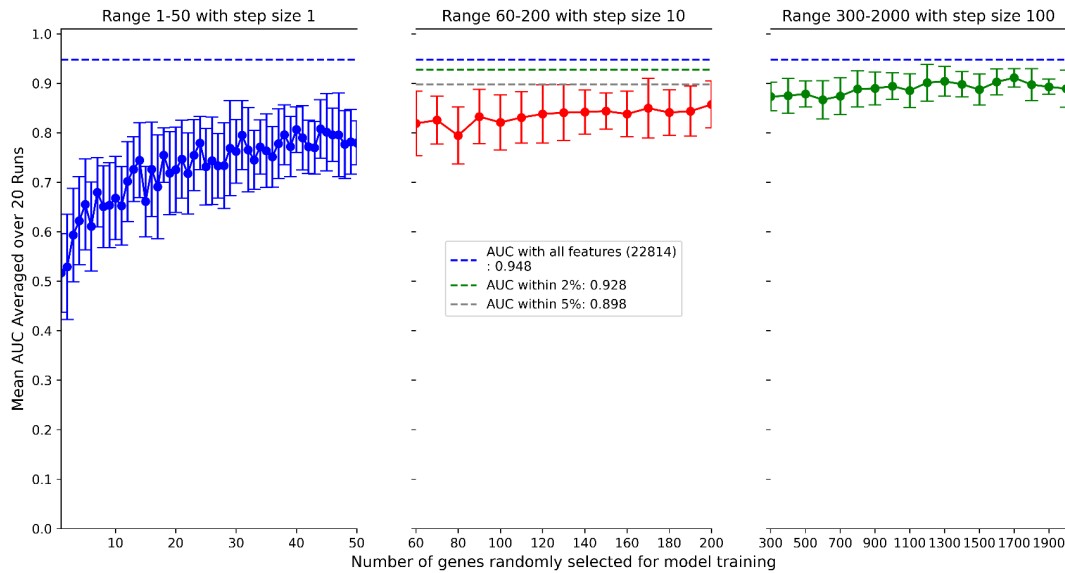

*Figure 13.* RF results with Bowel (GSE3365) dataset. Models trained and tested on 80:20 split shows that a random subset of size ∼500 (∼2.2% of all features) is able to match within-5% Accuracy and AUC of all features.

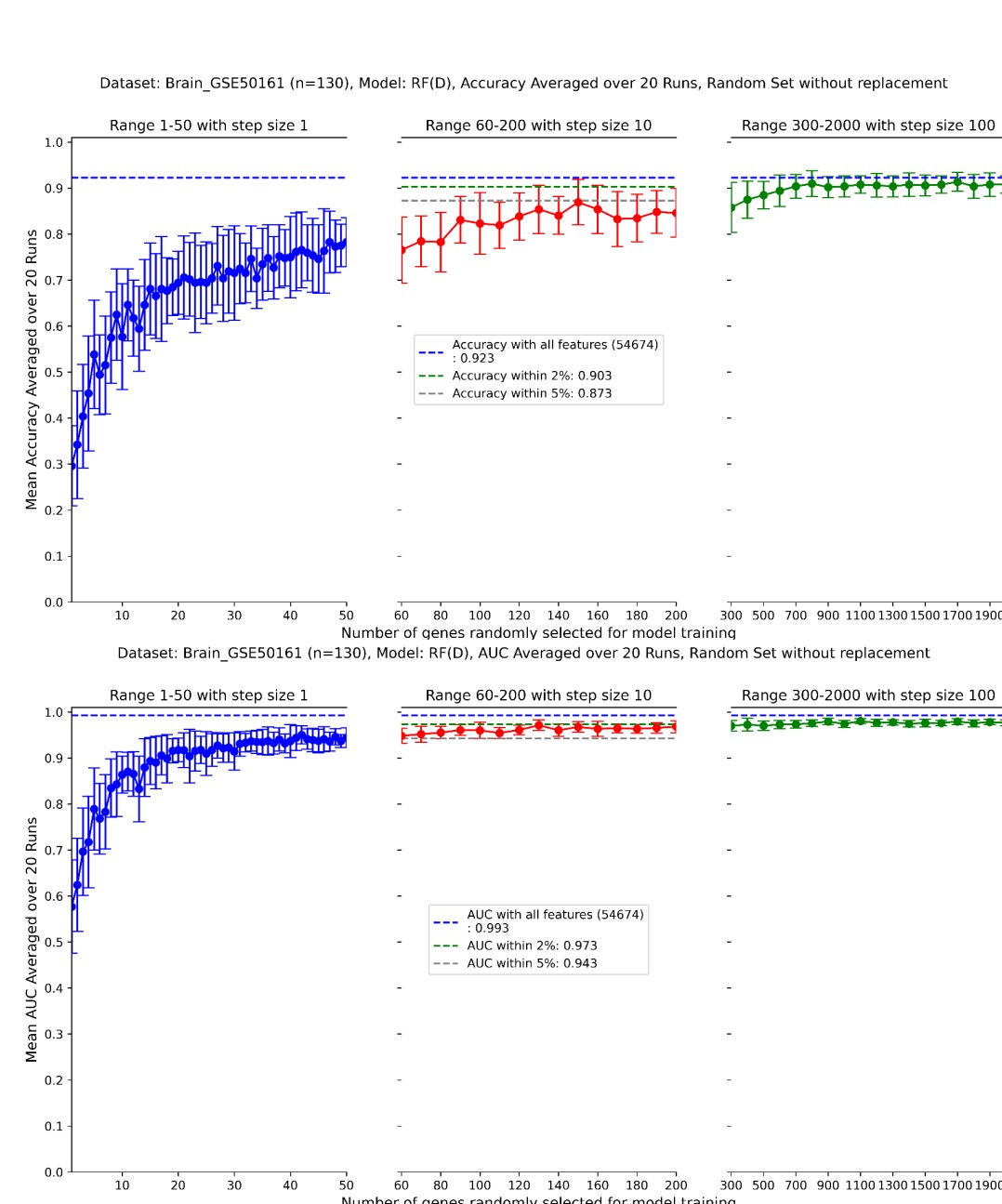

*Figure 14.* Random Forest performance with Brain (GSE50161) dataset (mean and standard deviation are reported over 20 runs). Models trained and tested on 80:20 split shows that a random subset of size ∼50 (∼0.09% of all features) is able to match within-5% Accuracy and AUC of all features.

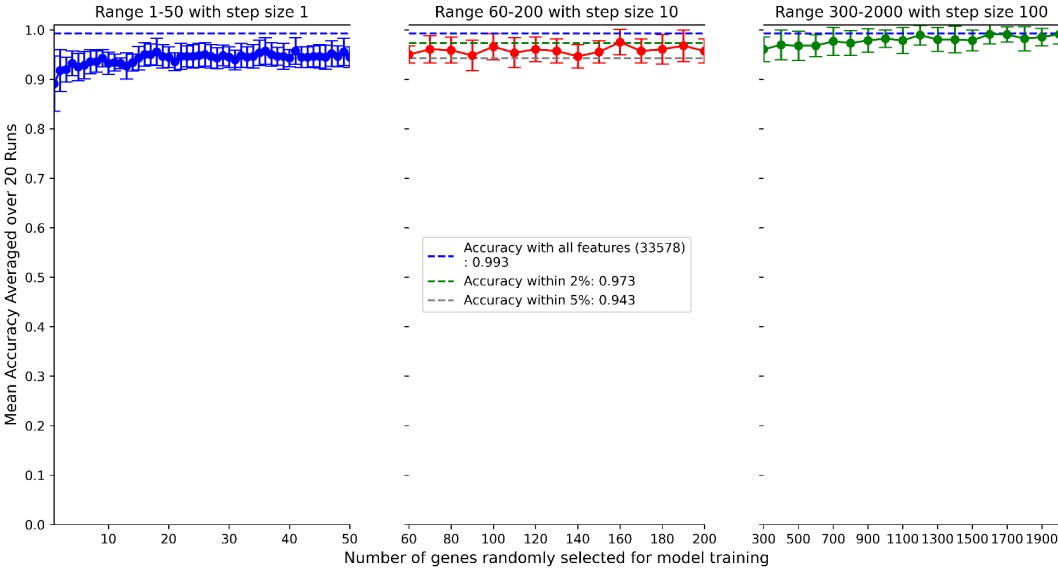

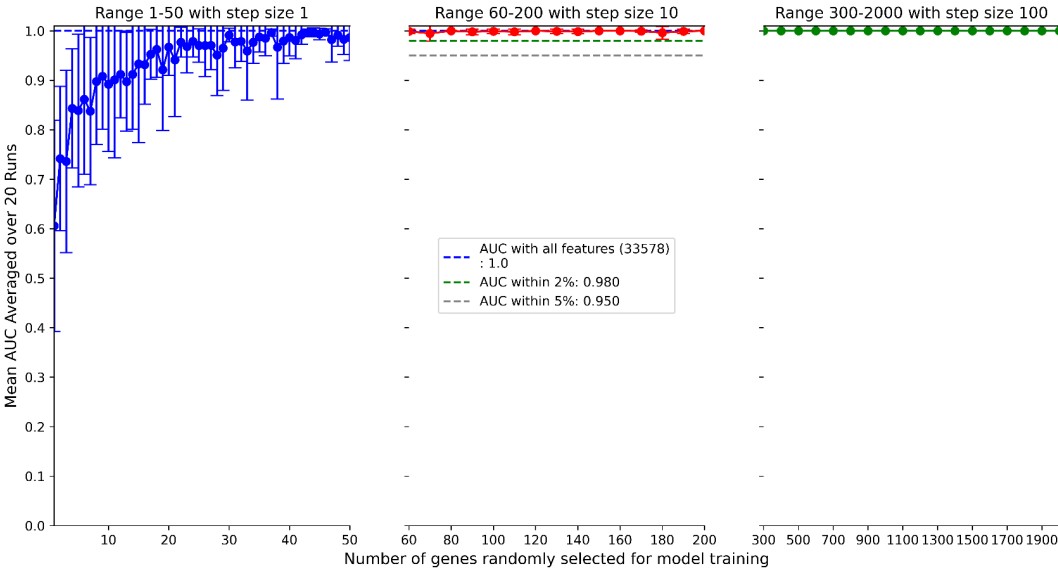

*Figure 15.* Random Forest performance with Breast (GSE22820) dataset (mean and standard deviation are reported over 20 runs). Models trained and tested on 80:20 split shows that a random subset of size ∼50 (∼0.14% of all features) is able to match within-5% Accuracy and full AUC of all features. (The unusually high accuracy with just one feature is because there is a severe class imbalance in this dataset).

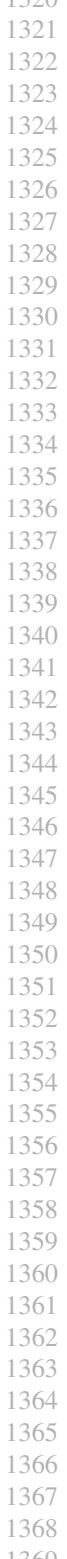

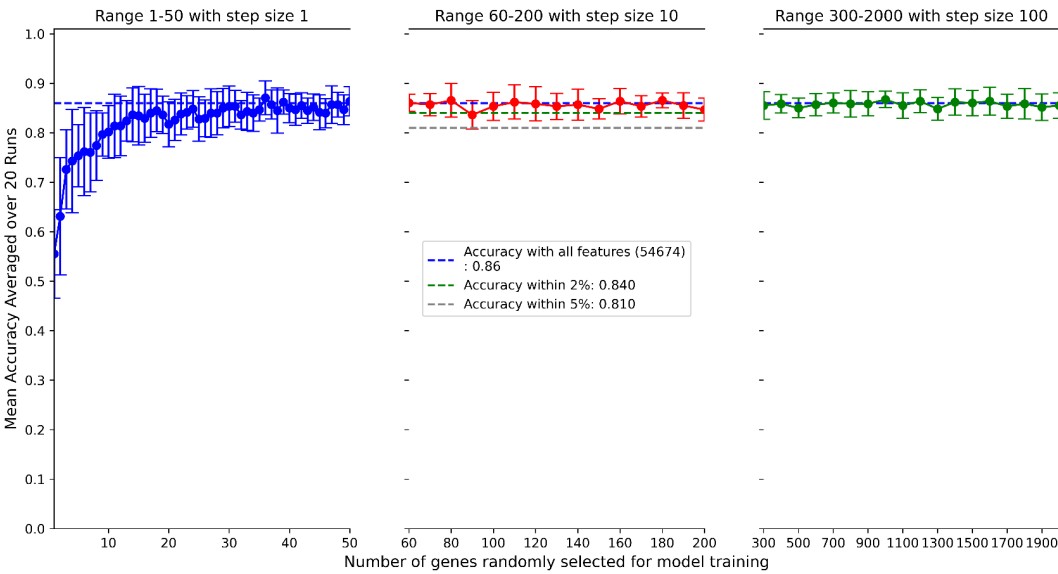

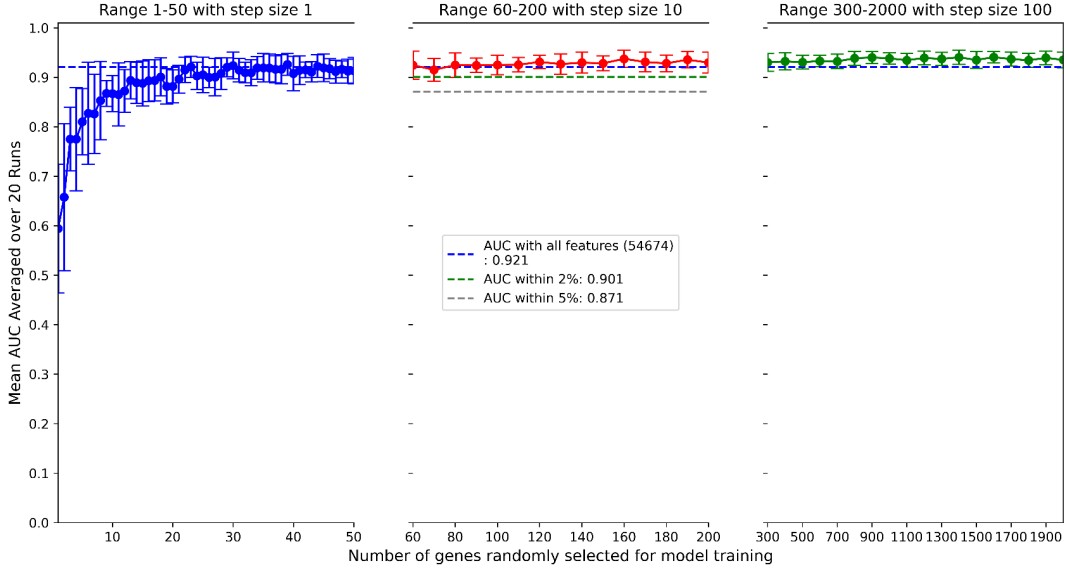

*Figure 16.* Random Forest performance with Renal (GSE53757) dataset (mean and standard deviation are reported over 20 runs). Models trained and tested on 80:20 split shows that a random subset of size ~30 (~0.06% of all features) is able to match full Accuracy and full AUC of all features.

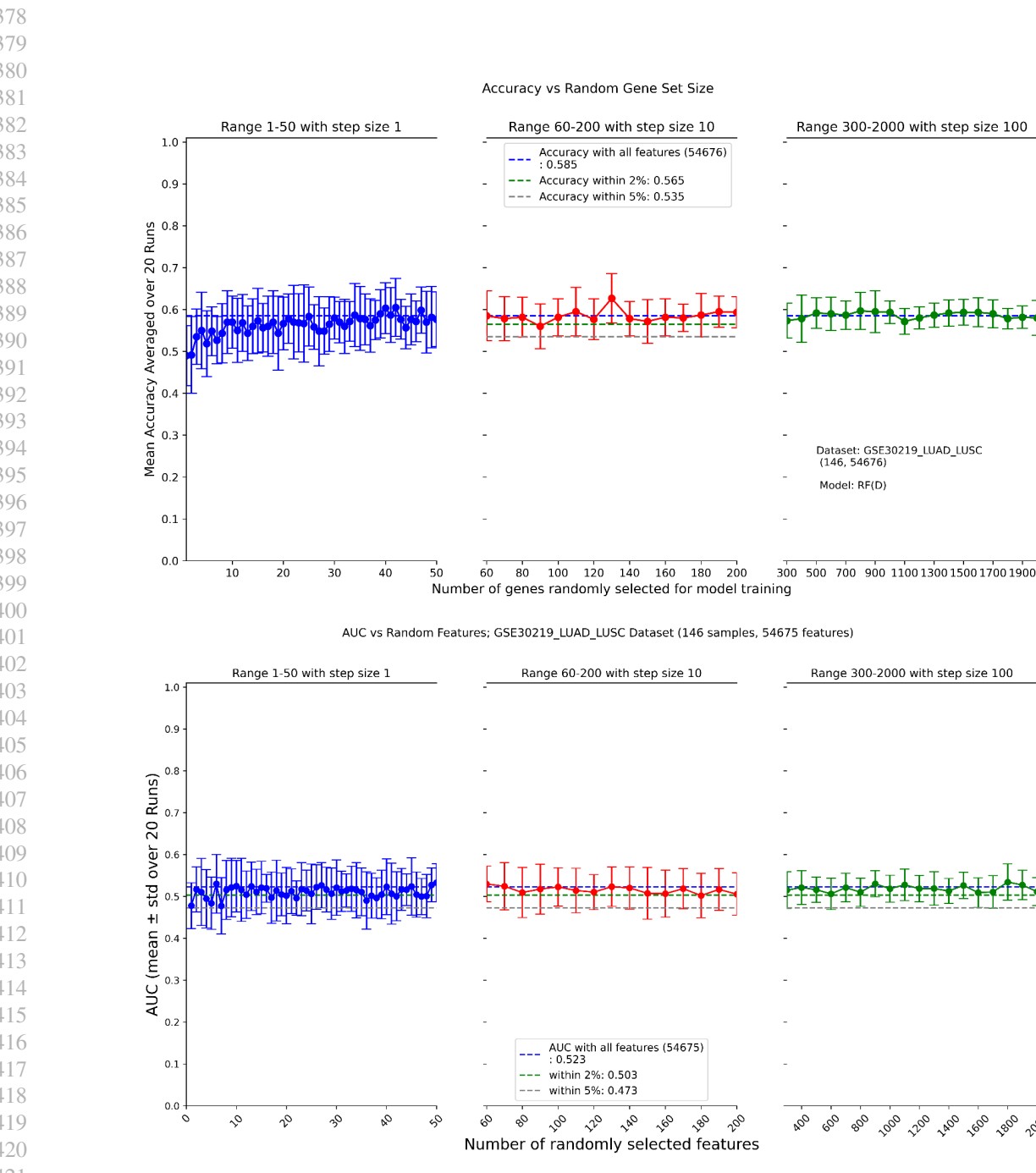

*Figure 17.* Random Forest performance with Lung Cancer (GSE30219) dataset (mean and standard deviation are reported over 20 runs). Models trained and tested on 80:20 split shows that a random subset is able to match full Accuracy and full AUC of all features.

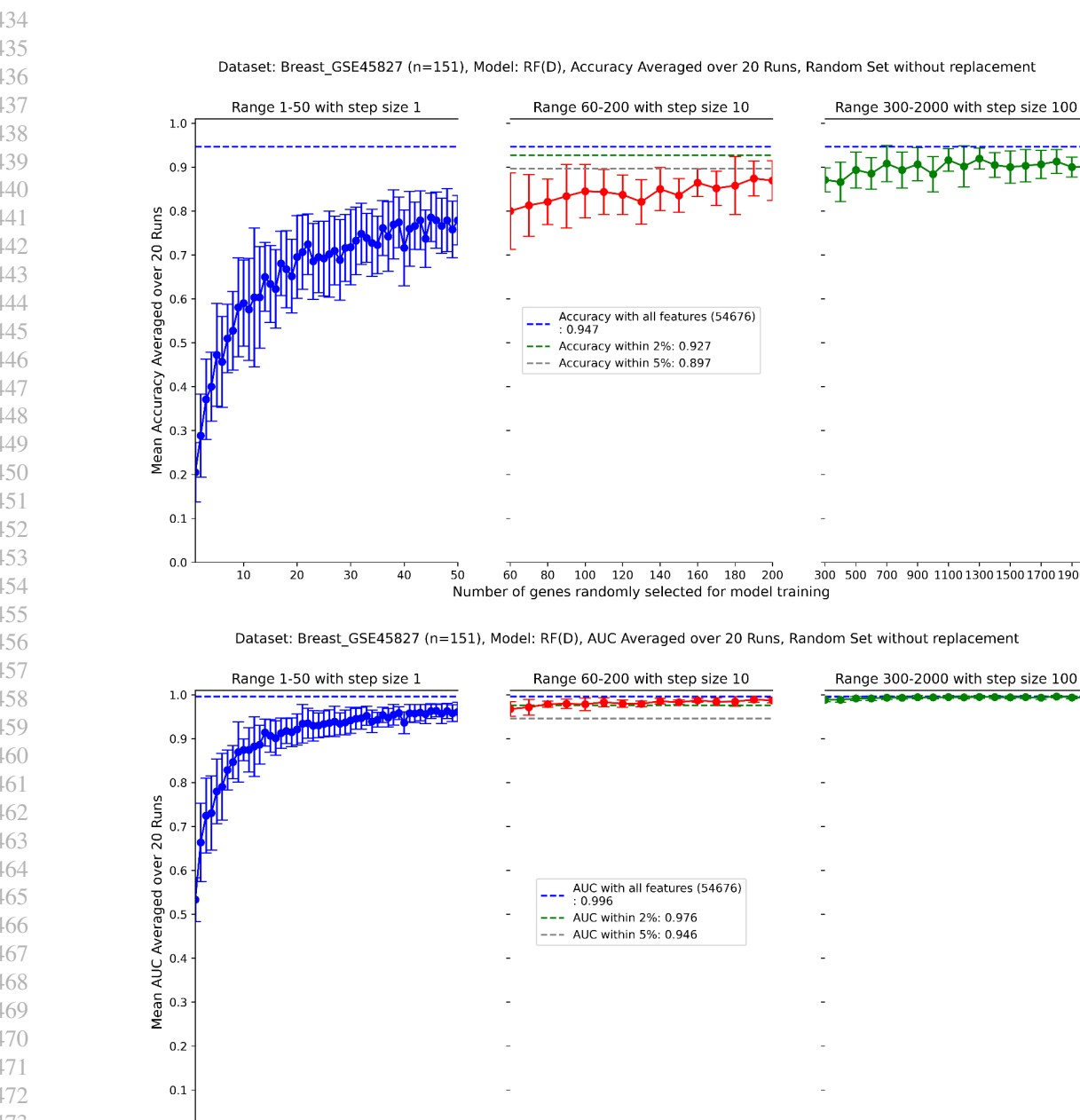

*Figure 18.* Random Forest performance with Breast Cancer (GSE45827) dataset (mean and standard deviation are reported over 20 runs). Models trained and tested on 80:20 split shows that a random subset is able to match within-5% Accuracy and full AUC of all features.

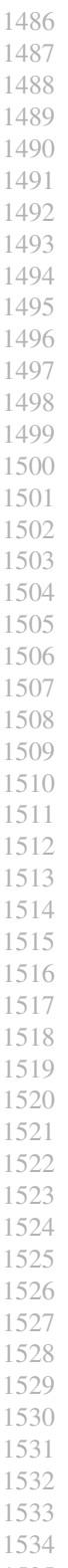

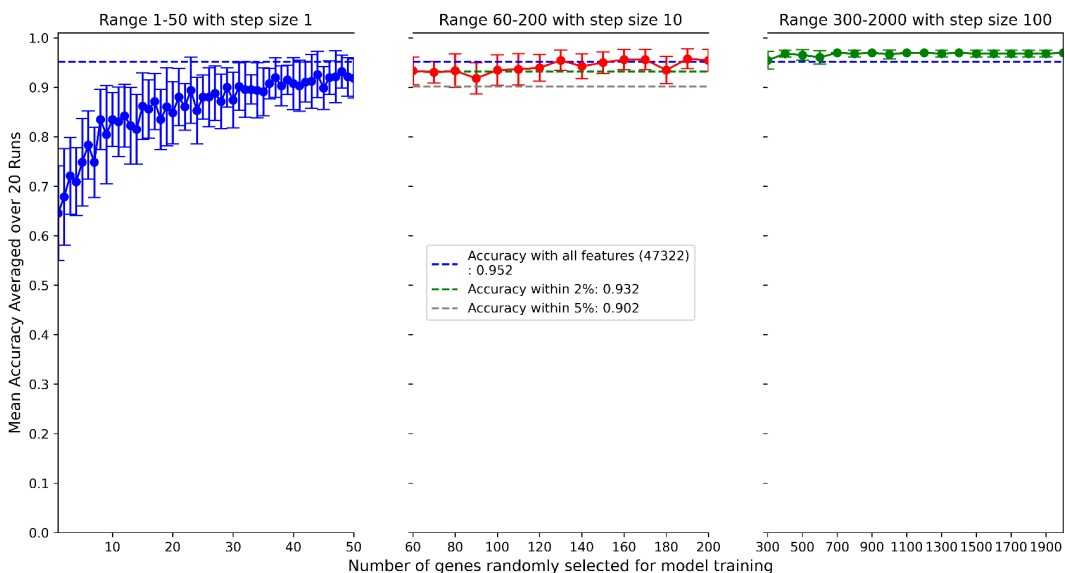

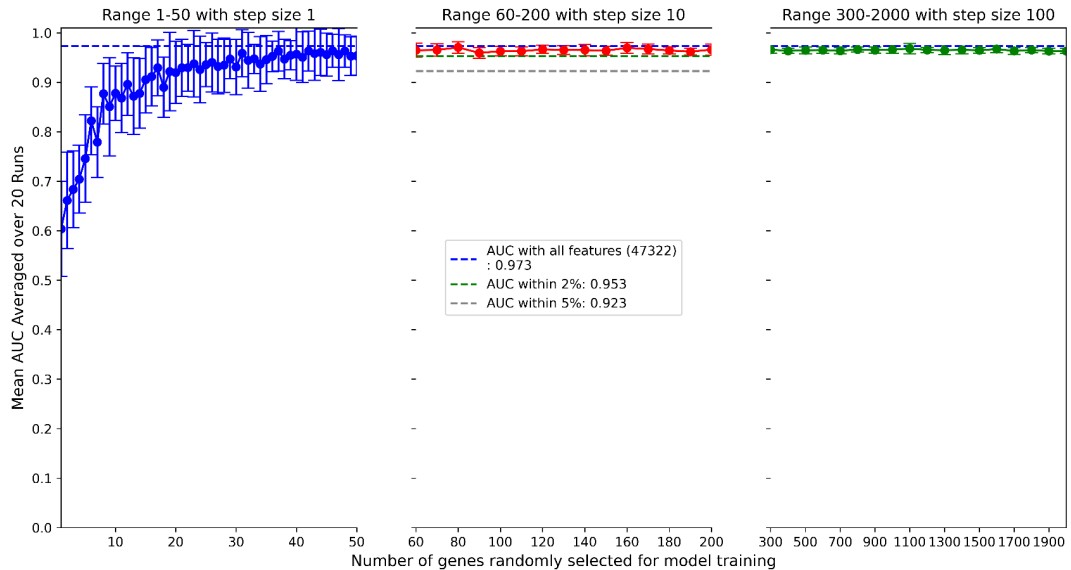

*Figure 19.* Random Forest performance with Liver Cancer (GSE76427) dataset (mean and standard deviation are reported over 20 runs). Models trained and tested on 80:20 split shows that a random subset is able to match full Accuracy and full AUC of all features.

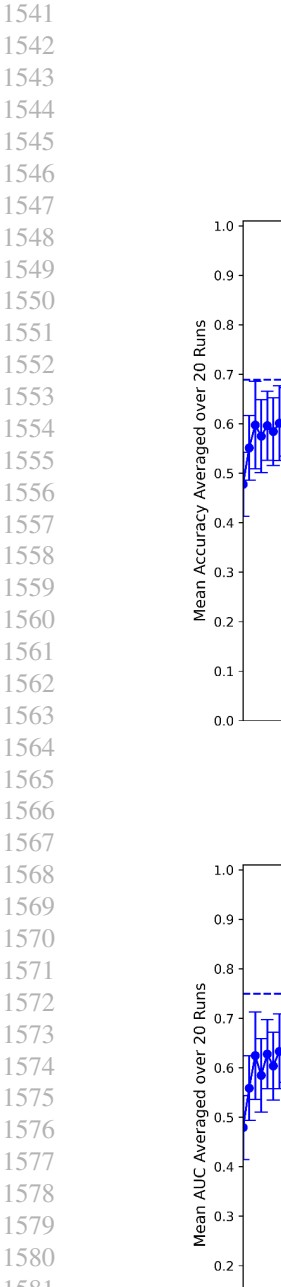
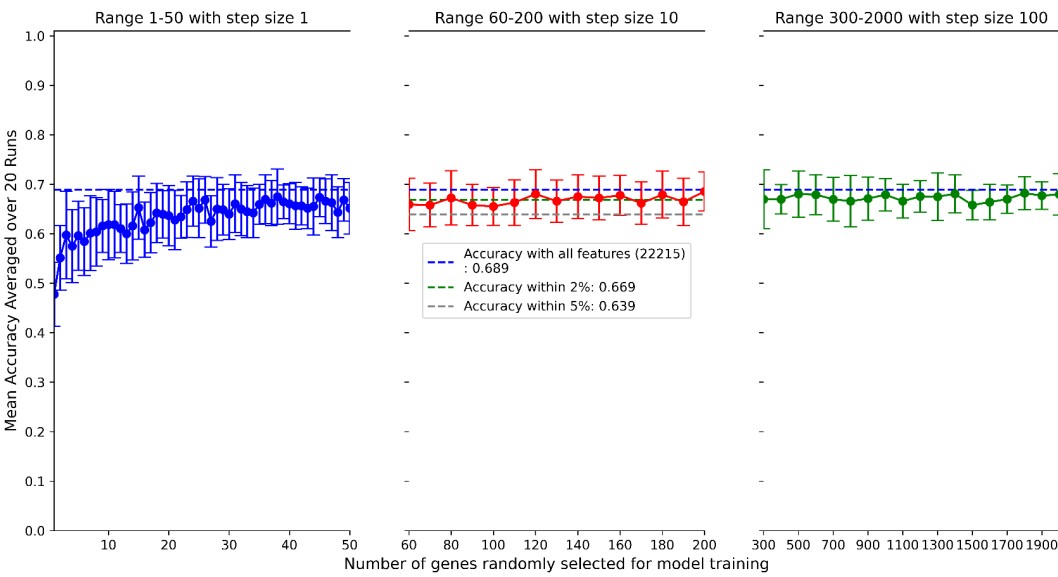
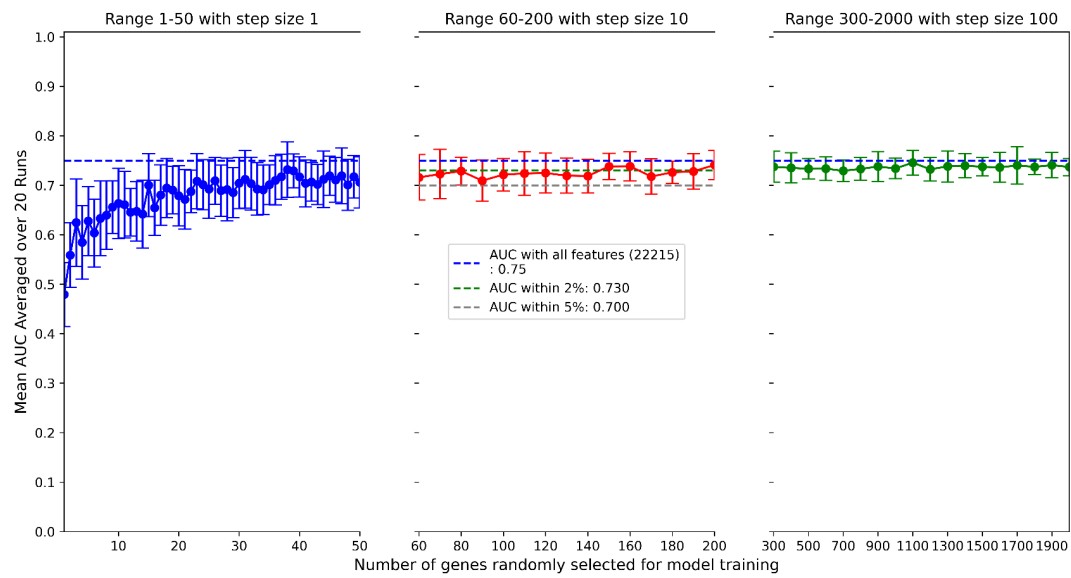

*Figure 20.* Random Forest performance with Lung Cancer (GSE4115) dataset (mean and standard deviation are reported over 20 runs). Models trained and tested on 80:20 split shows that a random subset is able to match within-2% Accuracy and AUC of all features.

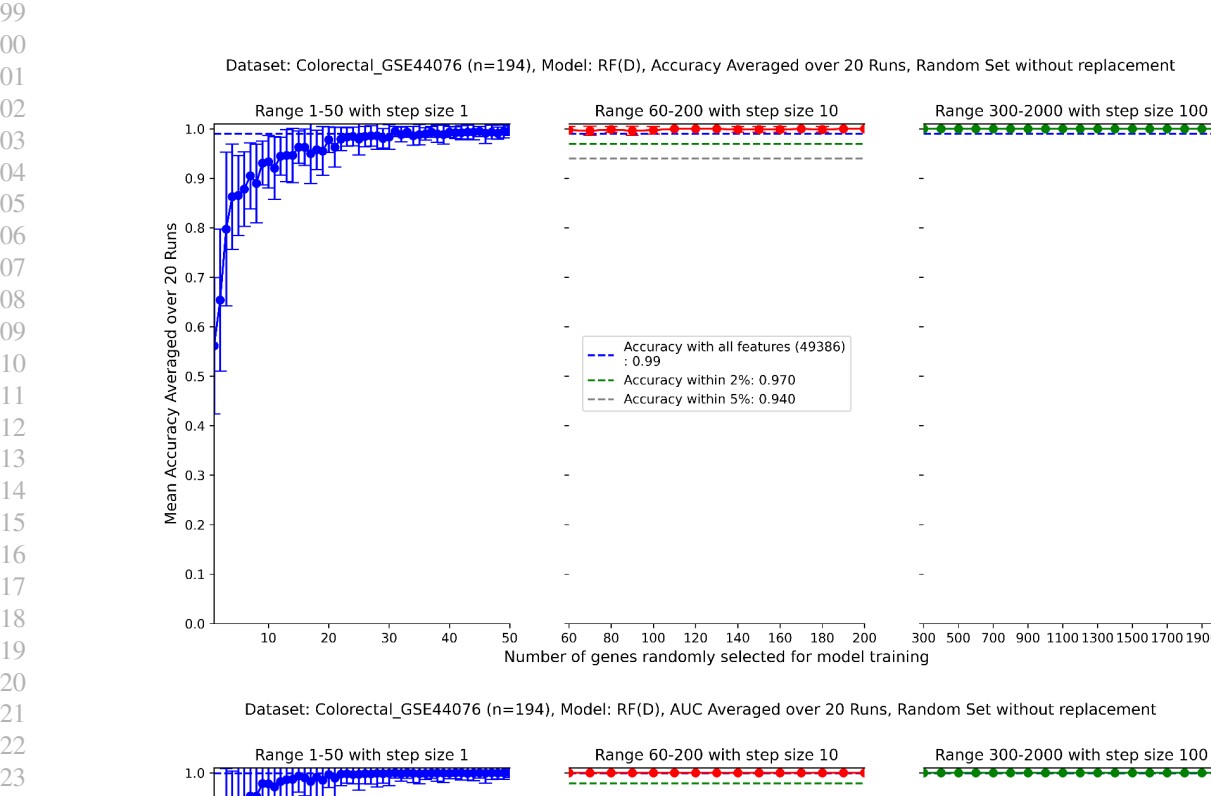

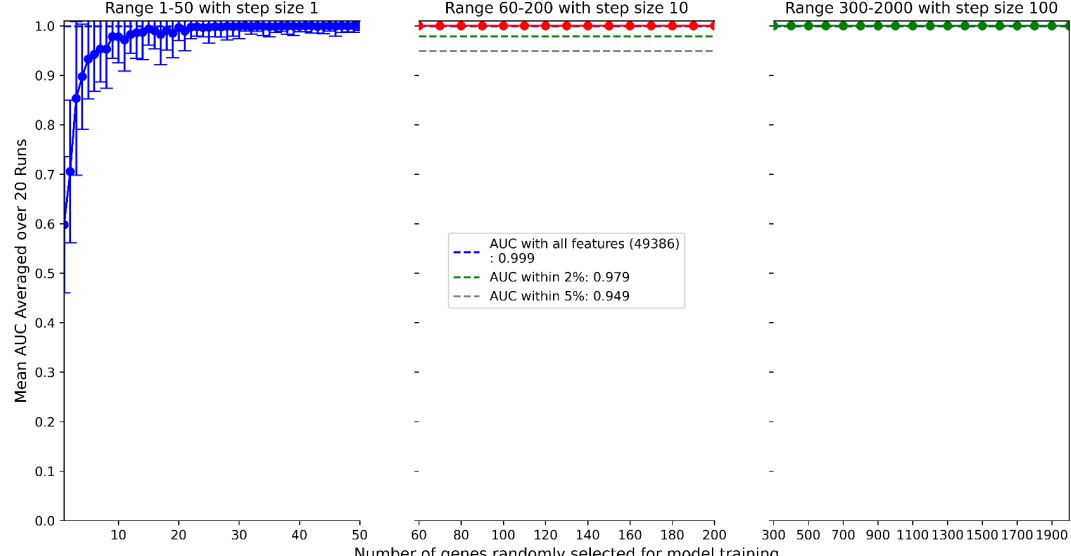

*Figure 21.* Random Forest performance with Colorectal Cancer (GSE44076) dataset (mean and standard deviation are reported over 20 runs). Models trained and tested on 80:20 split shows that a random subset is able to match full Accuracy and full AUC of all features.

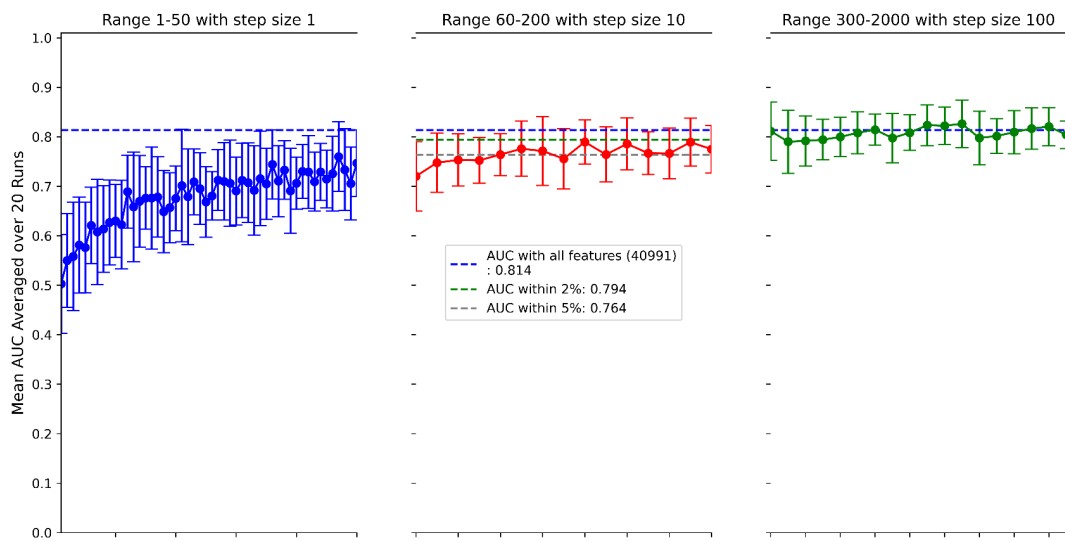

*Figure 22.* Random Forest performance with Colon Cancer (GSE11223) dataset (mean and standard deviation are reported over 20 runs). Models trained and tested on 80:20 split shows that a random subset is able to match full AUC of all features.

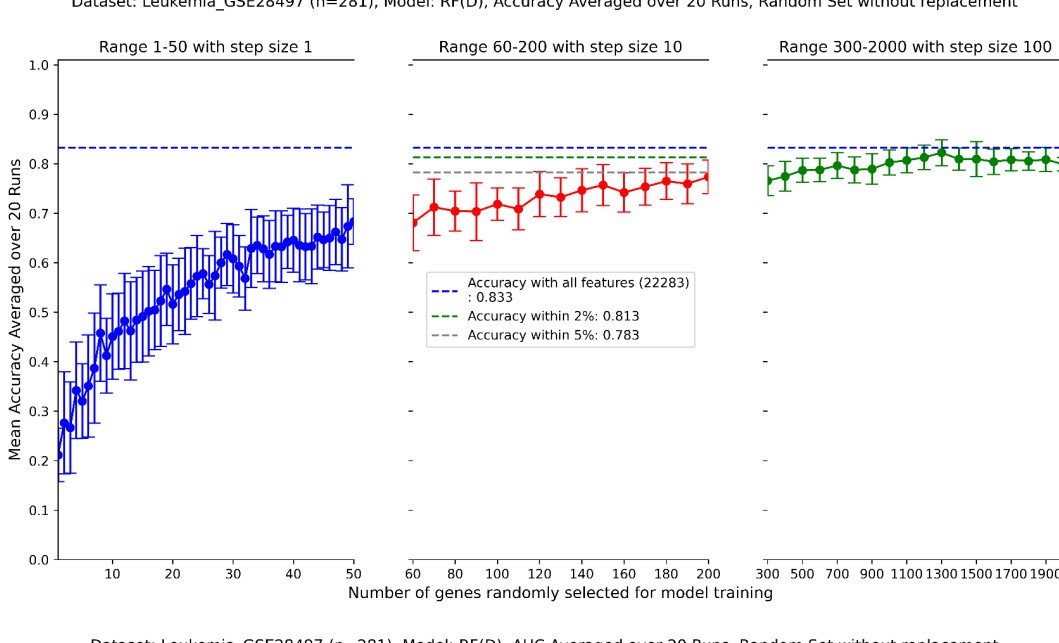

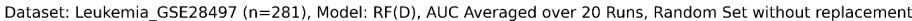

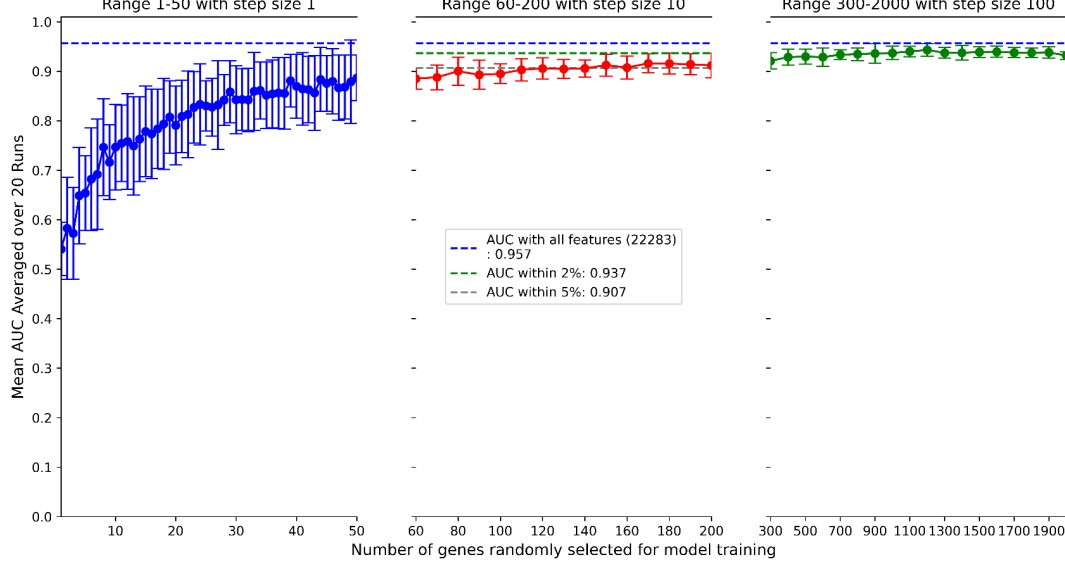

*Figure 23.* Random Forest performance with Leukemia (GSE28497) dataset (mean and standard deviation are reported over 20 runs). Models trained and tested on 80:20 split shows that a random subset is able to match within-2% Accuracy and AUC of all features.

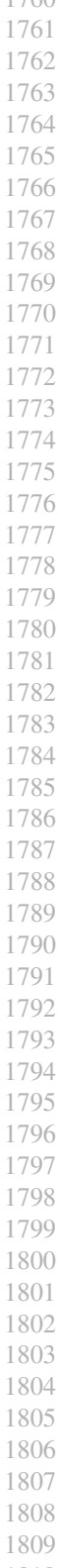

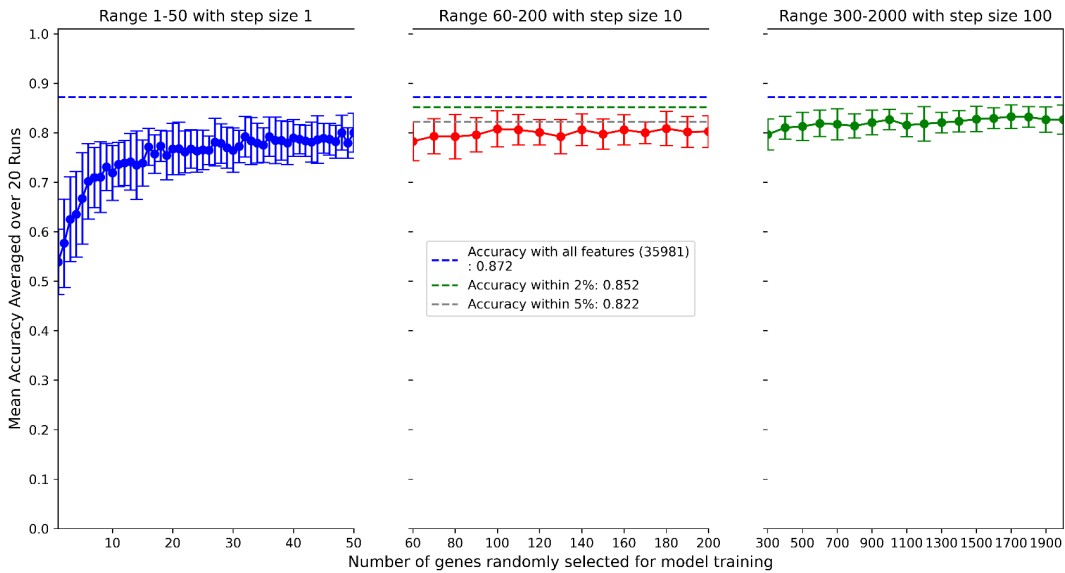

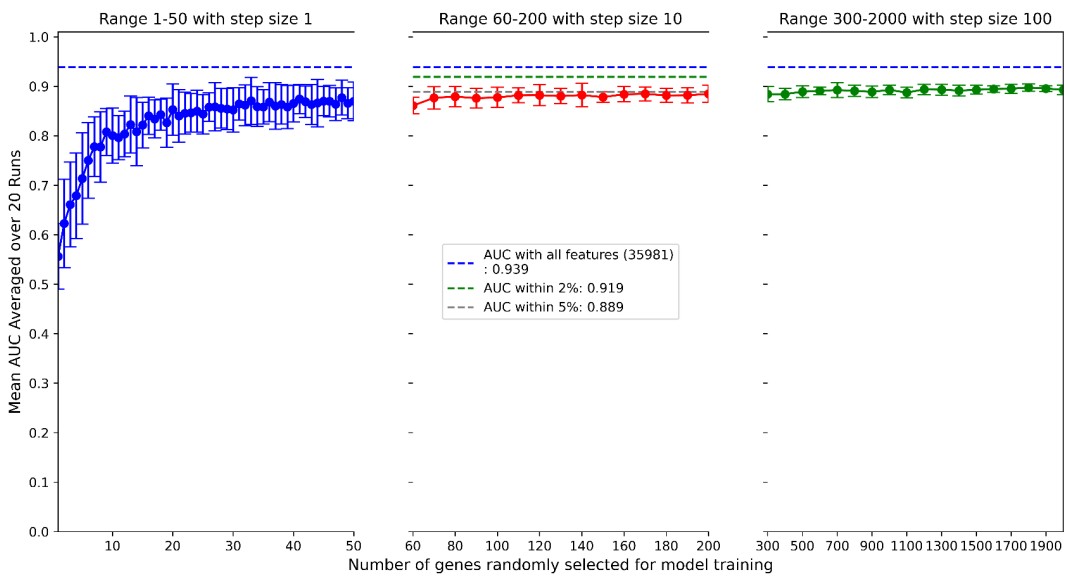

*Figure 24.* Random Forest performance with Breast Cancer (GSE70947) dataset (mean and standard deviation are reported over 20 runs). Models trained and tested on 80:20 split shows that a random subset is able to match within-5% Accuracy and AUC of all features.

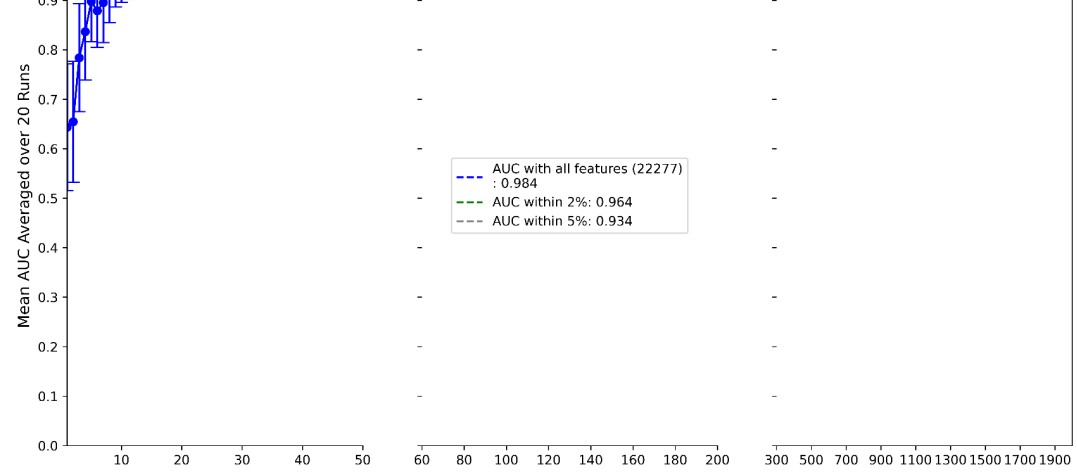

*Figure 25.* Random Forest performance with Liver Cancer (GSE14520) dataset (mean and standard deviation are reported over 20 runs). Models trained and tested on 80:20 split shows that a small random subset is able to match full Accuracy and AUC of all features.

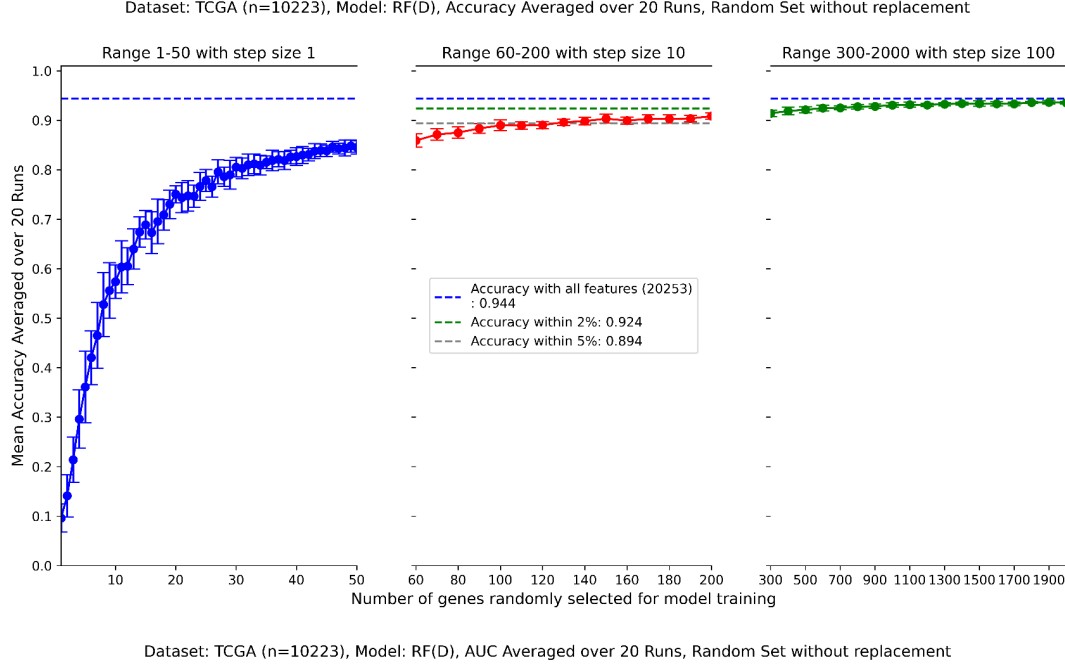

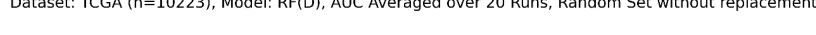

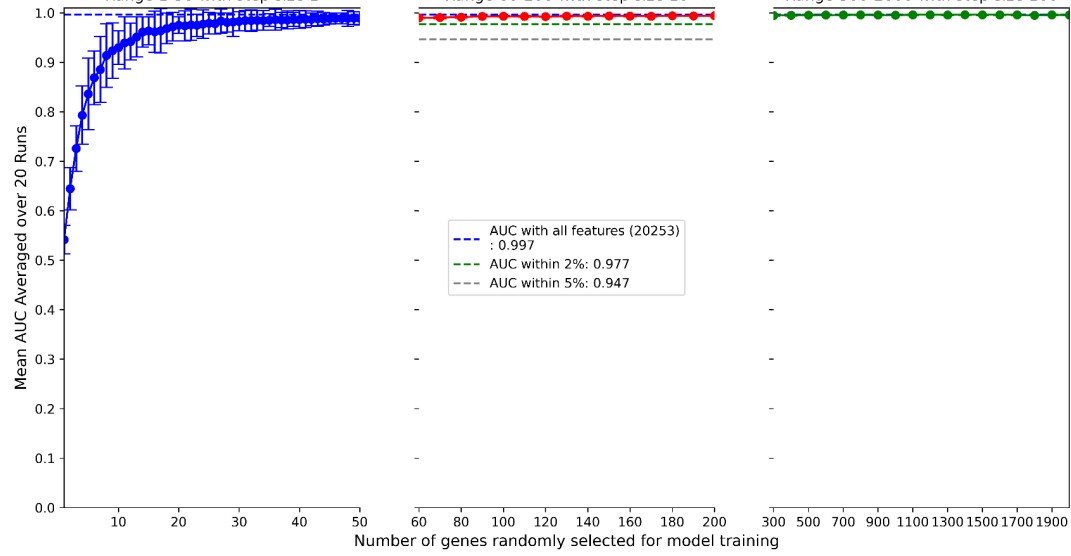

*Figure 26.* Random Forest performance with bulk RNA-Seq TCGA dataset with 33 classes (mean and standard deviation are reported over 20 runs). Models trained and tested on 80:20 split shows that a small random subset is able to match full Accuracy and AUC of all features.

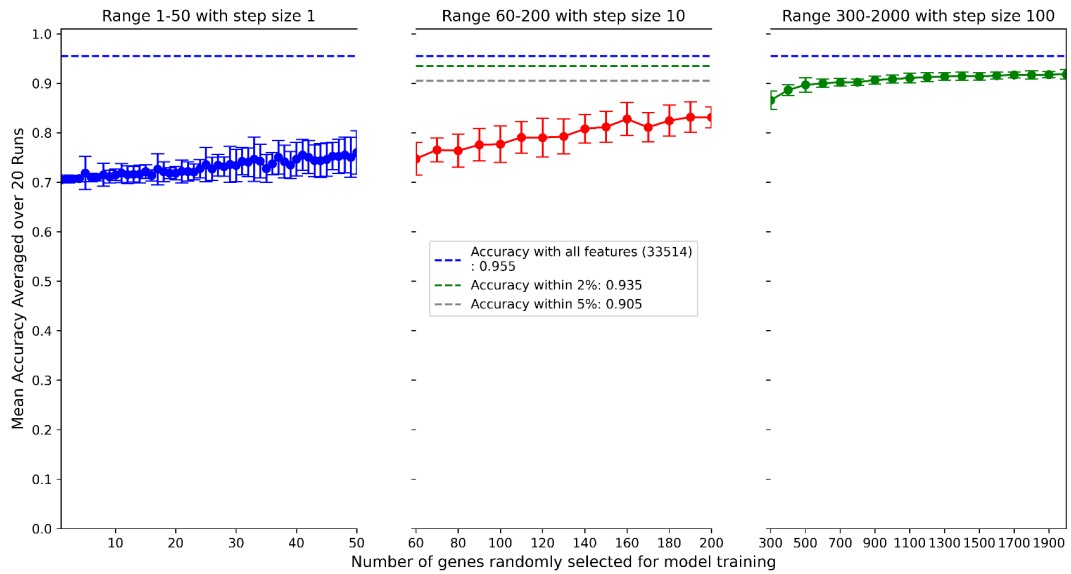

*Figure 27.* Random Forest performance with single-cell RNA-Seq Lung dataset with 9 classes (mean and standard deviation are reported over 20 runs). Models trained and tested on 80:20 split shows that a small random subset is able to match within-5% Accuracy and AUC of all features.

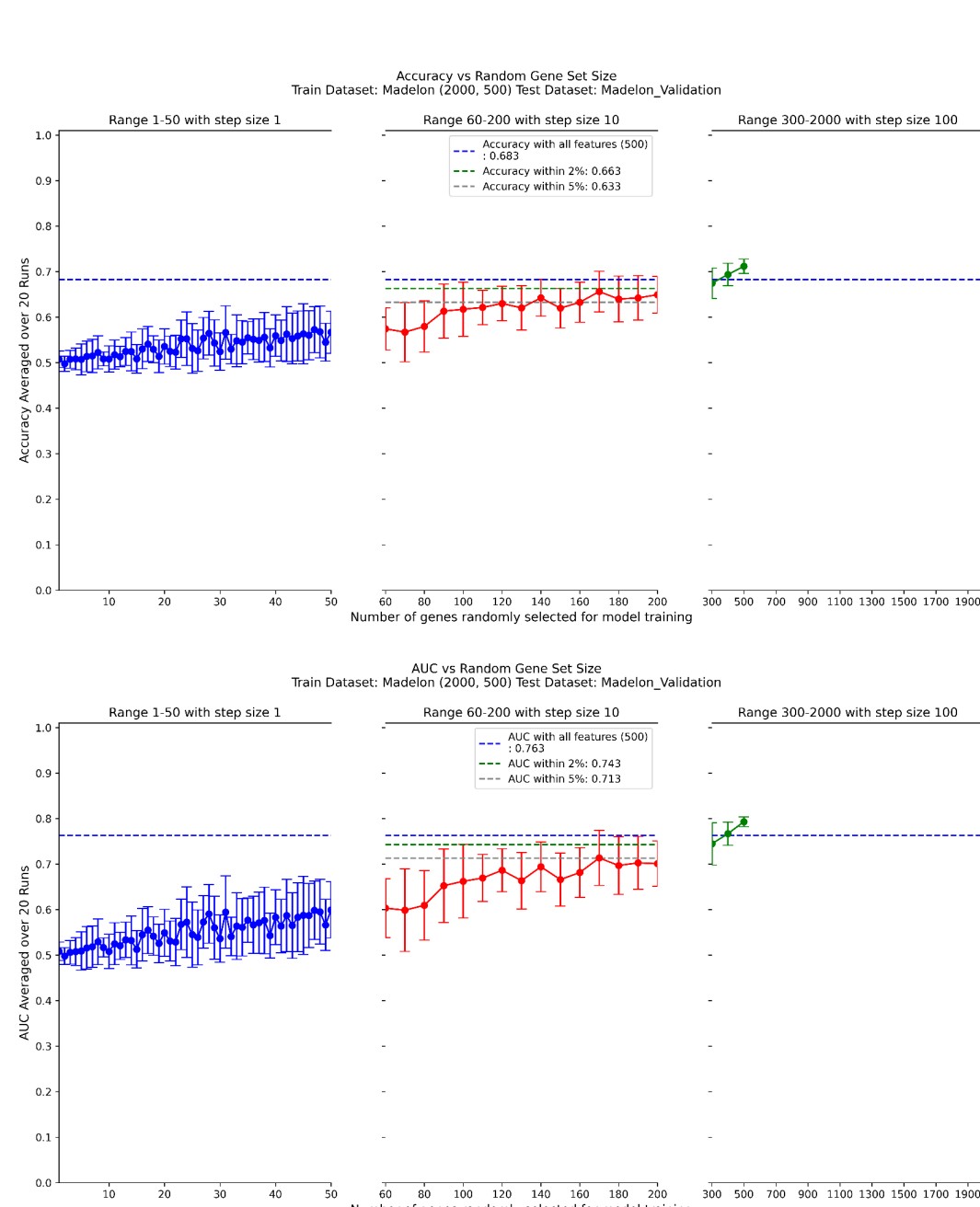

*Figure 28.* Random Forest performance with Madelon dataset (mean and standard deviation are reported over 20 runs). The task of MADELON is to classify random data. Models trained and tested on 80:20 split shows that a random subset is able to match within-5% Accuracy and AUC of all features. (As there are only 500 features in this dataset, there is no result beyond 500).

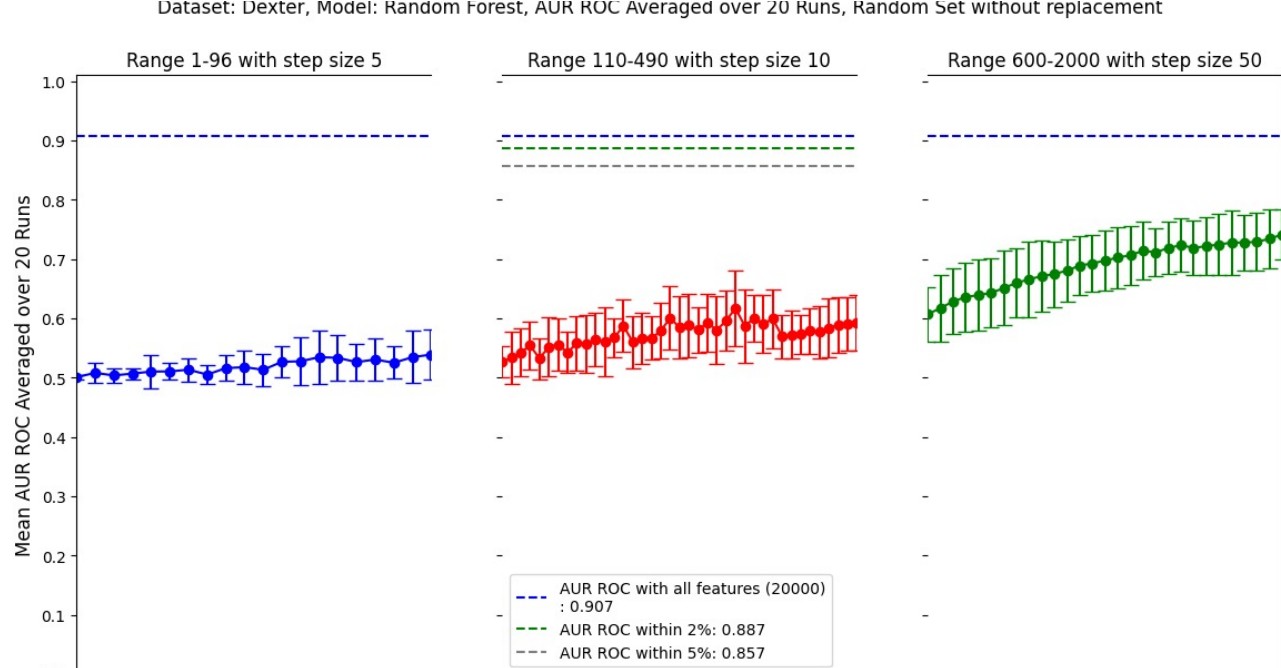

*Figure 29.* Random Forest performance with Dexter dataset (mean and standard deviation are reported over 20 runs). The task of DEXTER is to filter texts about "corporate acquisitions". Models trained and tested on 80:20 split shows that a random subset is NOT able to match AUC of all features.

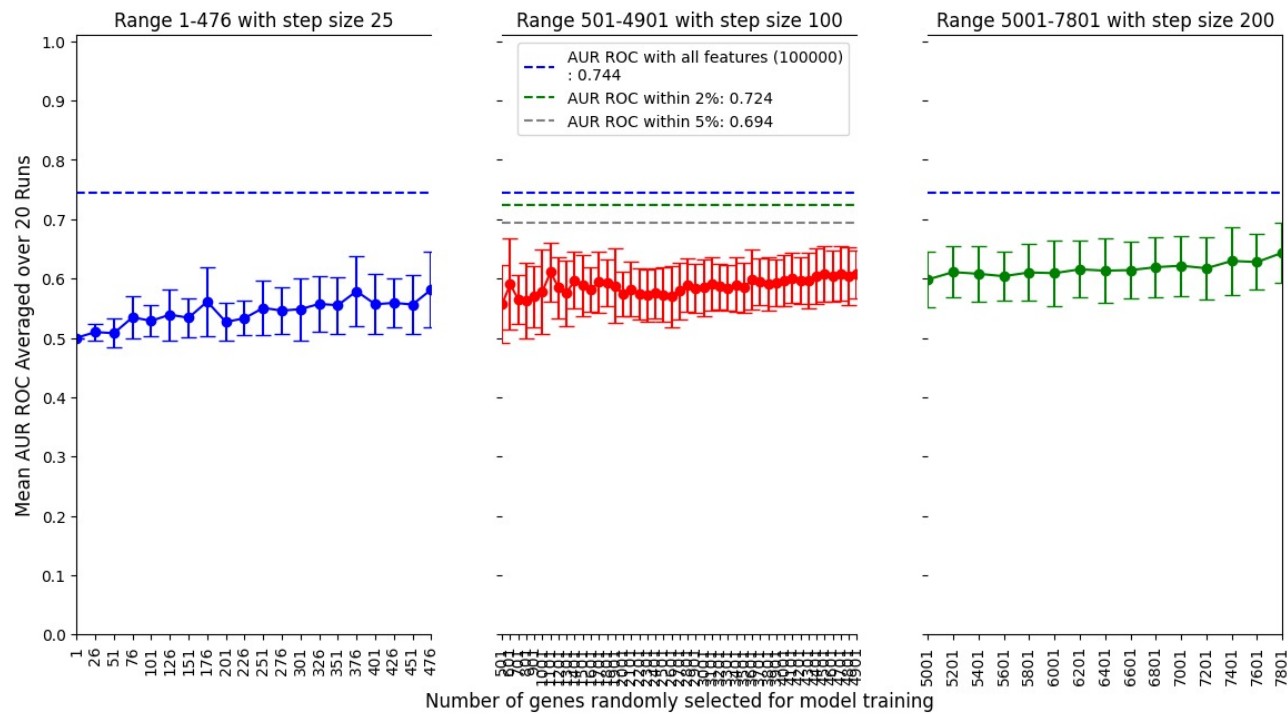

*Figure 30.* Random Forest performance with Dorothea dataset (mean and standard deviation are reported over 20 runs). The task of DOROTHEA is to predict which compounds bind to Thrombin. Models trained and tested on 80:20 split shows that a random subset is NOT able to match AUC of all features.

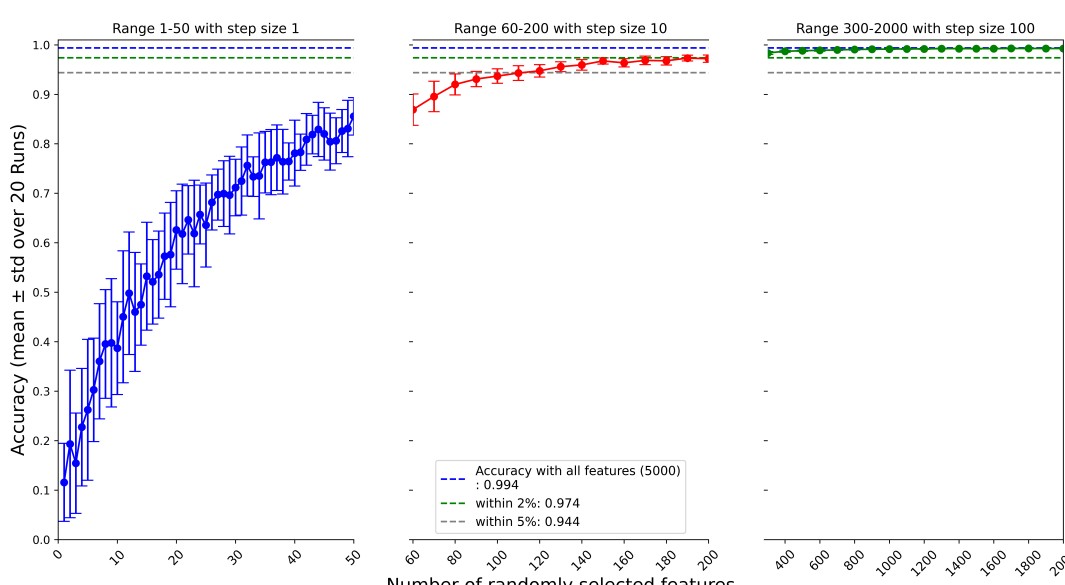

*Figure 31.* Random Forest performance with lung cancer sc-RNA-Seq Qian 2020 dataset (mean and standard deviation are reported over 20 runs). Models trained and tested on 80:20 split shows that any random subset of 100 features (2% of 5000) achieves within 5% accuracy of 5000 HVGs.

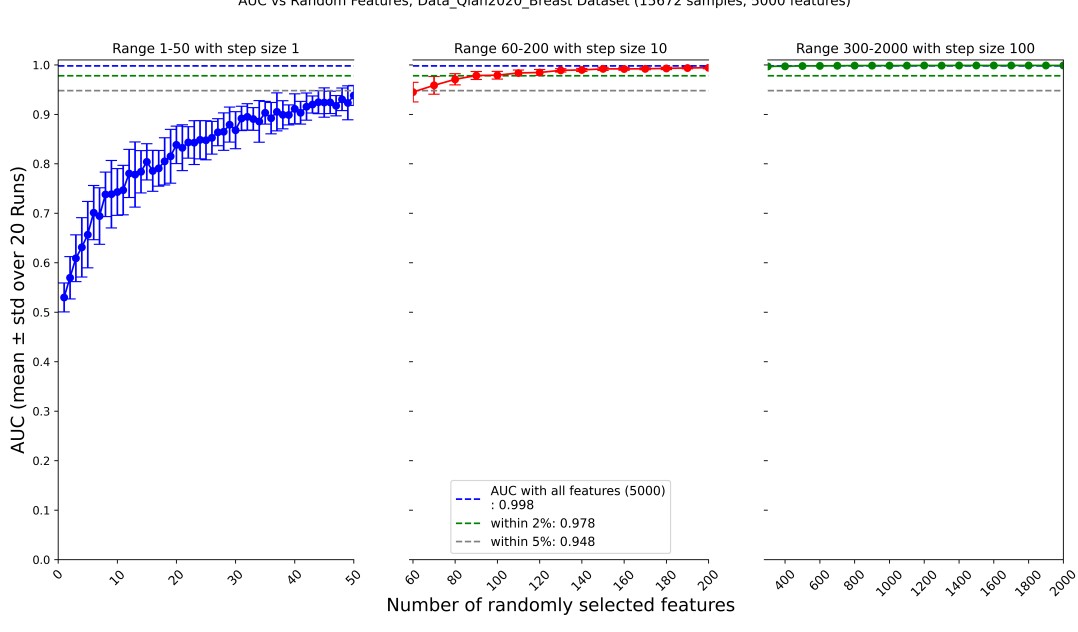

*Figure 32.* Random Forest performance with lung cancer sc-RNA-Seq Qian 2020 dataset (mean and standard deviation are reported over 20 runs). Models trained and tested on 80:20 split shows that any random subset of 100 features (2% of 5000) achieves within 5% AUC of 5000 HVGs.

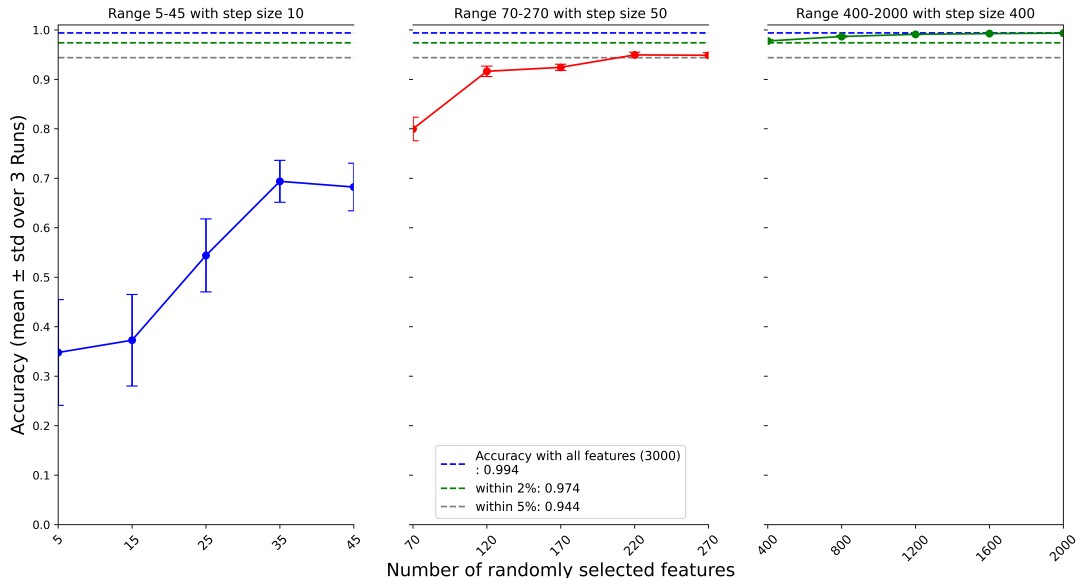

*Figure 33.* Random Forest performance with lung cancer sc-RNA-Seq Wu 2021 dataset (mean and standard deviation are reported over 20 runs). Models trained and tested on 80:20 split shows that any random subset of 200 features achieves within 5% accuracy of 3000 HVGs.

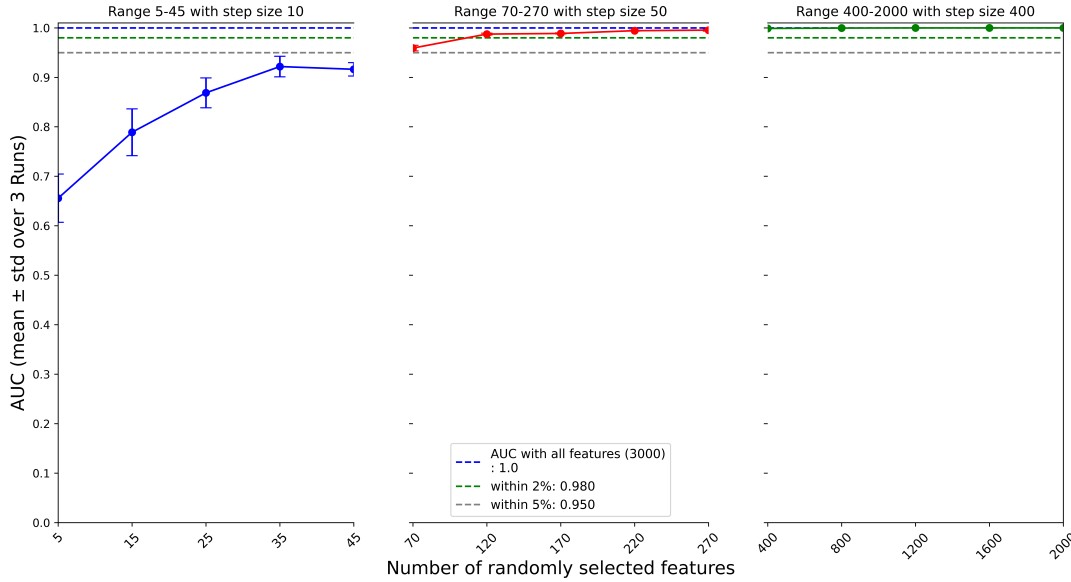

*Figure 34.* Random Forest performance with lung cancer sc-RNA-Seq Wu 2021 dataset (mean and standard deviation are reported over 20 runs). Models trained and tested on 80:20 split shows that any random subset of 200 features achieves within 5% AUC of 3000 HVGs.

## C. Results with other models

Additional figures supporting the main text are provided here (figs. 35–45). Rest of the plots are in the supplementary material. They include results for different model and datasets.

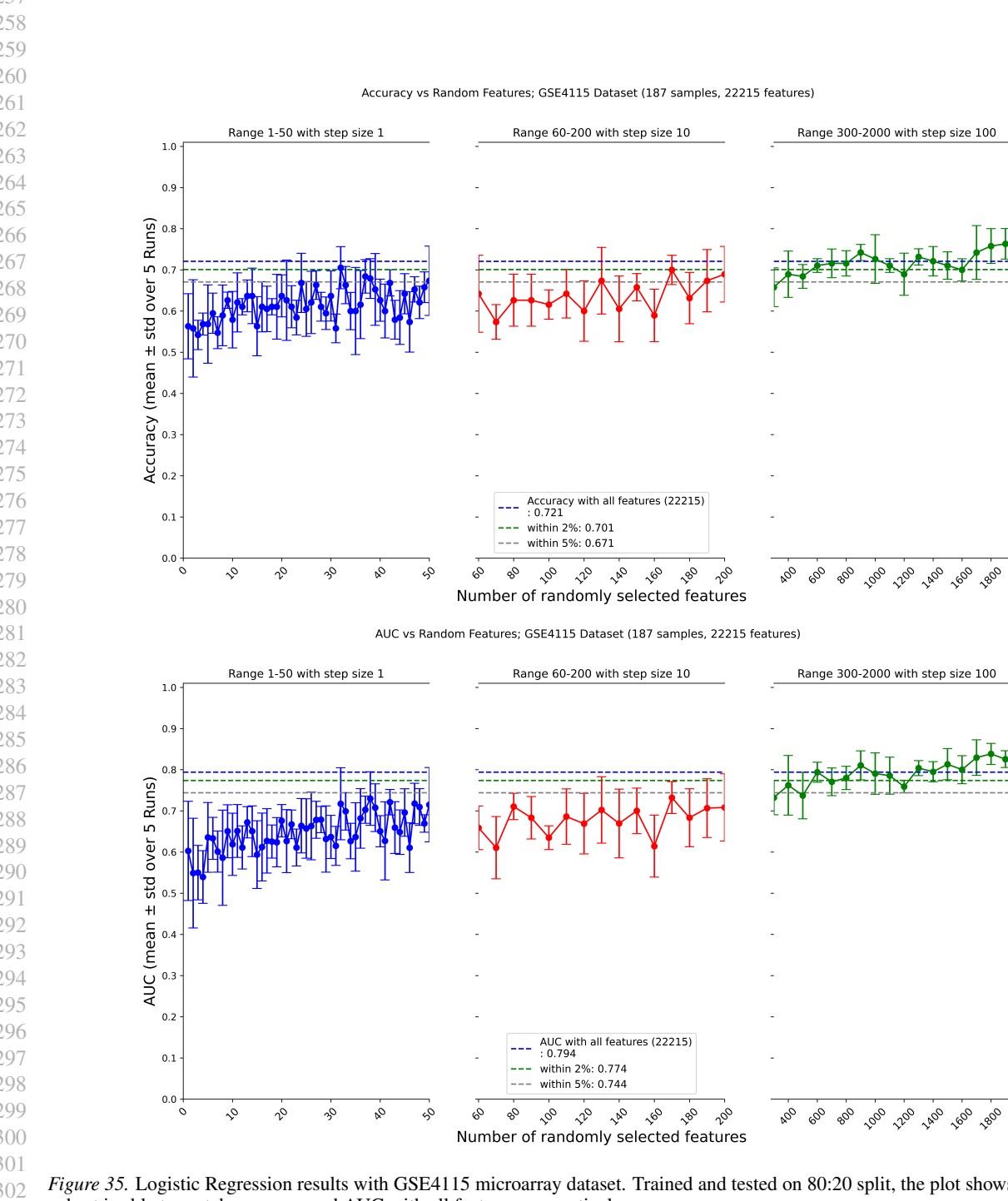

*Figure 35.* Logistic Regression results with GSE4115 microarray dataset. Trained and tested on 80:20 split, the plot shows that a random subset is able to match accuracy and AUC with all features, respectively.

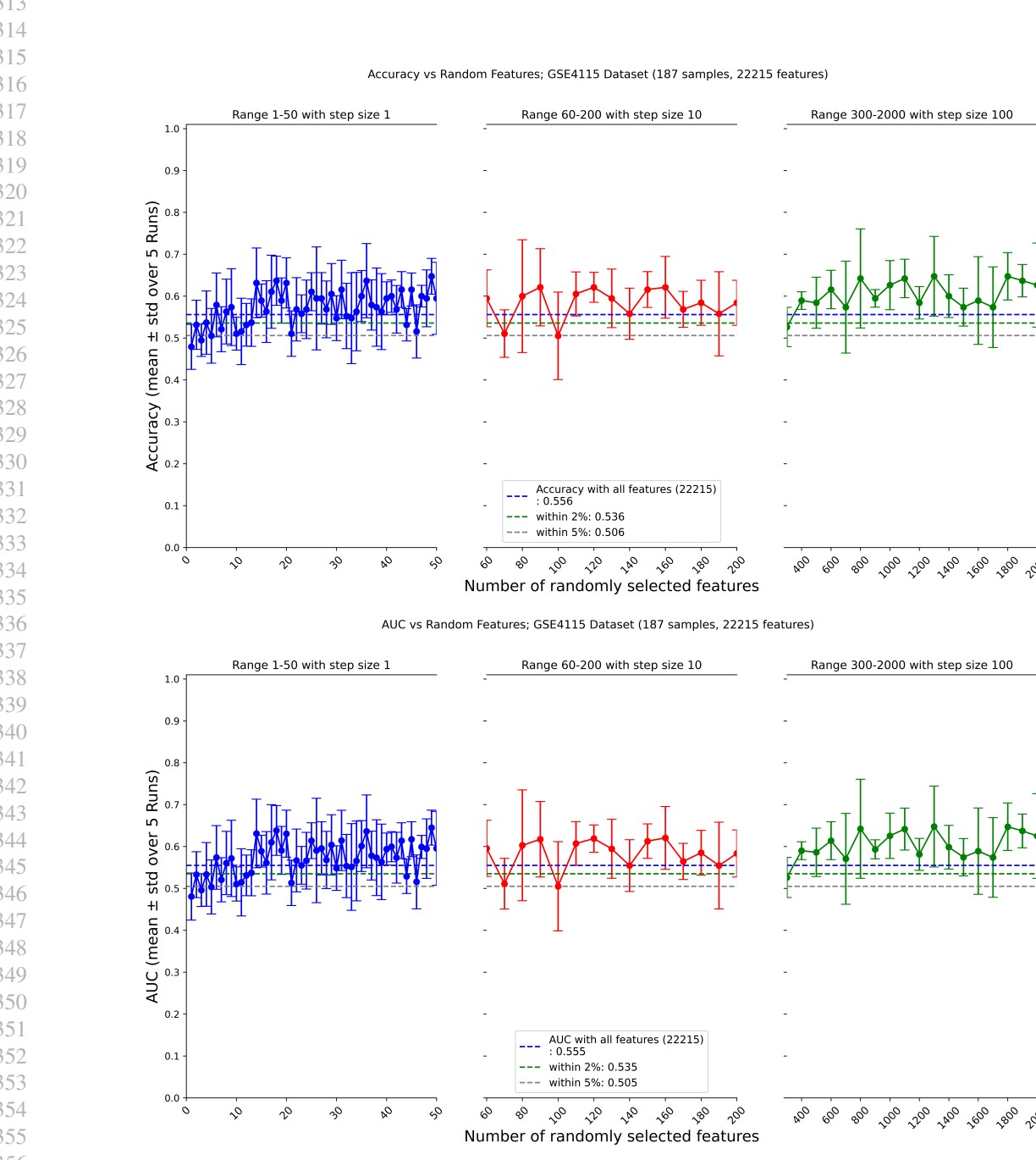

*Figure 36.* Decision Tree results with GSE4115 microarray dataset. Trained and tested on 80:20 split, the plot shows that a random subset is able to match accuracy and AUC with all features, respectively.

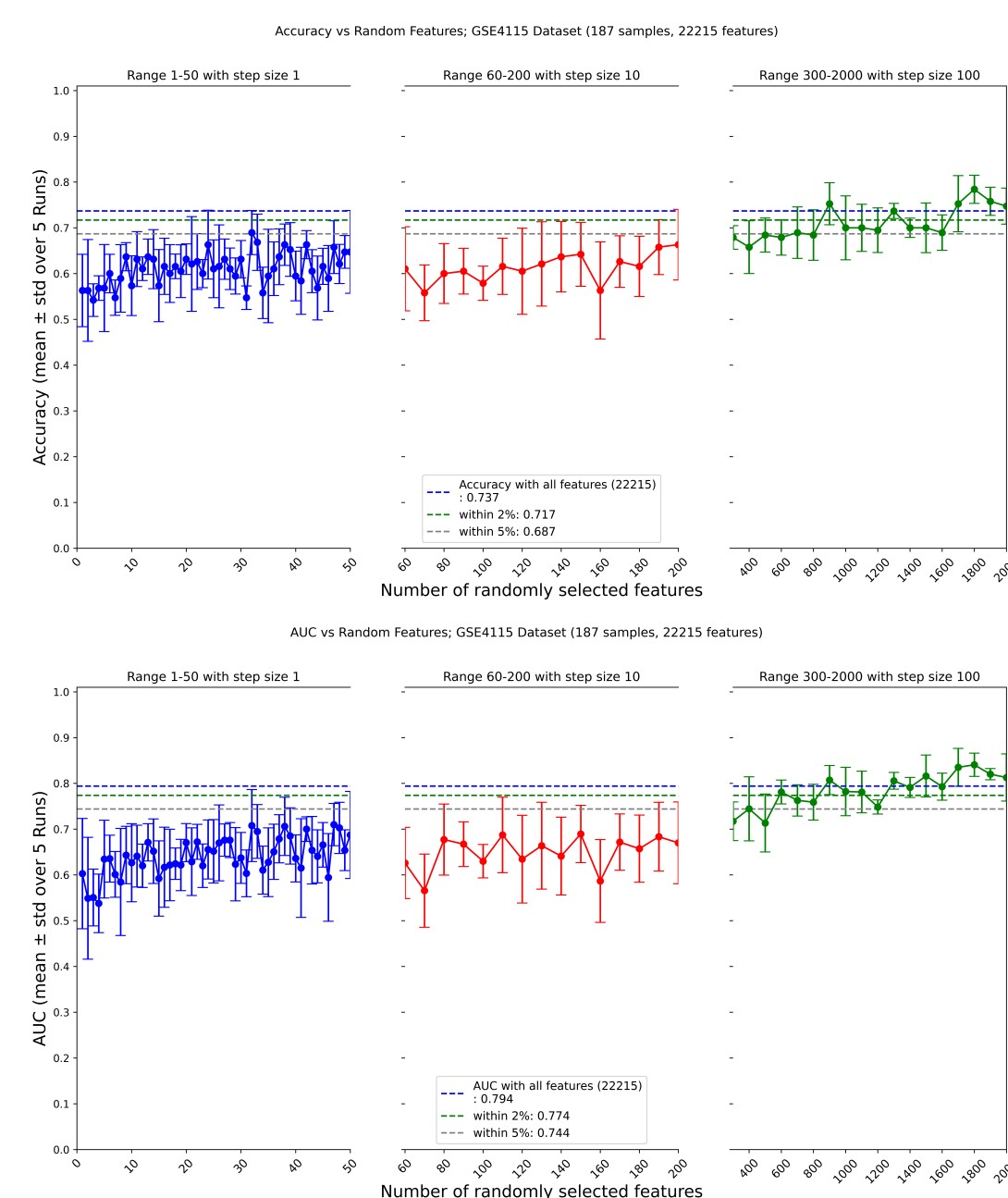

*Figure 37.* Support Vector Machine (SVM) results with GSE4115 microarray dataset. Trained and tested on 80:20 split, the plot shows that a random subset is able to match accuracy and AUC with all features, respectively.

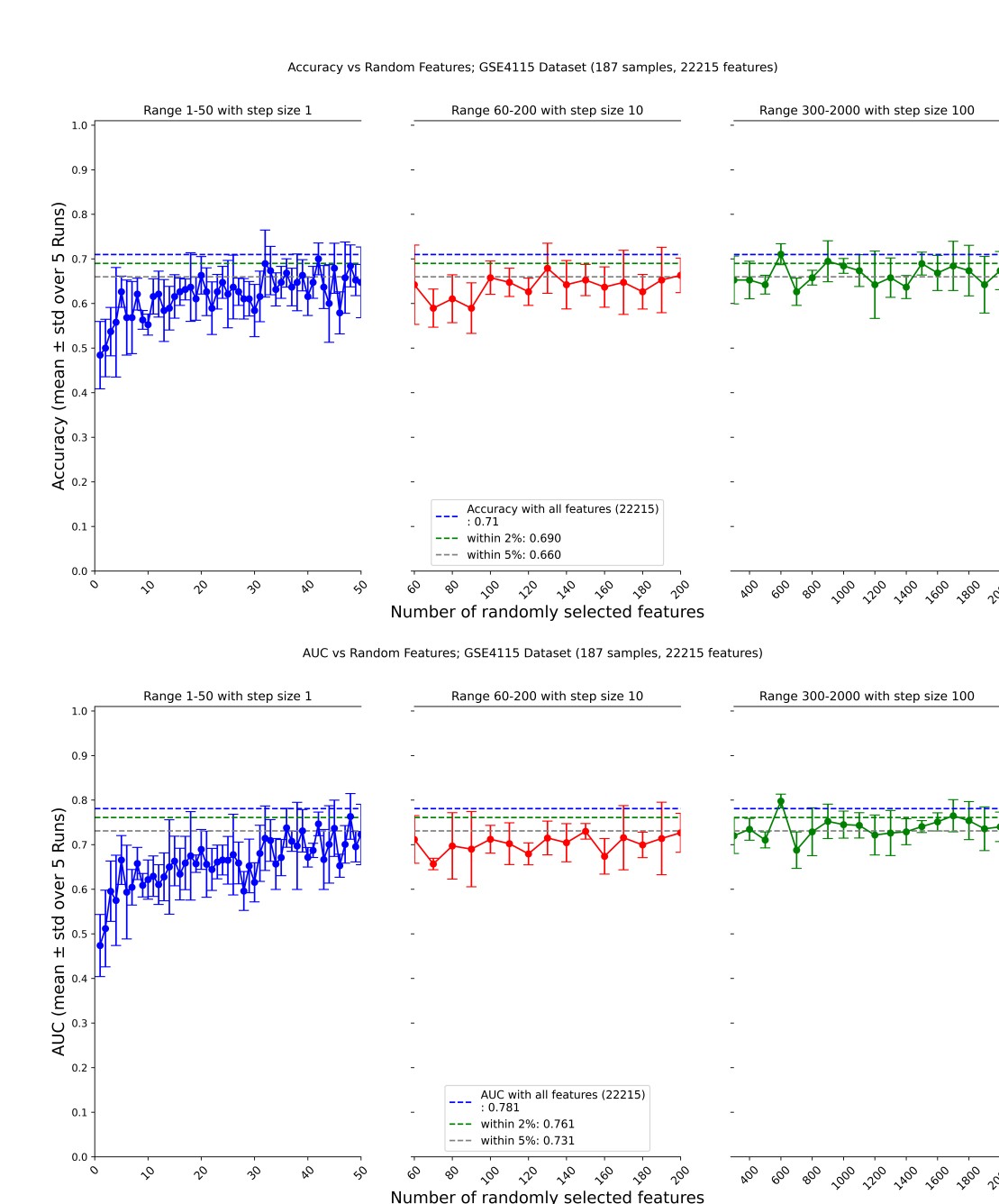

*Figure 38.* eXtreme Gradient Boosting (XGB) results with GSE4115 microarray dataset. Trained and tested on 80:20 split, the plot shows that a random subset is able to match accuracy and AUC with all features, respectively.

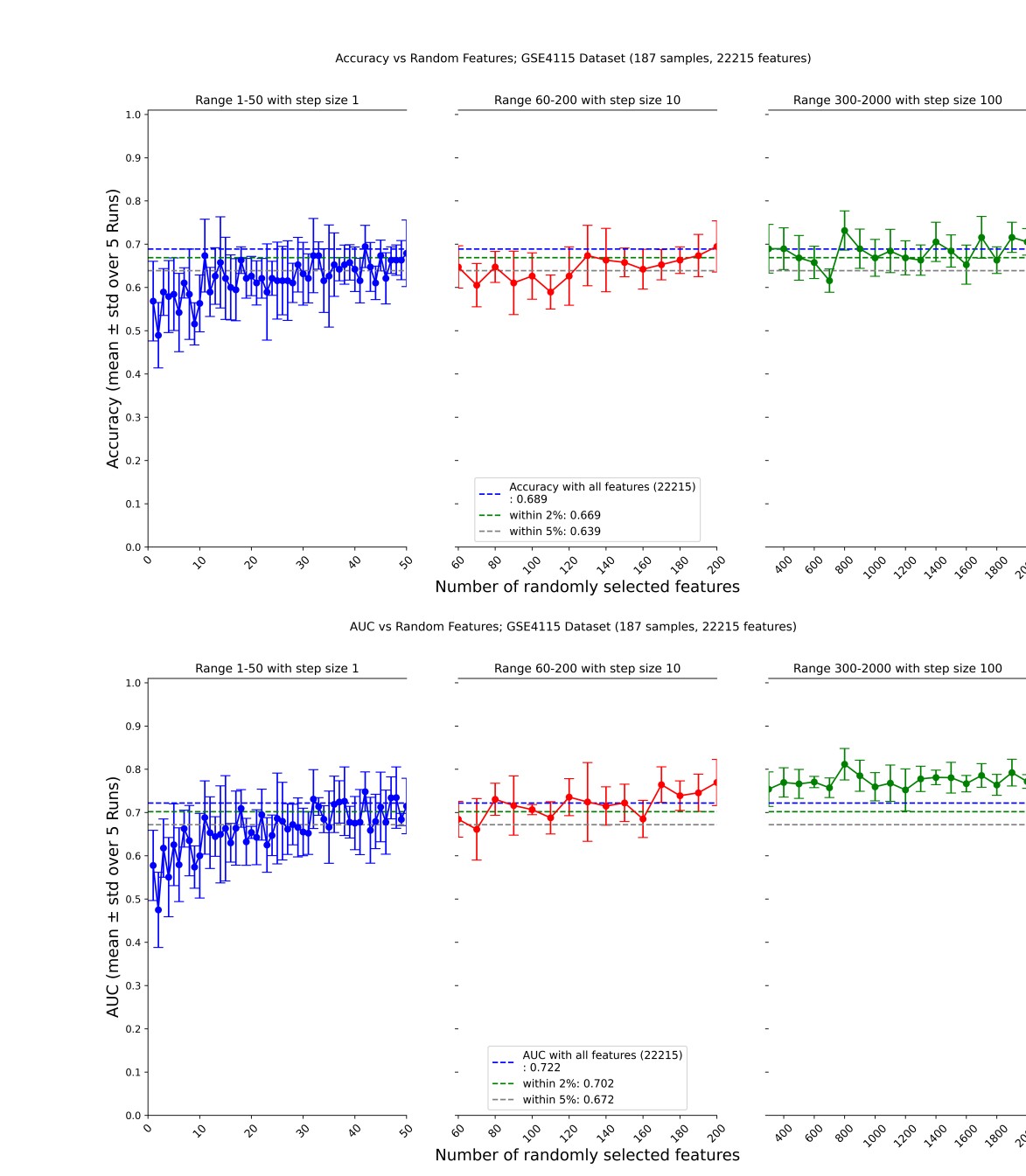

*Figure 39.* Multi Layer Perceptron (MLP) results with GSE4115 microarray dataset. Trained and tested on 80:20 split, the plot shows that a random subset is able to match accuracy and AUC with all features, respectively.

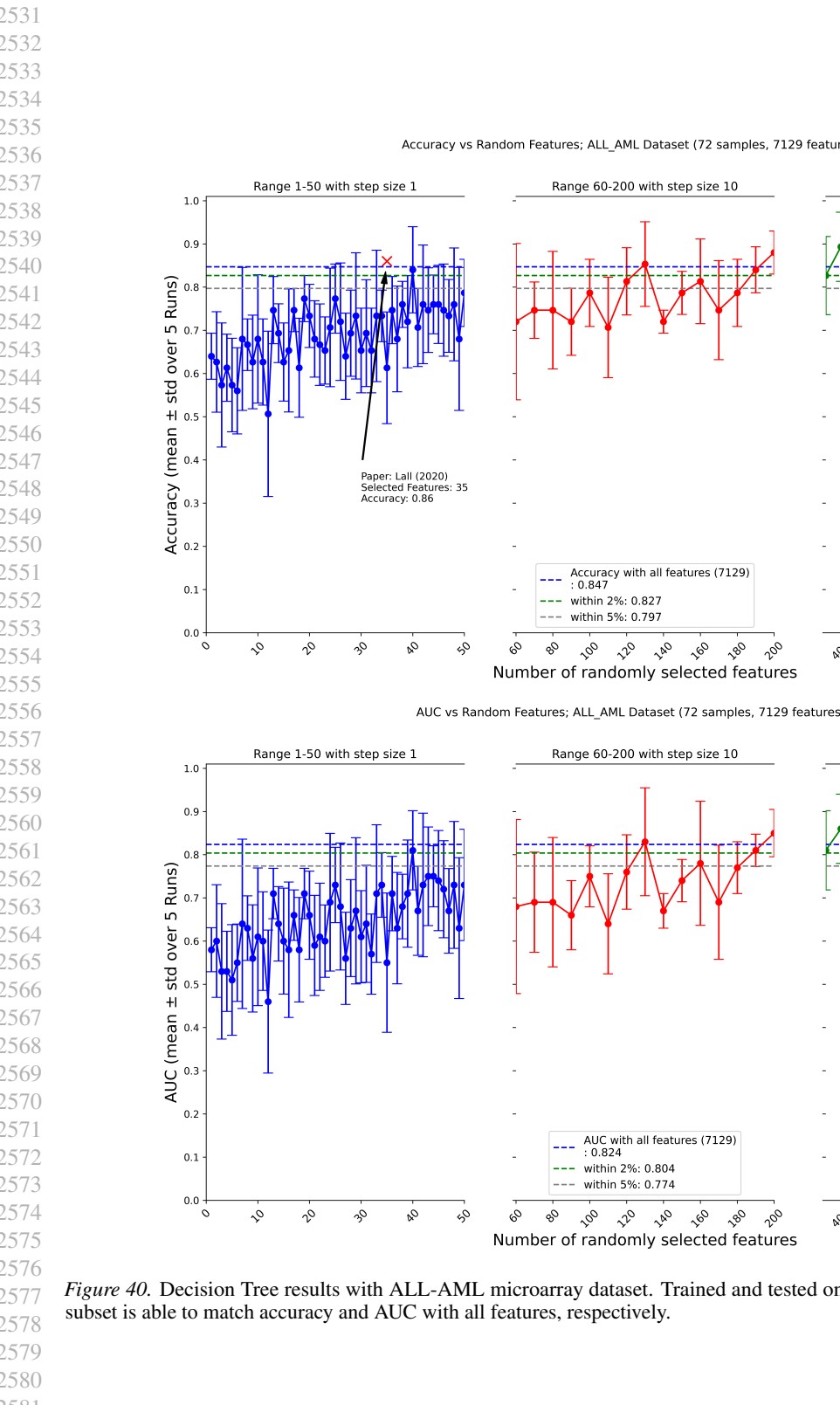

*Figure 40.* Decision Tree results with ALL-AML microarray dataset. Trained and tested on 80:20 split, the plot shows that a random subset is able to match accuracy and AUC with all features, respectively.

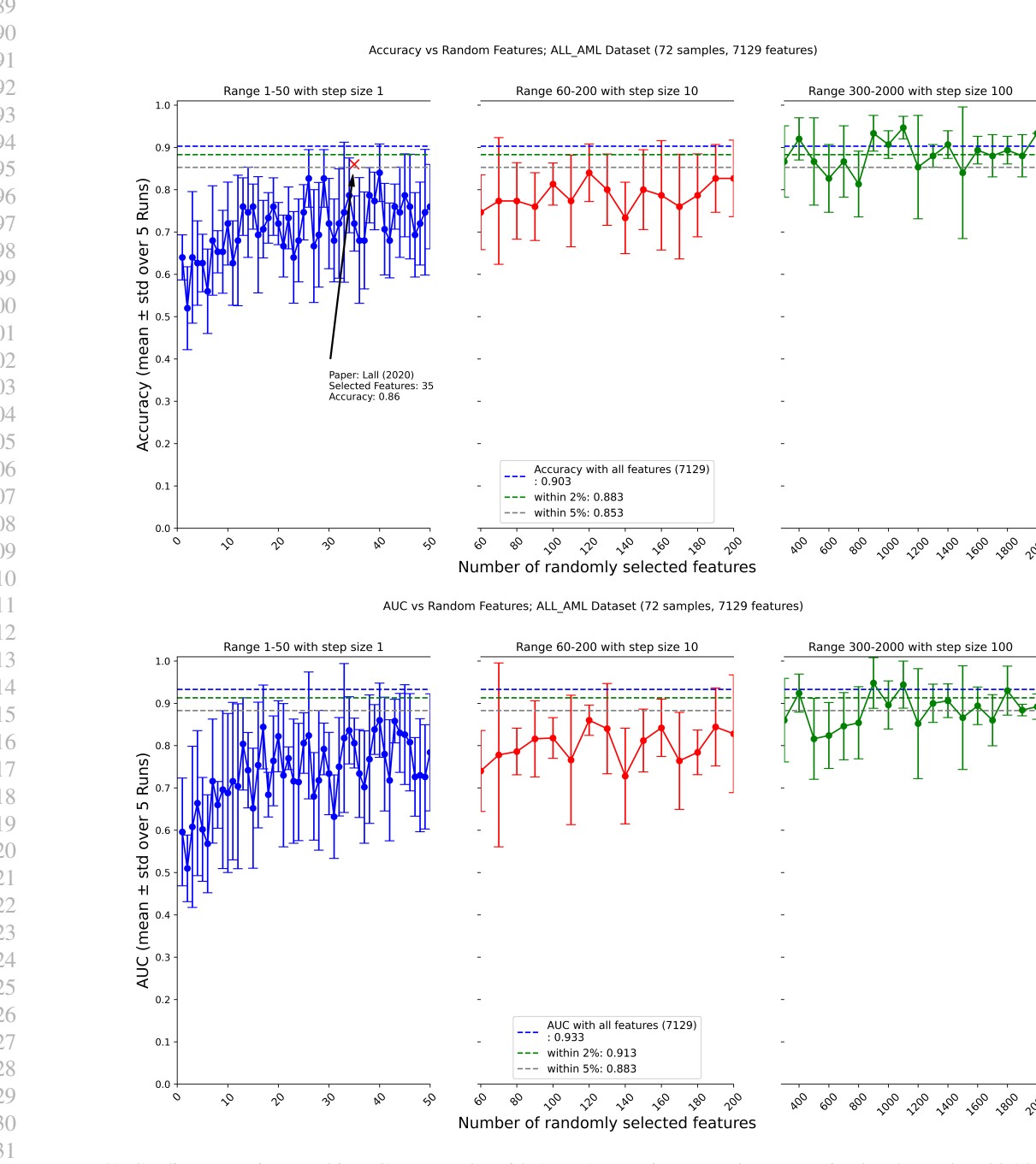

*Figure 41.* Gradient Boosting Machine (GBM) results with ALL-AML microarray dataset. Trained and tested on 80:20 split, the plot shows that a random subset is able to match accuracy and AUC with all features, respectively.

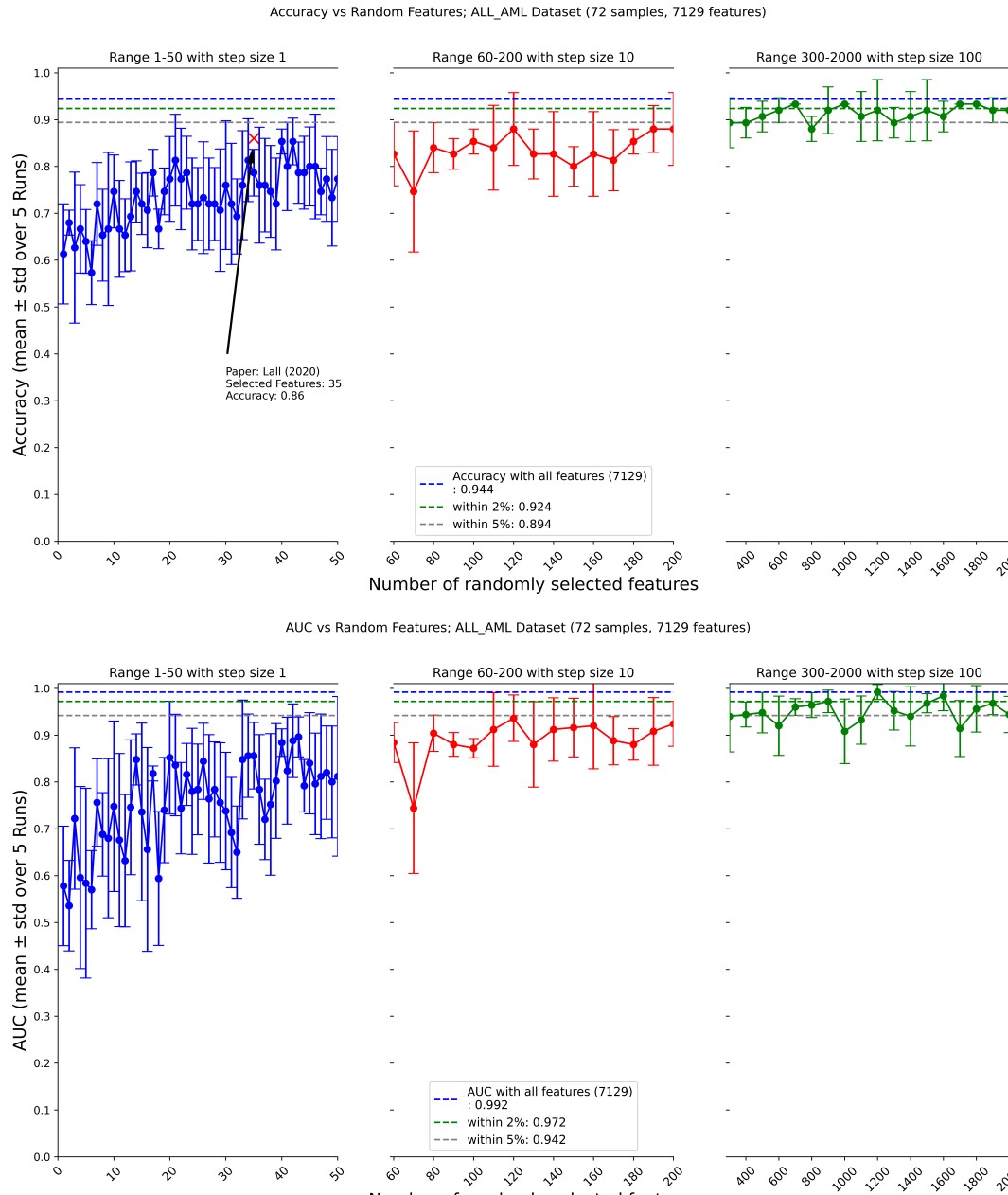

*Figure 42.* HistGradient Boosting classifier (HistGB) results with ALL-AML microarray dataset. Trained and tested on 80:20 split, the plot shows that a random subset is able to match accuracy and AUC with all features, respectively.

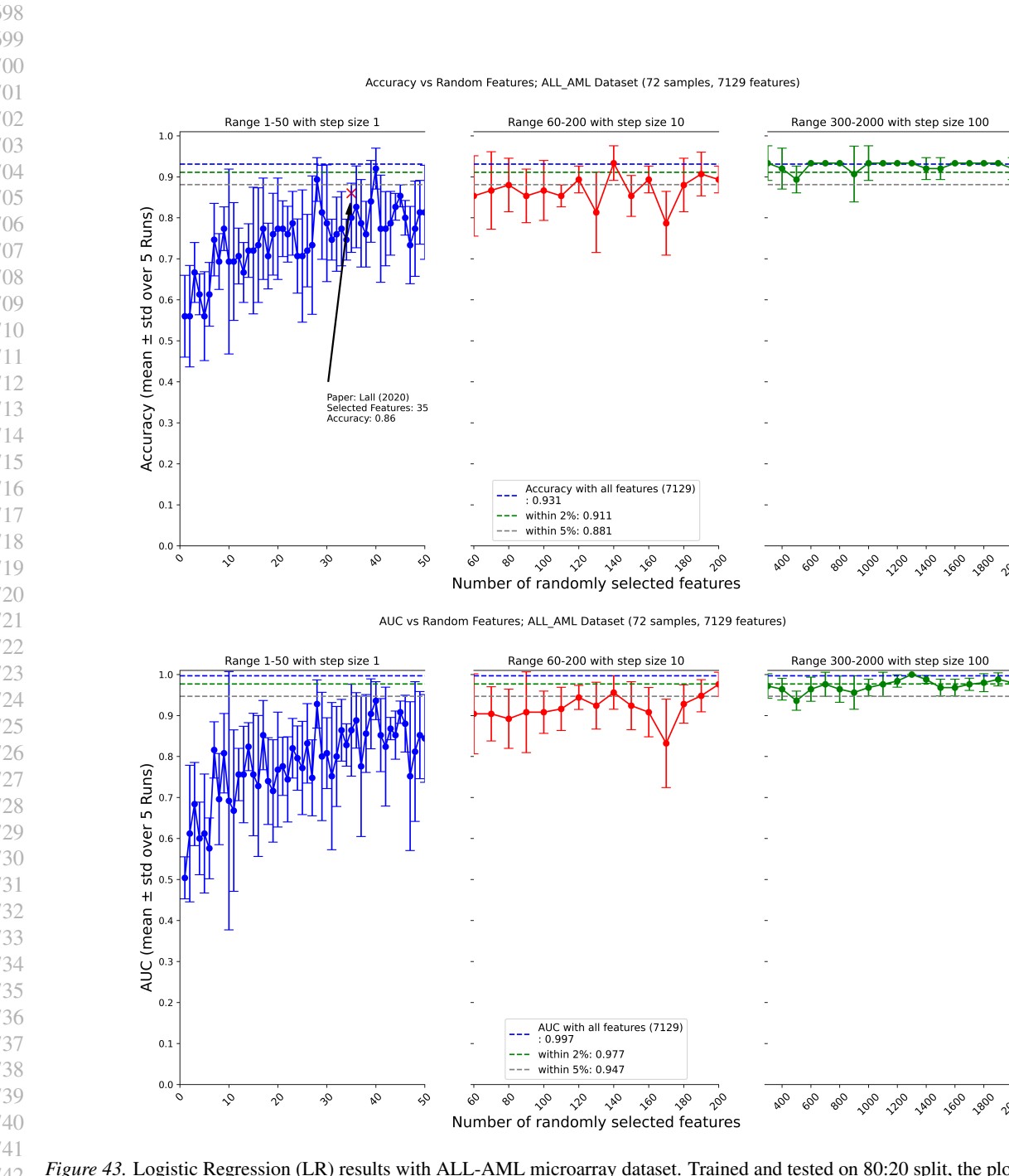

*Figure 43.* Logistic Regression (LR) results with ALL-AML microarray dataset. Trained and tested on 80:20 split, the plot shows that a random subset is able to match accuracy and AUC with all features, respectively.

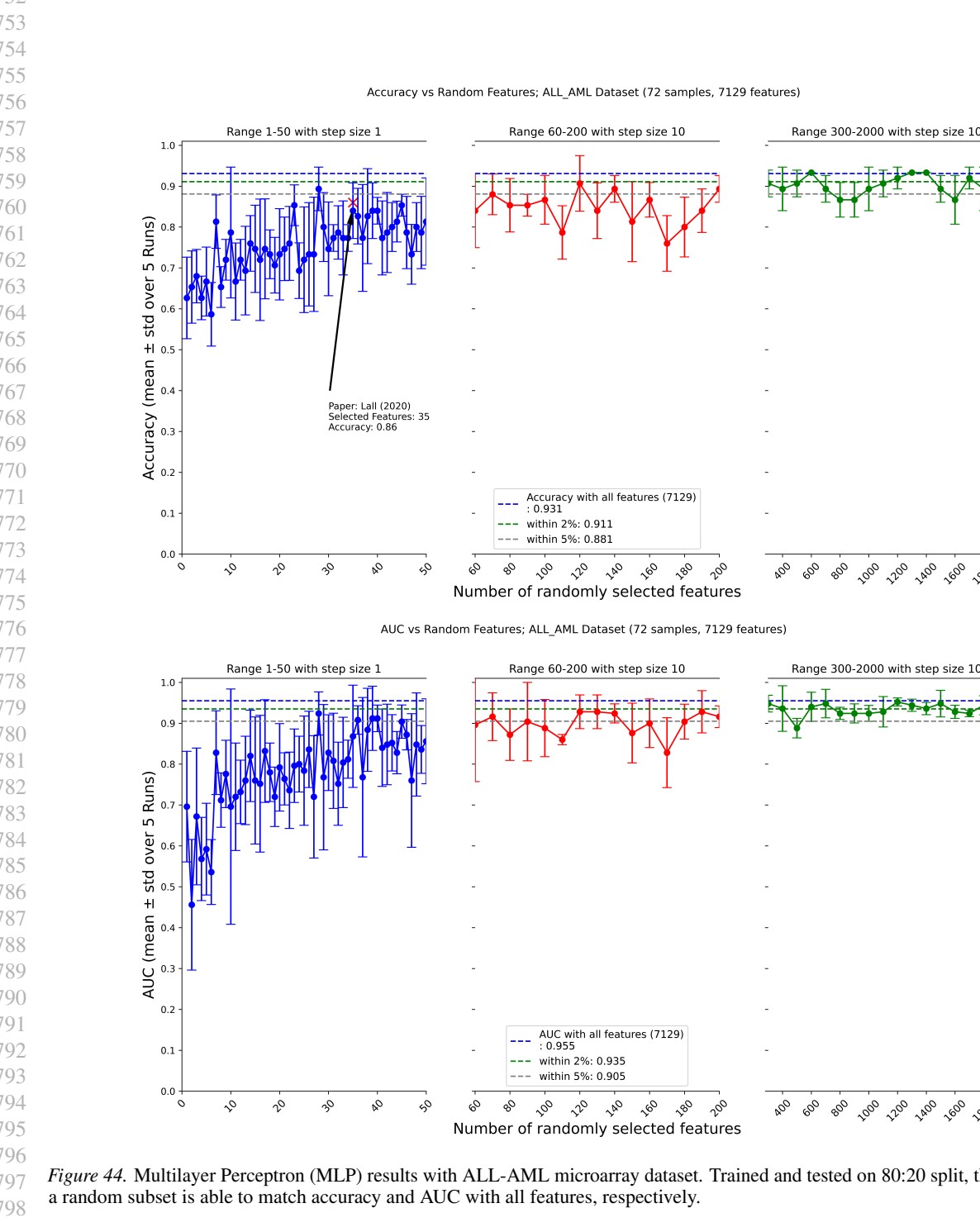

*Figure 44.* Multilayer Perceptron (MLP) results with ALL-AML microarray dataset. Trained and tested on 80:20 split, the plot shows that a random subset is able to match accuracy and AUC with all features, respectively.

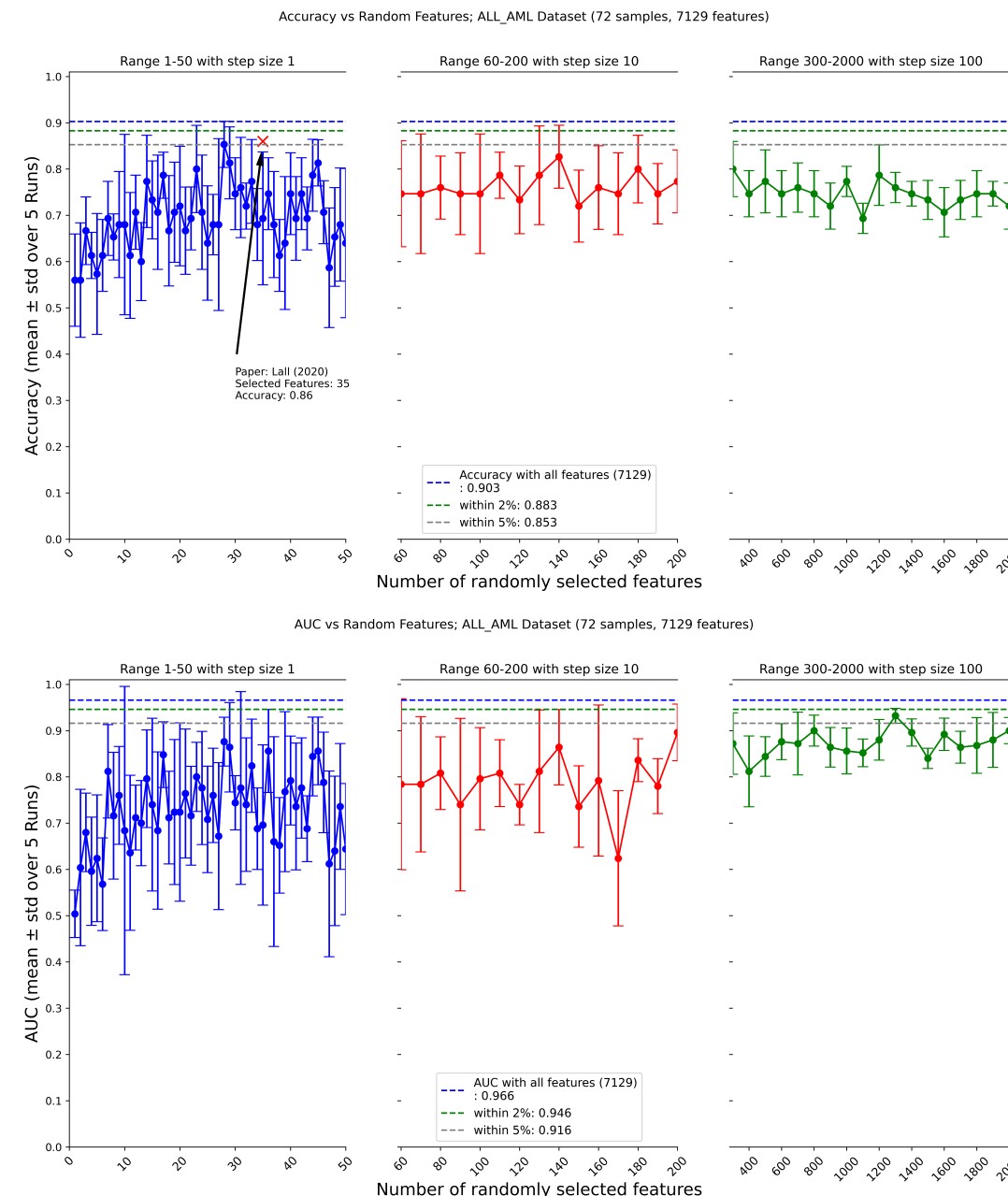

*Figure 45.* Ridge classifier results with ALL-AML microarray dataset. Trained and tested on 80:20 split, the plot shows that a random subset is able to match accuracy and AUC with all features, respectively.

