# OpenReview forum: "Predictive Performance is Often Insensitive to Feature Selection in High-Dimensional Biological Classification"
_ICML.cc/2026/Conference — Submitted to ICML 2026_

### Official Review · Reviewer_tS5D · 2026-03-01

**Soundness:** 3
**Presentation:** 3
**Significance:** 2
**Originality:** 2
**Overall Recommendation:** 3
**Confidence:** 3

**Summary:**

This paper presents a large-scale empirical evaluation of feature selection (FS) efficacy in high-dimensional biological classification tasks. By systematically analyzing 30 diverse benchmarks (including microarray, bulk RNA-Seq, mass spectrometry, and imaging data), the authors demonstrate a striking phenomenon: extremely small, randomly selected subsets of features (0.02-1% of the total) frequently achieve predictive performance comparable to—and sometimes statistically indistinguishable from—models trained on the entire feature space. The paper concludes that high predictive accuracy is insufficient to claim the biological or mechanistic importance of specific selected features due to massive signal redundancy.

**Compliance With Llm Reviewing Policy:**

Affirmed.

**Key Questions For Authors:**

In your study, you mention that random subsets still match the full-model performance in scRNA-Seq data, which is sparse (over 96% zeros). Could you elaborate on how the sparsity of the data might influence the selection of random features and the performance of the models? Are there any strategies you would suggest for dealing with this sparsity in future studies? The Minimum Sufficient Random Sample Size (MSRSS) is an interesting metric introduced in your paper. Can you provide further insights into its potential applications?  Could you provide more details on how redundancy in features might affect the interpretability of machine learning models, especially when it comes to identifying biologically meaningful features?

**Limitations:**

No. The authors have not adequately discussed the potential negative societal impacts of their work. An "Impact Statement" (which is required for ICML 2026) are currently missing.

**Strengths And Weaknesses:**

Strengths: The evaluation is extensive. Testing across 30 distinct datasets provides robust evidence for the paper's claims, mitigating the risk that the observed phenomenon is an artifact of a specific dataset or modality. The paper is generally well-written, and the core premise is easy to follow. The visual representations of the performance curves (as seen in the provided appendix/figures) effectively communicate the rapid plateau of accuracy/AUC with random feature subsets.

Weaknesses: The manuscript lacks an explicit impact statement, which is a requirement for ICML 2026. An impact statement should succinctly describe how this work advances the field and the potential future applications or influence it may have. There appears to be an issue with Table reference citation on page 7, line 351, where "See Table ??" is noted. This should be fixed to correctly reference the appropriate table number to avoid confusion. It would be helpful to explicitly compare the variance of predictions made by random subsets versus those made by established feature selection algorithms (e.g., LASSO, Random Forest feature importance) to see if "smart" selection offers benefits in model stability, even if mean accuracy is similar.

---

> ### Author Rebuttal · Authors · 2026-03-31
>
> We thank the reviewer for the positive and thoughtful review. We are encouraged that the reviewer finds the empirical observation interesting and the large-scale evaluation convincing. We also appreciate the concrete suggestions for improving the paper, and we agree that these changes will strengthen the final version.
>
> **(1) on impact statement and presentation issues**
>
> Thank you for pointing this out. we will add an explicit impact statement in the revision, including both the positive implications of the work and the potential negative consequences of over- or misinterpretation.
>
> In particular, we will emphasize that the main positive contribution of the paper is to encourage more rigorous evaluation of feature selection claims in high-dimensional biological settings, especially where selected features are interpreted as biomarkers or mechanistic signals. A practical takeaway is that random-subset baselines should be reported more routinely before strong biological or mechanistic claims are made from predictive models.
>
> At the same time, we will also clarify the possible negative risk that our findings could be misread as implying that feature selection is unnecessary in all settings, which is not our claim.
>
> We have now fixed the broken table reference (“See Table ??”) noted by the reviewer.
>
> **(2) on variance / stability comparisons**
>
> This is a very good suggestion. One of the central observations in the paper is that performance across random subsets is often surprisingly stable once subset size passes a dataset-specific threshold. We agree that it would be useful to more explicitly compare this against the variance or stability observed from standard FS methods such as lasso and random forest feature importance.
>
> We agree this is a useful extension, and if it cannot be included in the main paper due to space, we will include it in the supplement and discuss it explicitly in the revised manuscript. This would help clarify whether “clever” feature selection offers benefits in stability, even when mean predictive performance is similar.
>
> **(3) on sparsity in scRNA-seq data**
>
> This is an important point. We agree that the sparsity of scRNA-seq data likely interacts with the observed behavior in a nontrivial way. One possible reason random subsets can still perform competitively is that many genes may carry overlapping information about broad cell-type or state differences, even in sparse settings. At the same time, sparsity can also make some tasks require larger subset sizes than in certain microarray datasets, which is consistent with the more heterogeneous behavior we observe across datasets. More broadly, we see sparsity, redundancy, and effective dimensionality as closely related factors that can influence how quickly predictive performance saturates with random subset size.
>
> For future work, we agree that it would be valuable to study how different preprocessing choices (e.g. filtering low-expression genes, normalization, imputation, or feature grouping) affect this phenomenon in scRNA-seq specifically.
>
> **(4) on MSRSS and interpretability**
>
> Our motivation for introducing MSRSS is to provide a simple dataset-level summary of how quickly predictive signal saturates as random subset size increases. We see it as potentially useful in at least three ways:
> as a compact summary of dataset redundancy / predictive saturation,
> as a benchmark diagnostic when evaluating FS methods,
> and as a way to compare how sensitive different datasets are to feature subset size.
>
> We agree that the current draft does not explain these possible uses clearly enough, and we will improve this in the revision. More broadly, the main interpretability implication of our results is that when many distinct feature subsets achieve similar predictive performance, it becomes much harder to argue that any one selected subset is uniquely meaningful. In that sense, predictive sufficiency is not the same as feature importance or biological uniqueness. We will revise the discussion to make this distinction more explicit.
>
> Overall, we appreciate the reviewer’s constructive comments. We believe the suggested revisions will improve the paper substantially.

---

### Official Review · Reviewer_oZ51 · 2026-03-03

**Soundness:** 1
**Presentation:** 2
**Significance:** 2
**Originality:** 2
**Overall Recommendation:** 2
**Confidence:** 5

**Summary:**

The paper challenges the conventional wisdom that sophisticated feature selection (FS) is required to achieve high predictive performance in high-dimensional settings, particularly in biological domains. Through an extensive empirical study of 30 datasets, the authors show that models trained on tiny, randomly sampled feature subsets often achieve results statistically indistinguishable from those using the full feature set or features selected by standard methods like LASSO or Random Forest importance. The work introduces the Minimum Sufficient Random Subset Size (MSRSS) to quantify this saturation of predictive signal

**Compliance With Llm Reviewing Policy:**

Affirmed.

**Final Justification:**

The paper identifies an interesting empirical phenomenon in biological data. However, in my view, the work still lacks a sufficiently rigorous theoretical foundation, a thorough empirical evaluation, and fair comparisons with the most relevant baselines. Therefore, I remain on the side of rejection.

**Key Questions For Authors:**

The empirical study appears to focus heavily on microarray datasets. Could the authors clarify the rationale for this choice and discuss how it may affect the generality of the conclusions?

**Limitations:**

No. The discussion of limitations should more clearly acknowledge that the main empirical observations are derived predominantly from biological datasets. This is important because such datasets often have domain-specific properties, such as high dimensionality, redundancy, and low signal-to-noise ratio, that may make random feature subsets appear unusually competitive.

**Strengths And Weaknesses:**

**Strength**
* The study covers 30 biological datasets across various modalities, including microarray, RNA-Seq, and mass spectrometry.

**Weakness**
* **Depth of Theoretical Explanation:** While the authors attribute these findings to "substantial redundancy" and "low-dimensional separability", the paper remains largely empirical. A more formal analysis linking these observations to the intrinsic dimensionality of the data or specific properties of biological manifolds would strengthen the contribution.
* **Domain Specificity:** The dataset collection is heavily skewed toward genomics datasets (21 of 30 datasets are microarray). As biological data is often characterized by high noise-to-signal ratios and high feature correlation, the claim that these findings "challenge the conventional assumption" of FS would be more compelling if validated on a broader array of non-biological, high-dimensional tabular datasets.
* **Baseline comparisons:** The baselines used (LASSO, Elastic Net, RF Importance) are well-established but standard. The paper would benefit from comparing random selection against more recent, sophisticated FS techniques such as differentiable feature selection or those specifically designed for tabular deep learning (e.g., variants of TabNet or SAINT).

---

> ### Author Rebuttal · Authors · 2026-03-31
>
> We thank the reviewer for reading the paper carefully and for the useful comments. We are encouraged that the reviewer recognizes the central empirical finding and the value of studying random feature subsets as a baseline. We also agree that the current draft should do a better job of clarifying the intended scope of the claims and positioning the paper as an empirical benchmark / cautionary study, rather than as a theoretical treatment of feature selection.
>
>
> **(1) on theoretical explanation**
>
> Our main goal in this paper is to empirically test a class of commonly made assumptions, and to document a simple but surprisingly under-tested phenomenon across a broad benchmark set. We view this paper as primarily establishing the empirical pattern clearly enough that it deserves explanation.
>
> we agree that the current discussion can better connect the observed results to possible mechanisms such as:
> - strong feature redundancy,
> - low effective / intrinsic dimensionality,
> - class separability
>
> We also see a more formal theoretical treatment as an important direction for future work.
>
> **​​(2) on empirical scope and dataset composition**
>
> We agree that the benchmark collection is weighted toward biological datasets, especially transcriptomic ones. We emphasize this fact even in the title of the paper, since our motivation comes directly from the way feature selection is commonly used and interpreted in computational biology, where selected features are often treated as biomarkers or biologically meaningful signatures. Many of the strongest practical claims around FS are made in exactly these settings: small-sample, high-dimensional biological classification tasks. For this reason, we believe this is an important and appropriate domain in which to test the random-subset baseline.
>
>
> **(3) on baseline comparisons**
>
> This is a fair point. Our current goal was to compare against simple and widely used baselines that are commonly encountered in biological and applied machine learning workflows. However, we agree that the paper should better acknowledge that the present comparison is not intended to exhaust the modern FS literature.
>
> Importantly, the main point of the paper is not that random subsets beat every sophisticated FS method, but that a very simple null baseline, subset-size-matched random feature subsets, is often surprisingly competitive. If such a simple baseline already performs strongly, then it should ideally be reported before strong claims are made about the necessity or superiority of more sophisticated feature selection procedures.
>
> We will revise the related work and discussion to make this scope clearer. Where space allows, we will also note more recent FS directions, including differentiable / neural feature selection approaches, as important future comparison points.
>
> **(4) on limitations**
>
> We agree that the limitations section should more clearly acknowledge the domain-specific nature of the benchmark collection. In the revision, we will explicitly state that these datasets may have properties, including high redundancy, correlated features, and low effective dimensionality, that can make random subsets unusually competitive. However, ***this is precisely why the result matters: these are exactly the kinds of benchmark datasets on which strong FS and biomarker claims are often made.***
>
> Overall, we appreciate the reviewer’s comments and will revise the manuscript to narrow the claims, better discuss candidate mechanisms and limitations, and more clearly position the contribution as an empirical benchmark / cautionary paper.

---

> > ### Author Rebuttal · Reviewer_oZ51 · 2026-04-01
> >
> > Thank you for the response. The author presents an interesting empirical observation in biological data, but the work still requires a more rigorous theoretical foundation, more thorough examination, and fairer comparisons against relevant baselines, as noted above. Therefore, I will maintain my current score.

---

### Official Review · Reviewer_bQNt · 2026-03-07

**Soundness:** 3
**Presentation:** 3
**Significance:** 3
**Originality:** 3
**Overall Recommendation:** 4
**Confidence:** 5

**Summary:**

This paper empirically studies the effect of feature selection in high-dimensional biological classification tasks. The authors evaluate models trained on (1) all features, (2) randomly selected feature subsets, and (3) several standard feature selection approaches across 30 datasets spanning microarray, RNA-seq, scRNA-seq, imaging, and several benchmark datasets. The main finding is that, for many datasets, extremely small random feature subsets (sometimes 0.02–1% of all features) can achieve predictive performance comparable to models trained on the full feature set or on features selected by established methods. The authors interpret this as evidence that predictive performance alone is insufficient to justify claims about the importance or biological relevance of selected features, particularly in computational genomics.
I like the core empirical observation and think the paper raises an important cautionary point for biomarker-style ML claims. However, I am not yet convinced by the breadth of the conclusions. The benchmark collection is dominated by older microarray datasets, preprocessing effects are under-discussed, common univariate filter baselines are missing, and the paper does not sufficiently explain why some datasets admit tiny random subsets while others do not. Therefore, I see this as a promising benchmark/position paper with poster potential, but I would want the claims narrowed and the analysis strengthened before supporting a higher rating.

**Compliance With Llm Reviewing Policy:**

Affirmed.

**Key Questions For Authors:**

1. How were normalization, scaling, and batch effects handled across different datasets and modalities? Could the authors clarify the full preprocessing pipeline or some quality control steps?

2. For datasets where fewer than ~20 random features match full performance, can the authors provide deeper analysis explaining why this occurs?

3. Could the authors include common univariate feature selection methods (e.g., t-test, Wilcoxon, ANOVA) as additional baselines?

4. Do the authors believe their conclusions apply broadly to biological data, or primarily to transcriptomic classification tasks? What evaluation practices do the authors recommend researchers adopt based on these findings?

**Limitations:**

There is not enough discussion about limitations and potential negative societal impact.

**Strengths And Weaknesses:**

**Strengths**
1. Interesting empirical observation.
The paper documents a striking empirical pattern: across many datasets, predictive performance appears relatively insensitive to the specific subset of features used. This is an important observation that challenges common assumptions in feature selection literature.
2. Simple but important baseline.
The use of random feature subsets as a baseline is conceptually simple but often overlooked. Demonstrating that such a baseline can match sophisticated feature selection methods is valuable for the community.
3. Large benchmark study.
The authors evaluate multiple models and datasets across several modalities, which provides a broad empirical view of the phenomenon.


**Weaknesses**

*I wrote a lot in this section just so authors can understand what I want to raise, but there is no need to address all those concerns since this is a conference submission.*

1. Dataset composition heavily favors microarray datasets

The benchmark set is dominated by microarray datasets (21 of the 30 datasets). Microarrays represent an older technology and often involve curated probe sets and strong preprocessing pipelines. As a result, the observed redundancy may partly reflect the structure of microarray data rather than a general property of biological high-dimensional data.
More modern modalities such as scRNA-seq show somewhat different behavior, often requiring larger random subsets. This raises the question of whether the paper’s conclusions generalize beyond legacy transcriptomic benchmarks.

2. Preprocessing and normalization are insufficiently discussed

High-dimensional biological datasets are extremely sensitive to preprocessing choices (sorry about raising these concerns, but this is important for biologists), including normalization, scaling, batch correction, and feature filtering. The paper provides limited discussion of these steps across different modalities.
It is therefore difficult to determine whether the observed random-subset robustness is due to intrinsic biological redundancy or to characteristics of the preprocessed benchmark datasets.

3. Missing commonly used statistical feature-selection baselines

The authors compare against embedded methods such as LASSO and random forest feature importance. However, in many biological and pharmaceutical workflows, simple univariate statistical screening methods remain standard practice (e.g., t-tests, Wilcoxon tests, ANOVA, differential-expression ranking). Including these common filter-based baselines would provide a more realistic comparison to widely used feature-selection pipelines.

4. Biological task definitions need more domain-specific discussion

The paper includes many cancer datasets with heterogeneous biological contexts. Some classification tasks may be driven by large-scale tissue differences, tumor purity differences, or known confounding factors rather than subtle biological signals. For example, the manuscript sometimes treats related disease categories inconsistently (e.g., colon vs. colorectal cancer datasets), and the biological interpretation of these tasks is not discussed in detail. Domain expertise would strengthen the interpretation of these results.
This means **you need a cancer researcher to review your paper before the final version.**

5. Lack of explanation for cross-dataset variability

While the overall pattern is emphasized, the paper does not sufficiently analyze why some datasets behave differently from others. In particular, cases where extremely small random subsets (e.g., <20 features) match the performance of tens of thousands of features are surprising and deserve deeper investigation. **I hold strong doubts about those results, please review your code and your datasets again.**

If those are real results, please at least provide possible explanations such as: strong class separability,
highly correlated probes, preprocessing artifacts, or dataset-specific biases. Without deeper analysis, it is difficult to assess whether these results reflect genuine redundancy or benchmark artifacts.

6. Practical implications are unclear

The paper argues that predictive performance alone should not justify claims of biological importance, which is reasonable. However, the paper does not clearly articulate what methodological changes researchers should adopt as a result of these findings.

For example:

- Should random-subset baselines become standard in feature selection studies?

- Should the MSRSS metric be reported routinely?

- How should these findings influence biomarker discovery pipelines?

-  Clarifying the actionable implications would increase the paper’s impact.

7. Scope of the claims may be too broad

The title and discussion refer broadly to “high-dimensional biological classification,” but the evidence primarily concerns transcriptomic datasets. Other omics modalities such as SNV, methylation, or proteomics, are not included. The scope of the claims should therefore be narrowed or supported with additional data types.

---

> ### Author Rebuttal · Authors · 2026-03-31
>
> We thank the reviewer for the thoughtful and constructive review. We are encouraged that the reviewer finds the core empirical observation interesting and agrees that the paper raises an important cautionary point for biomarker-style ML claims. We also agree with the reviewer’s overall framing that the paper is strongest as an empirical benchmark / position paper, and that the scope of the claims should be stated more carefully.
>
> **(1) on dataset composition and scope of claims**
>
> We agree that the current benchmark collection is weighted toward transcriptomic datasets, especially microarray. While microarray datasets are older, they remain highly represented in the FS benchmarking literature and therefore are still relevant to the empirical question studied here. Our intended claim is not that the observed phenomenon necessarily holds for all biological high-dimensional data, but that it appears frequently across many widely used high-dimensional biological classification benchmarks, particularly transcriptomic ones. We also have, in the interim, derived results for many other data types (such as SCRNAseq), and our conclusions remain unchanged.
>
> **(2) on preprocessing / normalization / possible artifacts**
>
> As stated in Section 3.1 “None [of the datasets] were imputed before download, and we did not apply an imputation.”  In fact, for tree-based models we did not normalise or scale the data. Importantly, we view this not as a reason to dismiss the phenomenon, but as part of the point: if benchmark datasets are already structured such that many small random subsets perform well, then predictive performance alone becomes even weaker evidence for strong feature-importance claims.
>
> Our intention was to use publicly available benchmark datasets in the form typically used in prior literature, so that our comparisons remain as close as possible to the conditions under which feature selection claims are often made.
>
>
> **(3) on missing univariate statistical baselines**
>
> Thanks for this suggestion. We agree that common univariate filters such as t-test / wilcoxon / anova ranking are important baseline and will include it if space permits, or otherwise report it in the supplement.
>
> **(4) on cross-dataset variability and very small random subsets**
>
> We agree that this is one of the most interesting parts of the paper, and that the current draft does not analyze it deeply enough. We also appreciate the reviewer’s caution here.
>
> We have re-checked the code. We had the exact same worry as the reviewer, so we asked multiple other teams to ***independently*** rewrite all code and re-test everything, with similar results (see the footnote on page 2) and datasets carefully.
>
> Plausible contributing factors include:
> - strong class separability  (see the PCA plot in figure 1)
> - high feature correlation / redundancy (see the conclusion and future work section)
> - task difficulty - mostly binary classification (see the conclusion and future work section)
>
> **(5) on biological interpretation and practical implications**
>
> Our intention is not to claim that feature selection is unnecessary, nor that random subsets are useful biomarkers. Rather, our point is that predictive performance alone is often insufficient to justify claims that a selected feature set is uniquely informative or biologically meaningful.
>
> We are happy to revise the discussion to make the practical implications more explicit. In particular, we agree that the following are reasonable recommendations arising from our results:
> - random-subset baselines should be reported more routinely in feature selection studies,
> - subset-size-matched comparisons should be used when evaluating selected feature sets, and
> - strong biological interpretation should not be based on predictive performance
>
> Overall, we appreciate the reviewer’s suggested revisions which will improve the paper substantially. The main contribution of the work is not claiming that “feature selection is unnecessary,” but rather showing that for many widely used high-dimensional biological benchmarks, predictive performance is often much less informative about feature importance than is commonly assumed.

---

> > ### Author Rebuttal · Reviewer_bQNt · 2026-04-03
> >
> > Thank you for the detailed and well-organized rebuttal. I am good with the responses on dataset composition, the commitment to add univariate baselines, the independent code re-verification, and the clarified practical recommendations in texts; however, I didn't see any data. I will maintain my score of Weak Accept.

---

### Official Review · Reviewer_XYJT · 2026-03-19

**Soundness:** 1
**Presentation:** 2
**Significance:** 1
**Originality:** 1
**Overall Recommendation:** 1
**Confidence:** 5

**Summary:**

This paper works on feature selection in high-dimensional biological classification and investigate whether random feature subsection selection can achieve as good results as modern feature selection methods. The paper claims/concludes that predictive accuracy alone
is not sufficient to justify claims about the importance of specific selected features, and that randomly selected subsets, sometimes comprising as little as 0.02% − 1% of features, can match or exceed the performance of models trained on all features.

**Compliance With Llm Reviewing Policy:**

Affirmed.

**Ethical Review Flag:**

Flag this paper for an ethics review.

**Key Questions For Authors:**

1. Why do you think the observations or conclusions are new? This has been known for many years in feature selection and feature construction.

You claimed that "We find that randomly selected subsets, sometimes comprising as little as 0.02% − 1% of features, can match or exceed the performance of models trained on all features".  This claim is definitely correct, but this is a common sense but definitely not new in the Feature selection community.

2. Why do you only cite very old papers to support your claims, but did not cite (or read) recent feature selection papers in recent years particularly those survey papers or those published after 2020? For example:

Those in https://scholar.google.com/citations?view_op=search_authors&hl=en&mauthors=label:feature_selection

Ruwang Jiao, et al. "A Survey on Evolutionary Multiobjective Feature Selection in Classification: Approaches, Applications, and Challenges". IEEE Transactions on Evolutionary Computation.  DOI: 10.1109/TEVC.2023.3292527

Bing Xue. "A Survey on Evolutionary Computation Approaches to Feature Selection". IEEE Transaction on Evolutionary Computation. Vol. 20, Issue 4, 2016. pp. 606-626.

Binh Tran, Bing Xue and Mengjie Zhang. "Genetic Programming for Feature Construction and Selection in Classification on High-dimensional Data". Memetic Computing. vol 8, Issue 1. 2016. pp. 3-15. DOI: 10.1007/s12293-015-0173-y

Dr Soha Ahmed's work in Biomarker Detection: https://gpbib.cs.ucl.ac.uk/gp-html/SohaAhmed.html

(2) This paper assume a random subset size of 40 or 50 features is small. In many cases, this is true. However, also for many bio/medical data, there are over 50K or 100K or even millions of features available, but only less than 10 features are actually important. This is particularly the case in the areas of biomarker detection. It is very often that only a few features were treated as biomarkers. If you only treat "0.02% or 0.2%" as "small subsets" (your Table 1), this is very misleading ---- for example, 0.02% of a problem with 100,000 features would be 2000 features --- this is too large to be useful.  It is very often only need 5-20 features are considered important by biologists or medical doctors.

In other words, although your claim that only 0.02% of random feature subset could results in good performance is correct, it does NOT mean feature selection is not important or not needed, or further analysis of the feature importance is not needed, or developing new feature selection/construction methods is not needed or not important. So your results are correct, but very misleading.

(3) This paper also said that a small number of features or a small random subset of features can perform as well as the full feature set, which is also correct. However, this is not new at all --- there are so many papers have already found that a much smaller subset of features (than your tested random subsets) could lead to much better accuracy than using full feature set, and much smaller classifiers/models that could interpret by human experts.

**Limitations:**

Please see my comments above.

**Strengths And Weaknesses:**

Strengths: (1) this is a important and hot topic in AI/ML; (2) the results and major conclusions are correct, particularly randomly selected subsets with 0.02% − 1% of features can match or exceed the performance of models trained on all features for high-dimensional bio-data classification. (3) a very good number of datasets and experiments were carried out.

Weaknesses: (1) although the results and some important claims are correct, the major conclusions is very misleading --- this paper gives an impression that feature selection and analysis of the important features are not necessary or not needed.  (2) the assumptions of "randomly selected subsets sometimes comprising as little as 0.02% − 1% of features" was treated or indicated as a small number of features. This is often NOT correct. (3) the claims including a small number of features could achieve good (or as good as) performance or much better performance than full set of features are correct, but not new at all, which reflects the fact that this paper cited very few recent references; (4) most references are quite old, and some important and recent papers (even survey papers were missed). In fact, less than 10% of papers cited were published after 2020 (only one or two out of 43).

---

> ### Author Rebuttal · Authors · 2026-03-31
>
> We thank the reviewer for the careful reading and thoughtful comments. We agree that the current draft can do a better job in (i) positioning the paper relative to recent feature selection literature, and (ii) making the scope of our claims more precise. We will revise the manuscript accordingly.
>
> **(1) on recent feature selection literature:**
>
> Recent works (which are mostly surveys or methodological changes) are naturally underrepresented in our discussion. We are happy to expand the introduction and related work to include recent surveys and representative methodological papers, including the references suggested by the reviewer (e.g. xue et al., 2016; tran et al., 2016; jiao et al., 2023), along with additional recent work on feature selection, stability, and signature multiplicity. However, we emphasise that all of these works do things like suggesting better methods for feature selection: the issue we spot is NOT in this part, but in the downstream assumption that a certain selected set of statistically “important” features are biologically relevant, simply because they provide high predictive accuracy.
>
> Our intention was not to provide a broad survey of all FS methods, since the main goal of this paper is different. We are NOT proposing a new FS method. Instead, we ask a simpler empirical question: ***in high-dimensional biological classification, how much does predictive performance actually depend on the specific selected features?***
>
>
> **(2) on the interpretation of “small” feature subsets and biomarker discovery:**
>
> Our intention is NOT to claim that feature selection is unnecessary, or that very small biomarker sets (e.g. 5–20 features) are unimportant in biological or clinical applications. We are happy to add text to clarify this.
>
> In many biomarker discovery settings, the goal is not only predictive performance, but also compactness, interpretability, biological plausibility, and downstream validation. In such settings, a random subset of dozens or hundreds of features is clearly not a substitute for a carefully validated small biomarker panel. Our point is narrower: across many high-dimensional biological classification benchmarks, we find that predictive performance is often surprisingly insensitive to the specific small subset of features used. This means that predictive accuracy (which often receives a lot of attention and subsequent downstream wet-lab work) alone may not be sufficient evidence that a selected feature set is uniquely informative or biologically meaningful.
>
> We also agree that percentage-based descriptions can be misleading without absolute numbers. To clarify, 0.02% of 100,000 features corresponds to 20 features, not 2000. Table 1 on page 4 has both percentages and absolute numbers.
>
> **(3) on novelty:**
>
> We emphasize that the main novelty of our paper involves testing against a null hypothesis and NOT the general observation that a smaller feature subset can sometimes match or outperform the full feature set. It is that any subset can do so.
>
> We show, across 30 high-dimensional datasets, that a simple null baseline, subset-size-matched random feature subsets, is often unexpectedly competitive across a wide range of datasets and modalities. In many cases, performance across such random subsets is also remarkably stable, suggesting substantial redundancy in predictive signal. We also introduce MSRSS as a way to characterize how quickly predictive performance saturates with random subset size.
>
> In other words, the novelty is not that “small subsets can work,” but that for many high-dimensional biological classification datasets, ***predictive performance is often surprisingly insensitive to WHICH small subset of features is chosen.*** This has direct implications for how predictive success should, and should not, be interpreted in settings where selected features are often treated as biologically meaningful.
>
> Overall, we appreciate these comments. In the revision, we will revise the manuscript to make the scope of the paper clearer, to better cover the recent literature, and to avoid any interpretation that our results imply that feature selection is unimportant.
> We believe these revisions will make the paper’s contribution much clearer: not that feature selection is unnecessary, but that predictive performance alone is often insufficient to justify claims about feature importance in high-dimensional biological classification.

---

> > ### Author Rebuttal · Reviewer_XYJT · 2026-04-04
> >
> > Thanks for your providing responses to my questions and comments. while some of the responses have made limited clarifications, most of the big issues still remain questionable, particularly the choice of the datasets and the comparison with state-of-the-art methods (mostly only with basic methods). More importantly, this paper is misleading, and could mis-lead young researchers to wrong directions. Considering all aspects, I maintain my scores.

---

### Decision · Program_Chairs · 2026-04-30

**Decision:**

Reject

**Comment:**

This paper presents an investigation into how randomly selected features perform on high-dimensional biological datasets for classification tasks. Extensive experiments are conducted on 30 datasets.

The paper presents an interesting empirical observation: classification performance on high-dimensional biological datasets appears relatively insensitive to the choice of feature subsets. It also suggests that sophisticated feature selection methods may not significantly outperform randomly selected subsets. Both observations could be valuable to the machine learning community.
However, the writing is underdeveloped. Most of the weaknesses identified by the reviewers are valid, and the paper is not publishable in its current form.

That said, the authors are encouraged to revise the paper and submit it to a more suitable venue, as the investigation is potentially of interest to the community. Reviewer bQNt has raised a number of valuable questions, and addressing them would significantly improve the quality of the paper.

I have one additional comment. The class distribution should be clearly presented. When a classifier is trained on a random subset of features, its accuracy may approximate the proportion of the majority class. If this is the case, the first conclusion above may not hold, as the classifier would effectively be making a trivial prediction. Furthermore, if the model trained on the feature-selected dataset does not perform substantially better than this baseline, it may indicate that the dataset lacks strong predictive signals. In that case, the second conclusion may also not be valid.